# Remodeling articular immune homeostasis with an efferocytosis-informed nanoimitator mitigates rheumatoid arthritis in mice

Shengchang Zhang[1,9], Ying Liu[1,9], Weiqiang Jing[2,9], Qihao Chai[3,4], Chunwei Tang[1], Ziyang Li[2,3], Zhentao Man[3,4], Chen Chen[1], Jing Zhang[1], Peng Sun[5], Rui Zhang[1], Zhenmei Yang[1], Maosen Han[1], Yan Wang[1], Xia Wei[6], Jun Li[6], Wei Li[3,4], Mohnad Abdalla[1], Gongchang Yu[7], Bin Shi[7], Yuankai Zhang ●[8] ✉, Kun Zhao[1] & Xinyi Jiang ●[1] ✉

Massive intra-articular infiltration of proinflammatory macrophages is a prominent feature of rheumatoid arthritis (RA) lesions, which are thought to underlie articular immune dysfunction, severe synovitis and ultimately joint erosion. Here we report an efferocytosis-informed nanoimitator (EINI) for in situ targeted reprogramming of synovial inflammatory macrophages (SIMs) that thwarts their autoimmune attack and reestablishes articular immune homeostasis, which mitigates RA. The EINI consists of a drug-based core with an oxidative stress-responsive phosphatidylserine (PtdSer) corona and a shell composed of a P-selectin-blocking motif, low molecular weight heparin (LMWH). When systemically administered, the LMWH on the EINI first binds to P-selectin overexpressed on the endothelium in subsynovial capillaries, which functions as an antagonist, disrupting neutrophil synovial trafficking. Due to the strong dysregulation of the synovial microvasculature, the EINI is subsequently enriched in the joint synovium where the shell is disassembled upon the reactive oxygen species stimulation, and PtdSer corona is then exposed. In an efferocytosis-like manner, the PtdSer-coroneted core is in turn phagocytosed by SIMs, which synergistically terminate SIM-initiated pathological cascades and serially reestablish intra-articular immune homeostasis, conferring a chondroprotective effect. These findings demonstrate that SIMs can be precisely remodeled via the efferocytosis-mimetic strategy, which holds potential for RA treatment.

Rheumatoid arthritis (RA), a widespread and devastating systemic autoimmune disease, is characterized by articular immune dysfunction with serious synovitis and joint erosion that causes progressive disability[1,2]. Over the past two decades, multiple immunosuppressants have been approved by the U.S. FDA for RA treatment[3]. Despite the combined use of these immunosuppressants, approximately one-third of patients with RA fail to reach sustained clinical remission[4].

Moreover, the continuous use of immunosuppressive therapeutics often weakens the immune system and increases the risk of infections[5]. The pathogenesis of RA involves complicated inflammatory networks, including multiple cytokine targets and complex crosstalk among inflammatory cells[6,7]. Among these cells, synovial inflammatory macrophages (SIMs) have a pivotal function in orchestrating the cytokine environment by releasing various types of proinflammatory cytokines

and reactive oxygen species (ROS), which are thought to underlie articular immune dysfunction, synovitis, and ultimate joint erosion[8,9]. As such, terminating SIM-initiated cascades and serially reestablishing articular immune homeostasis may be a reversible approach for RA therapy, which remains largely unexplored.

Patients with RA can be divided into two major subsets based on the presence versus absence of anti-citrullinated protein antibodies (ACPAs)[10]. The ACPA⁺ subset of the disease accounts for approximately two-thirds of all cases of RA and generally has a more severe disease course[11]. Interferon regulatory factor 5 (IRF5) is a master regulator in defining the classical inflammatory phenotype of macrophages[12] and translates various signals related to SIMs in the RA synovium[13,14]. In humans, gain-of-function polymorphisms in the *IRF5* gene have been associated with an increased risk of developing autoimmune diseases, including RA[15,16]. Bioinformatically, we found that the expression of IRF5 had a positive correlation with the ACPA titer in the RA synovium (Supplementary Fig. 1a). Consistently, in patients with ACPA⁺ RA, the expression level of IRF5 protein was obviously elevated (Supplementary Fig. 1b) and positive staining of IRF5 overlaid well with positive staining for CD68 or F4/80, which are markers of macrophages (Supplementary Fig. 1c). Targeted silencing of *IRF5* in SIMs may therefore be an efficient strategy that could facilitate the anti-inflammatory polarization of macrophages and thus abort SIM-initiated cascades in ACPA⁺ RA.

Precisely targeting intra-articular macrophages is of utmost importance for efficiently regulating SIM-initiated cascades. The phagocytosis of apoptotic cells, called efferocytosis, is a critical innate function of macrophages, which maintains tissue homeostasis[17,18]. During efferocytosis, phosphatidylserine (PtdSer) exposure on the outer leaflet of the plasma membrane in the apoptotic cells is a key "eat-me" signal for macrophages[19]. PtdSer coronation may therefore enhance the targeted internalization of nanoformulations by macrophages. However, after systemic administration, PtdSer-conjugated nanoformulations may be taken up by pan macrophages, which results in severe adverse effects. Of note, intra-articular oxidative stress caused by high levels of ROS is one of the typical characteristics of RA lesions[20,21]. As such, ROS-responsive exfoliation may be an efficient way to manipulate the locoregional exposure of the PtdSer corona of designed nanoformulations intraarticularly, which enabled the specific phagocytosis of the nanoformulations by SIMs.

Here, we sought to develop an efferocytosis-mimicking self-deliverable nanoimitator for in situ targeted reprogramming of the SIMs and thus thwart their autoimmune attack and reestablish articular immune homeostasis for RA reversal. An siIRF5-carrying efferocytosis-informed nanoimitator (siIRF5@EINI) was for sequentially assembled from a drug-based core with an oxidative stress-responsive phosphatidylserine (PtdSer) corona and an outer shell of low molecular weight heparin (LMWH). With the shielding of LMWH, siIRF5@EINI is endowed with stealth properties in the circulation, enhanced retention in inflamed regions, and a blocking function of P-selectin that retards the articular trafficking of neutrophils. When systemically administered, LMWH on the nanoimitator first binds to P-selectin overexpressed on the endothelium in subsynovial capillaries, which functions as an antagonist, disrupting neutrophil synovial trafficking. Due to leakage of the intra-articular blood vessels, siIRF5@EINI is subsequently enriched in the joint synovium, where the shell is exfoliated upon ROS stimulation, and the PtdSer corona is then exposed. In an efferocytosis-like manner, the PtdSer-coroneted core is in turn phagocytosed by SIMs. We demonstrated that siIRF5@EINI synergistically terminated SIM-initiated pathological cascades and reconstructed intra-articular immune homeostasis and ultimately conferred a chondroprotective effect and restored joint function, all of which were comprehensively investigated in an RA mouse model.

## Results

### Preparation and characterization of EINI

By conjugating 4-nitrophenyl 4-(4,4,5,5-tetramethyl-1,3,2-dioxaborolan-2-yl) benzyl carbonate (NBC) with PtdSer, we first synthesized a ROS-responsive PtdSer-NBC conjugate (Supplementary Figs. 2b, 3b). The efferocytosis-informed nanoimitator was prepared (Fig. 1a). Briefly, dioleoylphosphatydic acid (DOPA)-stabilized siIRF5-laden metformin-Zn²⁺-based nanocores were constructed using the reverse microemulsion method, and then coated with PtdSer-NBC. Subsequently, the corona of the nanocore was cloaked with LMWH via a reversible boronate ester linker by the reaction of cis-diols and phenylboronic acid groups, and the siIRF5-laden efferocytosis-informed nanoimitator (siIRF5@EINI) was thus generated. Under transmission electron microscopy (TEM), the obtained nanoimitator showed a spherical morphology (Fig. 1b). Dynamic light scattering (DLS) measurements showed that the nanoimitator had a hydrodynamic diameter of ~72 nm (Fig. 1c). After dual coating with PtdSer-NBC and LMWH, the zeta potential of the nanoformulation decreased from $44.5 \pm 0.4$ mV to $-18.7 \pm 0.03$ mV (Supplementary Fig. 4). In terms of siRNA loading, the encapsulation efficiency was quantified by incorporating Cy5.5-labeled siRNA at increasing concentrations (Supplementary Fig. 5). It was demonstrated that siRNA could be incorporated with high efficiency over a wide range of inputs, and siIRF5@EINI nanoparticles were fabricated using 20 μM siRNA, 30 mM $Zn(NO_3)_2 \cdot 6(H_2O)$, and 30 mM metformin for subsequent studies. We next investigated the stimulation-triggered presentation of PtdSer by siIRF5@EINI in an RA-mimicking microenvironment. As shown in Fig. 1d and Supplementary Fig. 6, 66.8% of the PtdSer-presenting nanoparticles were determined to bind to fluorescein isothiocyanate (FITC)-annexin V in a solution of 0.1 mM $H_2O_2$, while this value was merely 19.8% in the PBS control group, indicating that oxidative stress in RA tissues could convert PtdSer-NBC of the nanoimitator into PtdSer.

Next, we evaluated the acid-triggered drug release of the nanoimitator in vitro. The morphology of the nanoimitator exhibited significant degradation into ultrasmall species in acidic environments, as shown by TEM images (Fig. 1e). Structural disassembly was further confirmed with a DLS assay (Supplementary Fig. 7). Consistently, we found that under acidic conditions (pH 5.0), the release of metformin from the nanoimitator increased compared to that under physiological conditions (pH 7.4) (Fig. 1f). With an electrophoresis assay (Fig. 1h), the siIRF5 release profile was also monitored. At a lysosomal pH of 5.0, the siIRF5 burst released from the nanoformulation, while at a physical pH of 7.4, we did not detect any release of siIRF5 from the siIRF5@EINI. The ROS-responsive exfoliation of the EINI was further investigated with a $H_2O_2$ scavenging assay. The ROS-treated nanoimitator exhibited a much higher LMWH release efficiency (Supplementary Fig. 8), which may be attributed to ROS-responsive exfoliation of the nanoimitator. Figure 1g shows a time-dependent ROS elimination of the nanoimitator upon ROS stimulation, implying ROS-stimulated deshielding of the nanoimitator under RA pathological conditions.

### Cellular engulfment specificity of the nanoimitator in vitro

PtdSer exposure triggers the efferocytosis of macrophages. We next investigated the engulfment and intracellular trafficking of the nanoformulation in vitro using bone marrow derived macrophages (BMDMs) that were isolated from mice (Supplementary Fig. 9). BMDMs were pre-activated with TNF and then treated with the nanoimitator preconditioned with 0.1 mM $H_2O_2$. As shown in Fig. 2a, significant fluorescence was detected in macrophages, indicating that abundant nanoimitators were engulfed into macrophages. With mouse fibroblast-like synoviocytes (FLSs) (Supplementary Fig. 10) activated with TNF as a control, we also determined the cellular engulfment specificity of the nanoimitator. A minimal Cy5.5 signal was detected in FLSs (Fig. 2a). The results indicate that the EINI was specifically

engulfed by macrophages when ROS were present. Moreover, under RA-mimicking conditions, increased fluorescence intensity compared to that under physiological microenvironment was observed in macrophages. A similar trend was found in flow cytometry quantification (Fig. 2b, c). With a coculture system, we further compared the uptake of the nanoimitator by the pre-activated macrophages or FLSs under RA-mimetic condition. Fluorescence visualization with confocal laser scanning microscopy (CLSM) also indicates the specific engulfment of the PtdSer-coroneted core by macrophages rather than FLSs (Fig. 2d). We further evaluated the specific engulfment of nanoimitators by macrophages in vitro using human FLSs isolated from the synovium of RA patients (Supplementary Fig. 10) and healthy human peripheral blood-derived macrophages (HPBDMs) harvested and differentiated

with consent. Consistently, the results indicated that there was nearly no uptake of the nanoimitator (red) by PKH67-labeled FLSs (green) (Supplementary Fig. 11), which confirmed the specific uptake of the nanoimitator by macrophages.

Next, we wished to ascertain whether the siIRF5-laden nanoimitator would escape lyso/endosomes and release the siRNA into the cytoplasm after efficient uptake by macrophages. Co-localization with CLSM showed that the nanoimitators were internalized by the cell via an endocytosis pathway and then escaped from lyso/endosomes, during which the siRNA was released into the cytoplasm (Fig. 2e). These observations were confirmed by quantitative analyses of the co-localization of Cy5.5-siIRF5 with endo/lysosomes in confocal fluorescence images using Manders' coefficients M1 and M2[22]. As shown in

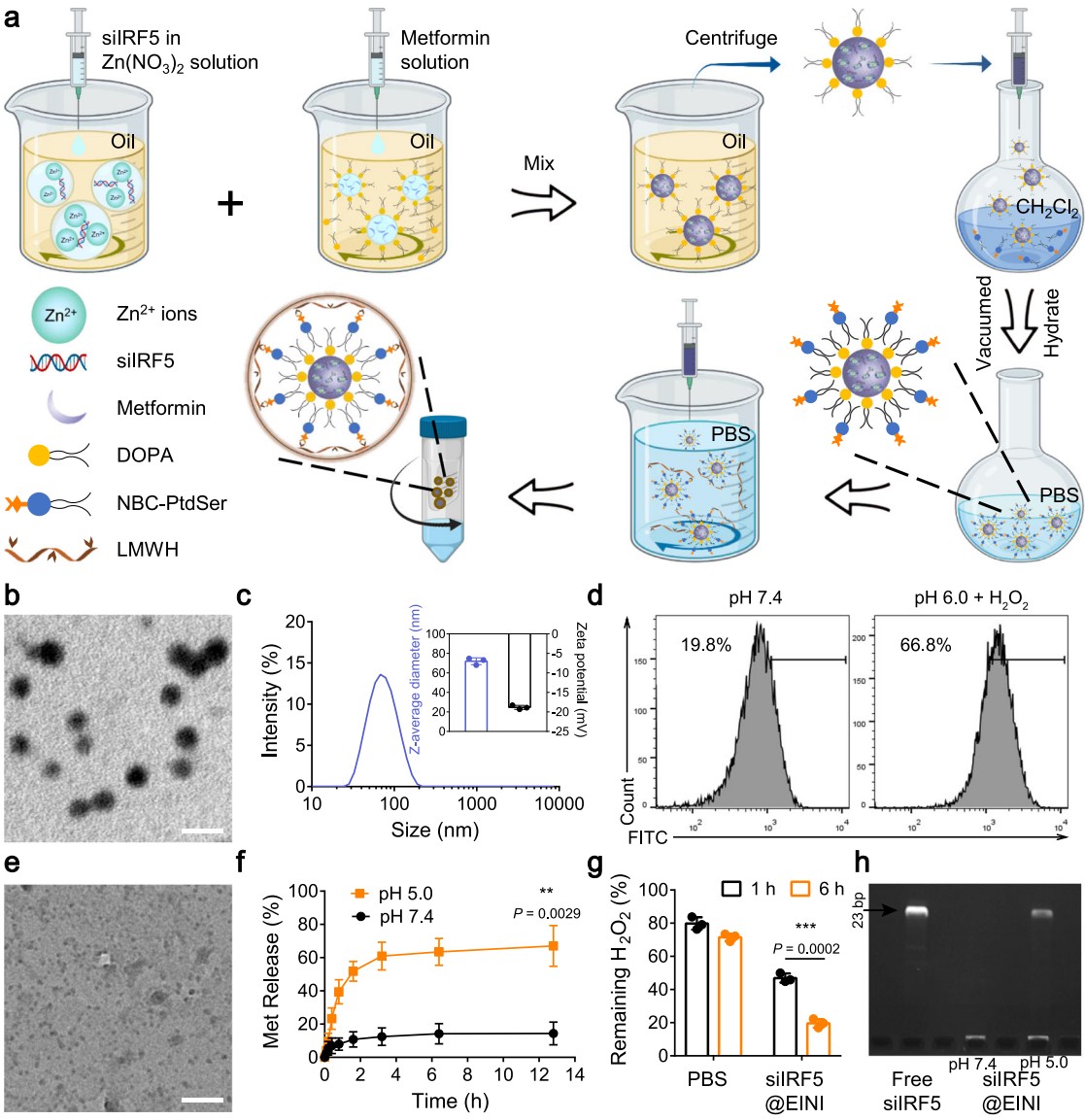

**Fig. 1 | Preparation and characterization of siIRF5@EINI. a** Schematic of the preparation of the nanoimitator by a modified reverse microemulsion method. **b** Transmission electron microscope image of siIRF5@EINI (n = 3 independent experiments). Scale bar = 100 nm. **c** Hydrodynamic size and zeta potential determined with DLS. Data are presented as the mean ± s.d. (n = 3 independent experiments). **d** Flow cytometry analysis of PtdSer-presenting with FITC-Annexin V treatment. (n = 3 independent experiments). **e** TEM images of the nanoimitator after degradation triggered under acidic conditions (pH 5.0) (n = 3 independent experiments). Scale bar = 100 nm. **f** Drug release of the nanoimitator in PBS at

different pH values. Data are presented as the mean ± s.d. (n = 3 independent experiments). (exact P value: P = 0.0029); **P < 0.01. **g** ROS scavenging ability of the nanoimitator. The results are reported as the mean ± s.d. (n = 3 independent experiments). The P value was calculated by comparison with the PBS group. (exact P value: P = 0.0002); ***P < 0.001. **h** Electrophoretic gel assay showing the pH-responsive release of siIRF5 from the nanoimitator. The experiments were repeated three times independently. Statistical significance was determined by a two-sided Student's t test in **f** and **g**. Source data are provided as a Source Data file.

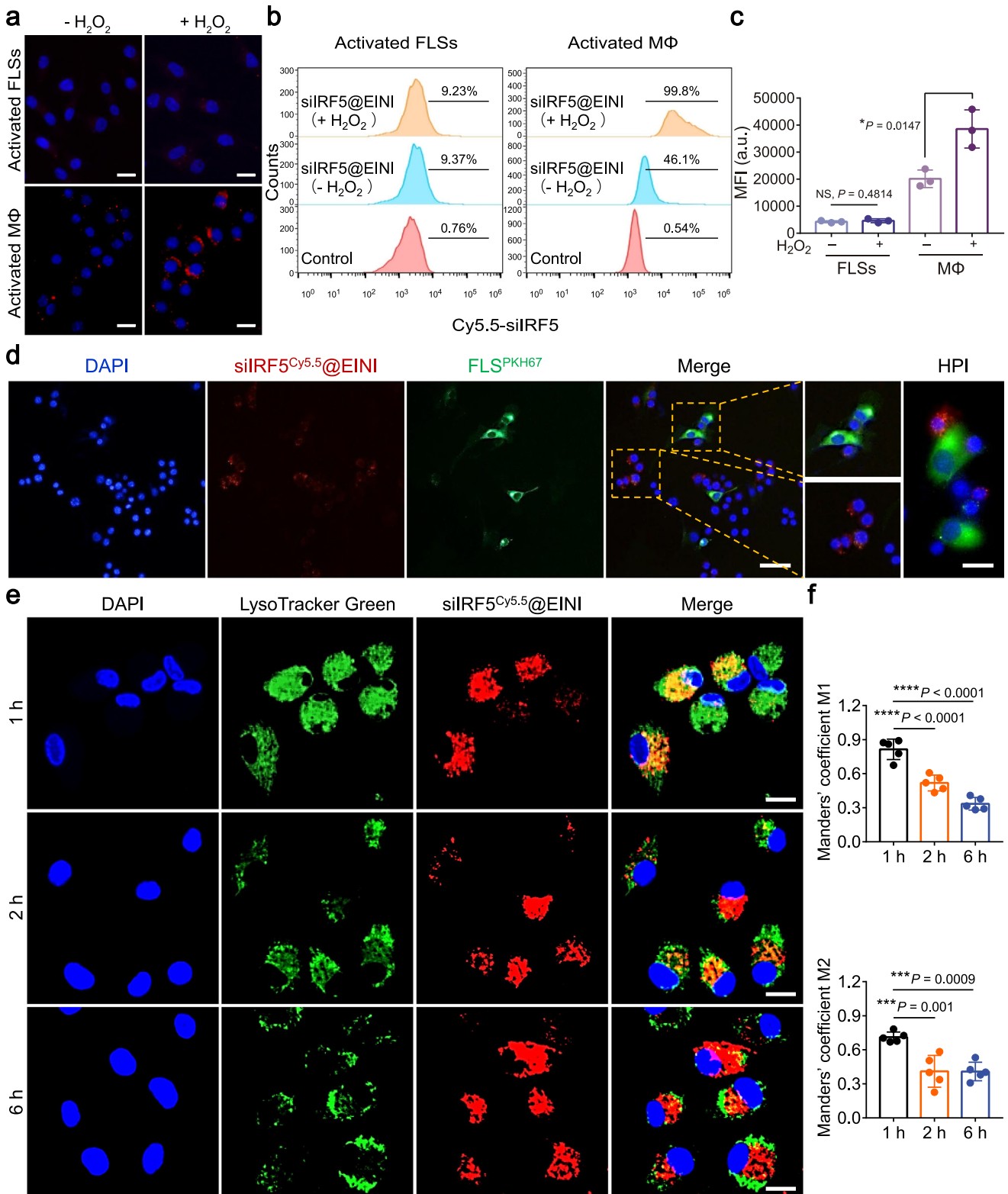

Fig. 2f, M1 is close to 1 at 1 h, indicating that the nanoimitators were taken up via endocytosis and stayed in endo/lysosomes. In contrast, M1 is much <1 at the other time points (particularly 6 h), indicating that the nanoimitators can achieve effective endo/lysosomal escape of the encapsulated siRNA. The M2 data show that nearly all of the endo/lysosomes were not merged with Cy5.5-siIRF5 at 6 h. Collectively, these data suggest that the low-pH-triggered structural disassembly of the nanoimitator facilitates endo/lysosomal escape for cytosolic delivery

of siRNAs. In summary, these results indicated that the ROS-responsive exposure of PtdSer in siIRF5@EINI-induced efficient phagocytosis by macrophages, and the lysosomal pH-triggered dissociation of the nanocarrier resulted in successful lyso/endosome escape.

## Inflammatory modulation by the nanoformulation

Comprehensively regulating the production of proinflammatory cytokines, oxidative stress, and recruitment of neutrophils is essential

**Fig. 2 | Cellular uptake and intracellular distribution of the nanoformulation.**
**a** Fluorescence images of macrophages and FLSs after incubation with siIRF5@EINI that with or without $H_2O_2$ pretreatment. Cells were activated with TNF before nanoformulation incubation ($n = 3$ independent experiments). Scale bar, 20 μm.
**b** Representative flow cytometric analysis of nanoimitators internalized by TNF activated macrophages or FLSs. **c** Quantitative analysis of nanoimitator internalized by TNF activated macrophages or FLSs. Data are presented as the mean ± s.d. ($n = 3$ independent experiments). (exact P values: $P = 0.4814$, $P = 0.0147$); *$P < 0.05$. NS, not significant. **d** Representative images of $H_2O_2$-pretreated siIRF5@EINI (red) engulfed by macrophages (not labeled)/FLSs (green) in a coculture pattern ($n = 3$ independent experiments). Scale bar, 50 μm. The right panel shows representative high-power images (HPI). Scale bar, 20 μm. **e** Fluorescence visualization of siRNA localization in macrophages 1, 2, or 6 h after incubation with siIRF5@EINI. The cell

nuclei were stained using DAPI (blue), the endo/lysosomes were stained using LysoTracker Green (green), and siIRF5 was labeled with Cy5.5 (red). Scale bar, 20 μm. **f** Quantitative analysis of the co-localization of Cy5.5-siIRF5 with endo/lysosomes labeled with LysoTracker Green. Manders' coefficient M1 denotes the fraction of Cy5.5-siIRF5 overlapping with LysoTracker Green, and M2 denotes the fraction of LysoTracker Green overlapping with Cy5.5-siIRF5. The coefficients are close to 1 if they are highly co-localized ($n = 5$ images from three independent experiments). Data are shown as the mean ± s.d. (exact P values: M1: $P = 9.06631E-05$, $P = 6.65959E-07$; M2: $P = 9.4541E-04$, $P = 9.76757E-04$); ***$P < 0.001$, ****$P < 0.0001$. Statistical significance was determined by a two-sided Student's $t$ test in **c** and one-way ANOVA with Tukey's post hoc test in **f**. Source data are provided as a Source Data file.

for reestablishing articular immune homeostasis. LMWH can block the P-selectin-initiated cell adhesion cascade, and thus inhibit neutrophil migration into inflamed regions[23]. We next evaluated the blocking effect of the nanoimitator with a monolayer model of TNF-activated human umbilical vein endothelial cells (HUVECs). The results showed that the adhesion of neutrophils to HUVECs was weakened after pre-incubation with the nanoimitator. (Fig. 3a, Supplementary Fig. 13, 14). To further confirm that the nanoimitator retards the articular trafficking of neutrophils, immunofluorescence of synovial slices was used to evaluate inflammatory neutrophil infiltration. We found that the iv injected nanoimitator could adhere to the blood vessel walls of the synovium (Fig. 3b). In contrast, control nanoparticles did not specifically attach to the synovial vasculature. Through its capability to bind to P-selectin overexpressed on the endothelium in RA subsynovial capillaries, the nanoimitator thus interrupted the endothelium tethering of neutrophils and prevented their subsequent extravasation. We further examined the gene silencing efficiency of the nanoformation after uptake by macrophages. As shown in Fig. 3c, the relative level of IRF5 mRNA in TNF-activated BMDMs showed a dramatic decrease after treatment with siIRF5@EINI, which is consistent with the protein data determined by western blotting (Supplementary Fig. 15). IRF5 has a critical function in the secretion of multiple chemokines, including the macrophage-derived chemokine CXCL1, a neutrophil attractant[14]. We also detected the level of CXCL1 secreted from the treated macrophages. As shown in Fig. 3d, treatment with siIRF5@EINI dramatically downregulated macrophage-derived CXCL1 expression. Furthermore, neutrophil migration was also examined. We found that *IRF5* ablation significantly inhibited the transendothelial migration of neutrophils (Fig. 3e). Therefore, we inferred that the nanoimitator may function as an antagonist disrupting the synovial trafficking of neutrophils.

Macrophages are phenotypically plastic[24]. The expression of IRF5 in macrophages defines the classical inflammatory phenotype[12]. We next evaluated the modulatory effect of the nanoformulations on the phenotypic conversion of macrophages. As shown in Fig. 3f, treatment with siIRF5@EINI reduced the mRNA expression level of proinflammatory markers including tumor necrosis factor (TNF) and inducible nitric oxide synthase (iNOS), while significantly increasing the levels of anti-inflammatory markers, such as mannose receptor (CD206) and arginase-1 (Arg-1). Similar results were obtained in the images from immunofluorescence staining of BMDMs (Fig. 3g, h). Notably, treatment with a nanoformulation carrying metformin along resulted in a strong increase in the size of the M2-like subpopulation with high expression of CD206. Concurrent RNAi-mediated gene silencing and metformin-mediated metabolic reprogramming further increased the immunoregulatory phenotypic conversion of macrophages. Metformin can target the electron transport chain of mitochondria and thus reprogram mitochondrial metabolism[25,26]. As shown in Supplementary Fig. 16a, b, metformin-containing treatment significantly prevented excessive ROS production, which synergistically led to immunoregulatory phenotypic conversion. ROS-generating

efficiency was also evaluated by the determination of the $NAD^+/NADH$ ratio. As shown in Supplementary Fig. 16c, a significant reduction in the $NAD^+/NADH$ ratio was detected in macrophages treated with the metformin-containing formulation. These results suggested that siIRF5@EINI could promote the reprogramming of macrophages to an immunoregulatory phenotype and thus significantly reduce macrophage-derived proinflammatory cytokine levels, which is conducive to reestablishing articular immune homeostasis.

## Articular localization of the nanoformulation in vivo

A collagen-induced arthritis (CIA) mouse model based on DBA/1J mice was first established as previously reported[27,28]. The in vivo biodistribution and pharmacokinetic properties of the self-deliverable nanoimitator were next evaluated. As shown in Fig. 4a, the nanoimitator was efficiently deposited in the inflamed joints of the diseased mice 96 h post intravenous administration. In contrast, the fluorescence signal arising from the paws of nanoimitator-injected non-arthritic mice reached a maximum in the initial 3 h and decayed very rapidly (Supplementary Fig. 17). The major organs and paws of the CIA mice from each group were collected 96 h post-injection and subjected to fluorescence analysis ex vivo. siIRF5@EINI was prominently enriched in the arthritic area of CIA mice (Fig. 4b). Quantification of the fluorescence intensity further confirmed a significant increase in the mean fluorescence intensity (MFI) in siIRF5@EINI-treated mice compared with that in the free drug-treated group (Fig. 4c). The retention profile of the nanoimitator in the synovium of arthritic joints was further evaluated. Seventy-two hours after each treatment, the CIA mice were sacrificed, and the synovium cryosections were obtained. Figure 4d, e shows that the nanoimitator was preferentially enriched in the inflammatory synovial sites compared to the free dye-treated group. Further immunofluorescence staining demonstrated that Cy5.5-labeled siIRF5 co-localized with the synovial macrophages (Fig. 4f), indicating that the nanoimitator could efficiently accumulate intraarticularly and specifically target synovial macrophages in situ.

To further investigate the effects of nanoimitator targeting on inflammation, we constructed another acute inflammatory model, a mouse model of DSS-induced acute colitis, according to a previous protocol[29]. We assessed the targeting ability of nanoimitators to inflammation sites by intravenous injection of Dir-labeled nanoimitator into mice after DSS administration. The administered nanoimitator accumulated in the DSS-inflamed colon; however, the nanoimitator was not detected in the colon of healthy mice (Supplementary Fig. 18a, b). Consistently, we also observed a similar trend in inflammation targeting in the cystitis model (Supplementary Fig. 18c, d). In summary, the proposed efferocytosis-informed nanoimitator could be broadly applicable for inflammation-targeted drug delivery.

The potential toxicity of the systemically administered nanoformulation was also tested. No adverse effects were detected in the hematological assay of the siIRF5@EINI-treated mice compared with normal mice in terms of white blood cell (WBC) counts, including

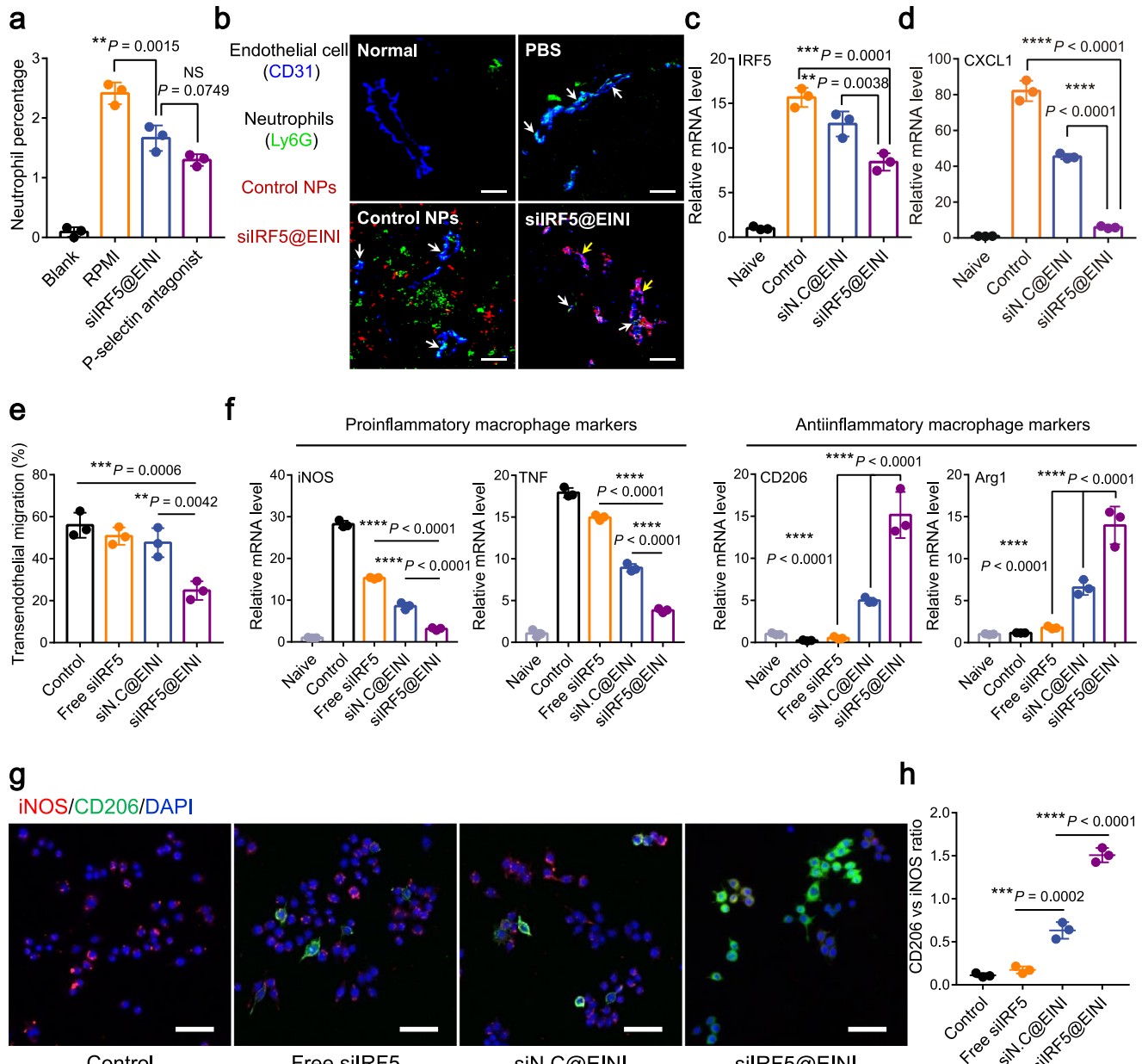

**Fig. 3 | Inflammatory modulation by the nanoimitator. a** Quantitative analysis of the adhered neutrophils on HUVEC monolayers. RPMI culture medium was used as a negative control. Data are presented as the mean ± s.d. (*n* = 3 independent experiments). (exact *P* values: *P* = 0.0015, *P* = 0.0749); **\**P* < 0.01. NS, not significant. **b** Immunofluorescence images of neutrophil infiltration in the synovium after treatment with PBS, control nanoparticles (control NPs), or siIRF5@EINI. Neutrophils were immunostained with an anti-Ly-6G antibody (green) and endothelial cells were stained using CD31 (blue). The nanoparticles were visualized by siRNA labeled with Cy5.5 (red). White arrows indicate firmly adhered neutrophils. Yellow arrows indicate the colocalization of the nanoimitator and endothelial cells labeled with anti-CD31 antibodies (*n* = 3 independent experiments). Scale bars, 50 μm. **c**, **d** IRF5 (**c**) and CXCL1 (**d**) mRNA expression levels in activated macrophages treated with each formulation. Data are shown as the mean ± s.d. (*n* = 3 independent experiments). (exact *P* values: IRF5: *P* = 0.0001, *P* = 0.0038; CXCL1: *P* = 8.58116E-09, *P* = 9.69073E-07); **\**P* < 0.01, ***\**P* < 0.001, ****\**P* < 0.0001. **e** The ratio of transendothelial neutrophils to all neutrophils after intervention with different treatments in the HUVEC monolayer model. The monolayers of HUVECs were stimulated with 10 ng/ml TNF. Data are shown as the mean ± s.d. (*n* = 3 independent experiments). (exact *P* values: *P* = 0.0006, *P* = 0.0042); **\**P* < 0.01, ***\**P* < 0.001. **f** qRT-*P*CR analysis of the phenotypic changes of macrophages that were pretreated with the nanoimitator in vitro. Data are presented as the mean ± s.d. (*n* = 3 independent experiments). (exact *P* values: iNOS: *P* = 6.04161E-10, *P* = 2.42334E-06; TNF: *P* = 1.87678E-10, *P* = 3.52302E-07; CD206: *P* = 3.9947E-07, *P* = 1.18656E-05; Arg-1: *P* = 6.26672E-07, *P* = 6.13112E-05); ****\**P* < 0.0001. **g, h** Immunostaining of macrophages with different phenotypes (**g**) and quantitative analysis of the relative fluorescence intensity of CD206⁺ staining versus iNOS⁺ staining in each group (**h**). Cells were pretreated with TNF and then incubated with each formulation for 24 h. Data are presented as the mean ± s.d. (*n* = 3 independent experiments). (exact *P* values: *P* = 0.0002, *P* = 1.28372E-06); ***\**P* < 0.001, ****\**P* < 0.0001. Scale bars, 50 μm. Statistical analysis was performed using one-way ANOVA with Tukey's post hoc test for **a**, **c**–**f**, **h**. Source data are provided as a Source Data file.

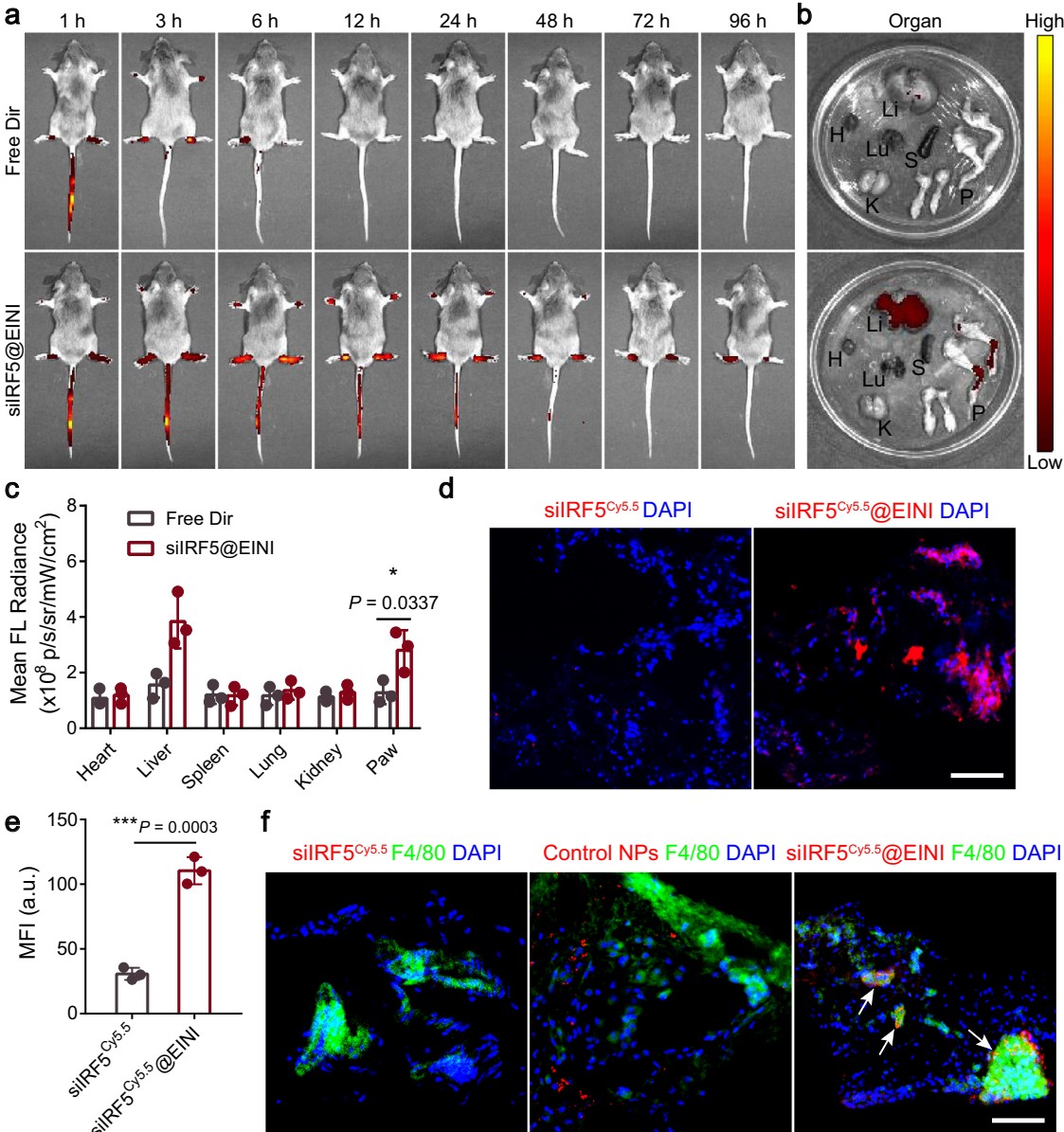

**Fig. 4 | The EINI was selectively enriched in the inflamed joints of CIA mice. a** In vivo fluorescence images of CIA model mice taken at different time points post-injection with Dir-labeled EINI and free Dir. **b** Fluorescence images of the excised major organs and paws harvested from the mice at 96 h post-injection. H, Heart; Li, Liver; S, Spleen; Lu, Lung; K, Kidneys; P, Paws. **c** Quantification of fluorescence intensity in the inflamed joints and major organs of CIA mice. Data are the mean ± s.d. (*n* = 3 biologically independent animals per group). (exact *P* values: *P* = 0.0337); *\*P* < 0.05. **d** Trafficking profile of the nanoimitator in the joint synovium. Red represents nanoimitators and blue represents nuclei. Scale bars, 50 μm.

**e** Quantitative analysis of the accumulation of the nanoformulation in the synovium of the treated mice. Data are presented as the mean ± s.d. (*n* = 3 independent experiments). (exact *P* values: *P* = 0.0003); *\*\*\*P* < 0.001. **f** Representative images of the nanoimitator taken up by synovial macrophages in vivo. Blue, DAPI-stained cell nuclei; Red, siIRF5$^{Cy5.5}$; Green, FITC-conjugated anti-F4/80 antibodies-labeled macrophages. The arrow indicates the co-localization of the nanoimitator and SIMs labeled with anti-F4/80 antibodies (*n* = 3 independent experiments). Scale bars, 50 μm. Statistical significance was determined by a two-sided Student's *t* test in **c** and **e**. Source data are provided as a Source Data file.

counts of neutrophils, lymphocytes, and monocytes (Supplementary Fig. 19a). Similarly, the nanoimitator did not affect the red blood cell (RBC) count or the concentration of hemoglobin (HGB). Splenocytes were also isolated and evaluated. As shown in Supplementary Fig. 19b, no obvious apoptosis was detected in the treated mice. Histological analysis of organs coincidentally informed upon the biosafety of the nanoimitator in vivo (Supplementary Fig. 19c). Next, we examined whether the administration of nanoimitatosr compromises the host immune defense against *Candida albicans. C. albicans*, an opportunistic pathogen with a high fatality rate, has shown an increased infection rate in immunocompromised individuals under a variety of conditions[30,31]. A dosing scheme similar to the therapeutic regimen in

the CIA mouse model was used: 6-week-old DBA/1J mice received an intravenous injection of the nanoimitator (1 nmol of siRNA dose per mouse, 10 mg/kg metformin-equivalent dose, *n* = 3) twice a week for a total of 5 weeks. Sterile PBS, as a negative control, was injected intravenously into mice on the same days. Methotrexate (5 mg/kg) was used as a positive control for immune suppression[32,33] and injected intravenously into mice twice times a week from day 28 to 60. For the evaluation of immune defense, nanoimitator-treated and untreated DBA/1J mice were injected intravenously with 1 × 10⁵ conidia *C. albicans* CA4 after the last administration.

No difference was seen in the fungal load in the spleen between mice treated with the nanoimitator and those in the PBS group

(Supplementary Fig. 20a, b). Importantly, the nanoimitator did not affect the proportions of CD19+, CD4+, or CD8+ lymphocytes in the blood and spleen (Supplementary Fig. 20c), which is often associated with therapeutics for autoimmune diseases[34]. However, methotrexate reduced the sizes of lymphocyte subpopulations in the blood and spleen. Together, administration of the nanoimitator did not significantly affect the normal adaptive immune responses of the treated mice.

## EINI-enabled reestablishment of immune homeostasis and chondroprotection in vivo

We next investigated the therapeutic potency of the nanoformulations using the abovementioned CIA mouse model (Fig. 5a). At the study endpoint, the transverse ankle diameter of mice injected with siIR-F5@EINI was significantly smaller than that of the PBS group, indicating a reduction in inflammation and edema (Supplementary Fig. 21). Meanwhile, the paws of mice receiving PBS showed a natural disease progression and developed severe swelling; this effect was significantly lessened for mice treated with siIRF5@EINI, as quantified by measuring the hindpaw volume (Fig. 5b). Moreover, blinded scoring of the swelling and redness of mouse paws was conducted to evaluate the severity of arthritis in the experimental mice. siIRF5@EINI-treated mice had significantly lower arthritis scores than the other groups, indicating the best treatment effect (Fig. 5c). We subsequently performed gait analysis of the treated mice using a catwalk system. As shown in Supplementary Fig. 22a, siIRF5@EINI treatment markedly ameliorated the abnormal footprint patterns of the CIA mice compared with the control mice. Regarding the gait analyses, the siIRF5@EINI-treated group showed significant improvements in various gait indices for the mice in this model, including a decrease in stride frequency, an increase in stride length, and improved stance times (Supplementary Fig. 22b). *IRF5* gene silencing efficiency was also confirmed in vivo in CIA mice. Treatment with the siIRF5-laden nanoformulation significantly downregulated its IRF5 in macrophages (Supplementary Fig. 23) isolated from the arthritic joint (Supplementary Fig. 24). Consistently with the in vitro results, *IRF5* silencing reduced the secretion of macrophage-derived CXCL1 in the inflamed joint (Supplementary Fig. 25), which in turn decreased the recruitment of neutrophils. We further detected the infiltration of neutrophils in the inflamed joint by immunohistochemical staining with anti-myeloperoxidase (MPO). As shown in Figs. 5d and 5h, the siIRF5@EINI-treated group showed minimal MPO expression with few positively stained cells in the inflamed sites, while many MPO-positive cells were detected in the control joints. Competitive binding of P-selectin on vascular endothelial cells can decrease the infiltration of neutrophils into arthritic joints and thus promote restoration of the articular immune microenvironment[23,35]. More importantly, treatment with siIRF5@EINI synergistically induced phenotypic alterations in synovial macrophages. Immunofluorescence analysis indicated that the proportion of the M1 subpopulation was significantly decreased, and the proportion of the M2 subpopulation was markedly increased in the siIRF5@EINI group (Fig. 5e, i). Phenotypic changes in the synovial macrophage subpopulation were further explored using flow cytometry analysis. Consistently, an increased population of macrophages with the M2 phenotype was also observed after siIRF5@EINI treatment, demonstrating that siIRF5@EINI induced an M2-phenotype shift in synovial macrophages (Supplementary Fig. 26). These results confirmed the very potent immunomodulatory activity of the nanoimitator in vivo and with a subsequent decrease in the unchecked infiltration of neutrophils in RA joints which was conducive to reestablishing the articular immune homeostasis.

We next wished to ascertain whether systemic injection of the efferocytosis-informed nanoimitator would control synovitis and reverse bone erosion in a CIA mouse model. In hematoxylin and eosin (H&E)-stained sections, an almost completely recovered articular cavity with a clear interface and without obvious synovitis and articular cartilage degeneration was detected in the siIRF5@EINI-treated mice (Fig. 5f, j), while severe pathological changes, including extensive inflammatory cell infiltration, synovial hyperplasia, pannus formation, and cartilage erosion, were observed in the control group. Pannus formation in the synovial intimal lining is a characteristic of the pathological change of RA[36]. Synovitis creates an oxidative stress microenvironment that actively induces a tumor-like invasive pannus and further exacerbates joint damage[20]. As shown in Supplementary Fig. 27a, angiogenesis in the synovial sublining was significantly reduced in the siIRF5@EINI-treated groups compared with the others. These pathogenetic changes in the synovium implied the reversal of pannus progression. Additionally, safranin O staining showed optimal preservation of the cartilage structure of the mice in the siIRF5@EINI-treated group (Fig. 5g, k). Consistent with the therapeutic efficacy results, tartrate-resistant acid phosphatase (TRAP) staining of osteoclasts further demonstrated that siIRF5@EINI treatment significantly decreased the number of osteoclasts in RA joints and achieved the least severe bone erosion (Supplementary Fig. 27b). CT was also performed to assess bone erosion. As shown in Fig. 5l, the reconstructed micro-CT images revealed that both the ankle and toe joints of the mice in the control groups suffered the most serious bone corrosion. Quantitative analyses of bone mineral density (BMD) demonstrated that the mean BMD in siIRF5@EINI-treated mice was almost completely recovered compared with that in healthy mice (Fig. 5m). To evaluate the potential systemic immune response, we further examined the serum levels of TNF and IL-1β in CIA mice. As summarized in Fig. 5n, the levels of these inflammation-related cytokines decreased after treatment, indicating an effective reduction in inflammation at the systemic level. These results collectively suggested that the efferocytosis-informed nanoimitator could effectively repolarize SIMs to an immunoregulatory phenotype in situ, serially reestablish articular immune homeostasis, and ultimately terminate bone erosion.

## Discussion

The current standard-of-care therapies in the clinic for patients with rheumatoid arthritis (RA) are palliative, and they still largely rely on multiple immunosuppressants[3]. Despite trying multiple immunosuppressants, approximately one-third of RA patients fail to achieve disease remission[4]. Most available immunosuppressant therapies suppress the immune system systemically, which increases the risk of infection[5]. Although the precise mechanism of RA remains elusive, the latest clinical pathogenic insights indicate that extensive infiltration of inflammatory cells occurs during RA formation and aggravation[37]. Of these cells, SIMs are key players involved in the secretion of various proinflammatory cytokines, leading to unchecked neutrophil transmigration and initiating osteoclast-dependent cartilage destruction and bone erosion[8,9,38]. To precisely reprogram SIMs for RA treatment, we developed the efferocytosis-informed nanoimitator (EINI) to terminate SIMs-initiated pathological cascades, which could serially reestablish intra-articular immune homeostasis, conferring a chondroprotective effect, and ultimately restoring joint function.

RA is a systemic autoimmune disease[1]. Symmetrical inflammatory polyarthritis is the primary clinical manifestation, usually beginning in the small joints of the hands and feet and spreading later to the larger joints[39]. Tracking and targeting the inflamed joint synovium is of utmost importance for the treatment of this systemic autoimmune disease. Efferocytosis is one of the critical innate functions of macrophages, which maintains tissue homeostasis[17,18]. PtdSer exposure on the outer leaflet of the plasma membrane in apoptotic cells is a key "eat-me" signal for macrophages[19]. The recognition of PtdSer by macrophages mainly depends on its head group[40]. We demonstrated that the PtdSer-coroneted core boosted internalization by macrophages both in vitro (Fig. 2a) and in vivo (Fig. 4f). Surficial painting of LMWH on the PtdSer-coroneted core reduced the innate vulnerability of PtdSer to phagocytosis by the mononuclear phagocyte system in

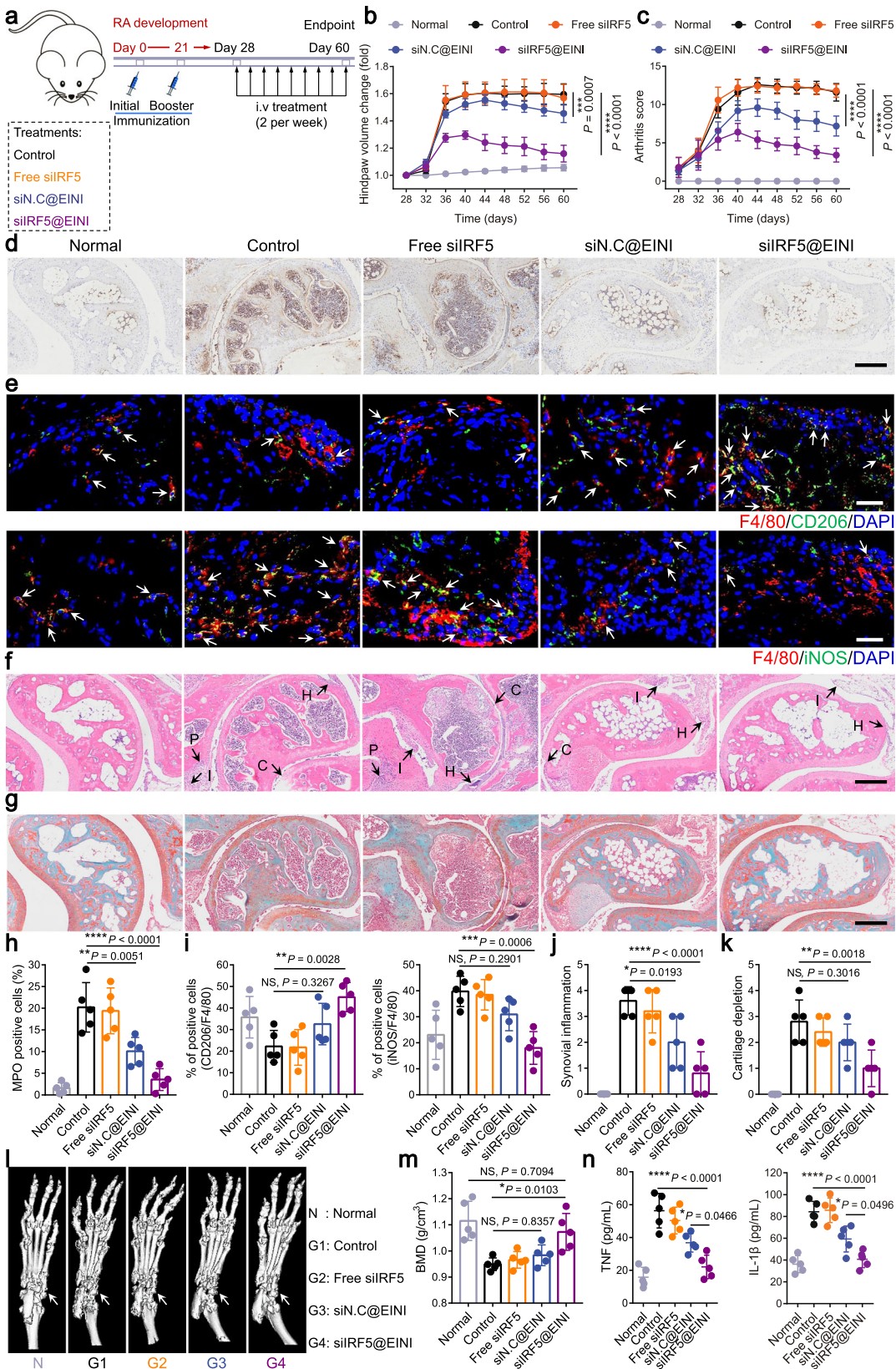

the bloodstream, thus avoiding rapid clearance (Fig. 4a). Moreover, the LMWH motif endowed EINI with enhanced retention in inflamed regions (Fig. 4b, c) and a competitive inhibiting function toward P-selectin[23] that disrupts the unchecked articular trafficking of neutrophils (Figs. 3b and 5d). Due to the leakage of the intra-articular blood vessels, EINI was subsequently enriched in the joint synovium,

where the shell was exfoliated upon ROS stimulation, and the PtdSer corona is then exposed. In an efferocytosis-like manner, the PtdSer-coroneted core was in turn phagocytosed by SIMs (Fig. 4f).

Gain-of-function polymorphisms in the *IRF5* gene are associated with an increased risk of developing RA[15]. *IRF5* translates various signals related to proinflammatory macrophages in the

**Fig. 5 | siIRF5@EINI-enabled reestablishment of immune homeostasis and chondroprotection in vivo. a** Therapeutic regimen in a CIA mouse model. **b, c** The severity of arthritis was determined via (**b**) the relative paw volume changes over time of the CIA mice and (**c**) an arthritis scoring system. Data are presented as the mean ± s.d. ($n$ = 5 biologically independent animals per group). (exact $P$ values: paw volume change: $P$ = 0.0007, $P$ < 1.0E-15; arthritis score: $P$ = 3.32291E-09, $P$ < 1.0E-15); ***$P$ < 0.001, ****$P$ < 0.0001. **d** The infiltration of neutrophils in arthritic joints in each group. Joint sections in each group were stained with anti-myeloperoxidase (MPO) antibodies. Scale bar, 200 μm. ($n$ = 5 biologically independent animals per group). **e** Synovium sections were stained for the pan macrophage marker F4/80 (red) and costained for either CD206 (green) or iNOS (green) to determine the phenotype of SIMs in CIA model mice treated with different formulations. Nuclei were stained with DAPI (blue). White arrows indicate double-positive cells (F4/80$^+$CD206$^+$ and F4/80$^+$iNOS$^+$). Scale bars, 50 μm. ($n$ = 5 biologically independent animals per group). **f, g** Histological analysis with H&E (scale bar 200 μm) (**f**) and safranin O staining (scale bar 200 μm) (**g**) of the joint tissues excised from the mice after different treatments. ($n$ = 5 biologically independent animals per group). H, synovium hyperplasia; I, immune cell infiltration; P, pannus formation; C, cartilage damage. **h–k**, The percentage of MPO-positive cells in **d** (**h**) and quantification of F4/80$^+$CD206$^+$ staining and F4/80$^+$iNOS$^+$ staining of macrophages in **e** (**i**). The histopathological evaluation of synovial inflammation (**j**) and cartilage depletion (**k**) were obtained from the images in **f** and **g**, respectively. Data are presented as the mean ± s.d. ($n$ = 5 biologically independent animals per group). (exact $P$ values: MPO$^+$: $P$ = 0.0051, $P$ = 1.57371E-05; CD206$^+$: $P$ = 0.3267, $P$ = 0.0028; iNOS$^+$: $P$ = 0.2901, $P$ = 0.0006; Synovial inflammation: $P$ = 0.0193, $P$ = 6.11004E-05; Cartilage depletion: $P$ = 0.3016, $P$ = 0.0018); *$P$ < 0.05, **$P$ < 0.01, ***$P$ < 0.001, ****$P$ < 0.0001. NS, not significant. **l** Representative micro-CT images used for quantification of bone erosion in mice. **m** Quantification of bone mineral density determined by micro-CT. Data are presented as the mean ± s.d. ($n$ = 5 biologically independent animals per group). (exact $P$ values: $P$ = 0.7094, $P$ = 0.8357, $P$ = 0.0103; *$P$ < 0.05. NS, not significant. **n** Concentrations of TNF and IL-1β in the serum from CIA mice receiving the indicated treatment. Data are presented as the mean ± s.d. ($n$ = 5 biologically independent animals per group). (exact $P$ values: TNF: $P$ = 0.0466, $P$ = 7.16875E-06; IL-1β: $P$ = 0.2901, $P$ = 7.47296E-06); *$P$ < 0.05, ****$P$ < 0.0001. Two-way ANOVA with Turkey's post hoc test (**b, c**) and one-way ANOVA with Turkey's post hoc test (**h–k, m, n**) was conducted. Source data are provided as a Source Data file.

joint synovium, including the secretion of neutrophil attractants-chemokine CXCL1[14]. *IRF5* deficiency supported a reduction in macrophage-derived CXCL1 levels and limited neutrophil infiltration into the inflamed joint. Our results demonstrated that targeted silencing of *IRF5* with our EINI could repolarize SIMs to an anti-inflammatory phenotype (Figs. 3g and 5e) and terminate SIM-initiated pathological cascades, thus inhibiting the unchecked recruitment of neutrophils (Figs. 3e and 5d) and the initiation of osteoclast-dependent cartilage destruction and bone erosion (Fig. 5g, l and Supplementary Fig. 27b). In addition, metformin interferes with key immunopathological mechanisms involved in systemic autoimmune diseases, such as ROS production, macrophage polarization and cytokine synthesis[25,41]. The metformin in the nanoimitator can prevent complex I-derived ROS and thus promote SIM-targeted reprogramming. Our data indicate that the metformin dissociated from the EINI significantly decreased the production of ROS in SIMs (Supplementary Fig. 16), which further blocked ROS-mediated inflammatory signals (Fig. 3f) and was conducive to reestablishing intraarticular immune homeostasis synergistically.

Patients with RA can be divided into two major subsets based on the presence versus absence of anti-citrullinated protein antibodies (ACPAs)[10]. The ACPA$^+$ subset of the disease accounts for approximately two-thirds of all cases of RA and generally has a more severe disease course[11]. ACPAs in the inflamed synovium have been shown to associate with citrullinated antigens to form ACPA-immune complexes (ACPA-ICs), resulting in progression of the inflammatory process[42]. ACPA-ICs subsequently drive a strong proinflammatory cytokine response in macrophage colony-stimulating factor differentiated macrophages[43]. Furthermore, ACPAs might contribute to osteoclastogenesis and the development of joint pathology by facilitating the proinflammation polarization of macrophages[44,45]. Here, we showed that the expression of IRF5 was positively correlated with the ACPA titer in the RA synovium (Supplementary Fig. 1a). Consistently, in patients with ACPA$^+$ RA, the expression level of IRF5 protein was obviously elevated (Supplementary Fig. 1b). Targeted silencing of IRF5 in SIMs could facilitate the anti-inflammatory polarization of macrophages (Fig. 5e, i and Supplementary Fig. 26) and thus abort SIM-initiated cascades in ACPA$^+$ RA (Fig. 5g, l). Our data provide evidence for the reestablishment of articular immune homeostasis during ACPA$^+$ RA immunopathogenesis through reprogramming of macrophages to an immunoregulatory phenotype. In contrast to the well-characterized pathogenic mechanisms of ACPA$^+$ RA, the etiology of ACPA$^-$ RA remains largely unknown. Although they exhibit similar clinical arthritic symptoms, the immune pathogeneses of ACPA$^-$ and

ACPA$^+$ RA patients are quite different[46] and they might require tailored treatment strategies. In subsequent work, we will systematically study whether the long-term outcomes differed for these two subsets of RA patients, in the hope that this will lead to stratified treatment in RA.

In sum, our findings establish that synovial inflammatory macrophages (SIMs) can be targeted and reprogrammed for RA reversible treatment with the proposed efferocytosis-informed nanoimitator (EINI) (Fig. 6). By manipulating the shielding and exposure of PtdSer in the nanoformulation, we demonstrated that the SIMs in RA joints could be precisely targeted in situ. Functionally, EINI effectively repolarized SIMs to an immunoregulatory phenotype in situ, serially terminating SIM-initiated pathological cascades and ultimately reestablishing articular immune homeostasis. Our work, therefore, provides a regulatory strategy for macrophage heterogeneity for RA reversible treatment, which may be extended to various macrophage-involving autoimmune diseases, such as atherosclerosis, idiopathic pulmonary fibrosis, and inflammatory bowel disease. Although we concentrated on IRF5 in this study, it is logical to extend this approach to other diseases harboring common transcription factors alterations, such as hypoxia-inducible factor (HIF)-1α or runt-related transcription factor 1 (RUNX1).

## Methods
### Animals
Male DBA/1 J mice (6 weeks old) and C57BL/6 mice (7 weeks old) were purchased from the Beijing Vital River Laboratory Animal Technology Co., Ltd. Mice were housed under conditions of a light/dark cycle of 12 h, an ambient temperature of 25 ± 2 °C, and a humidity of 60 ± 10%. All animal studies were performed following the protocols approved by Shandong University's Institutional Animal Care and Use Committee and in compliance with all relevant ethical regulations.

### Materials
Dioleoylphosphatydic acid (DOPA) and dioleoyl phosphatidylserine (PtdSer) were purchased from A.V.T. (Shanghai) Pharmaceutical Co., Ltd. (Shanghai, China). Metformin, cyclohexane, Igepal CO-520 and zinc nitrate hexahydrate (Zn(NO$_3$)$_2$·6(H$_2$O)) were obtained from SigmaAldrich (St. Louis, MO, USA). Low-molecular-weight-heparin (LMWH, MW 3800-5000, Mw/Mn (PDI) = 1.34) was purchased from Melonepharma (Dalian, China). Recombinant murine macrophage colony-stimulating factor (M-CSF) (315-02) and recombinant human TNF (300-01 A) were purchased from PeproTech (Rocky Hill, USA). RPMI 1640 medium, Dulbecco's modified Eagle's medium (DMEM), modified Eagle's medium, trypsin EDTA, type I collagenase, fetal

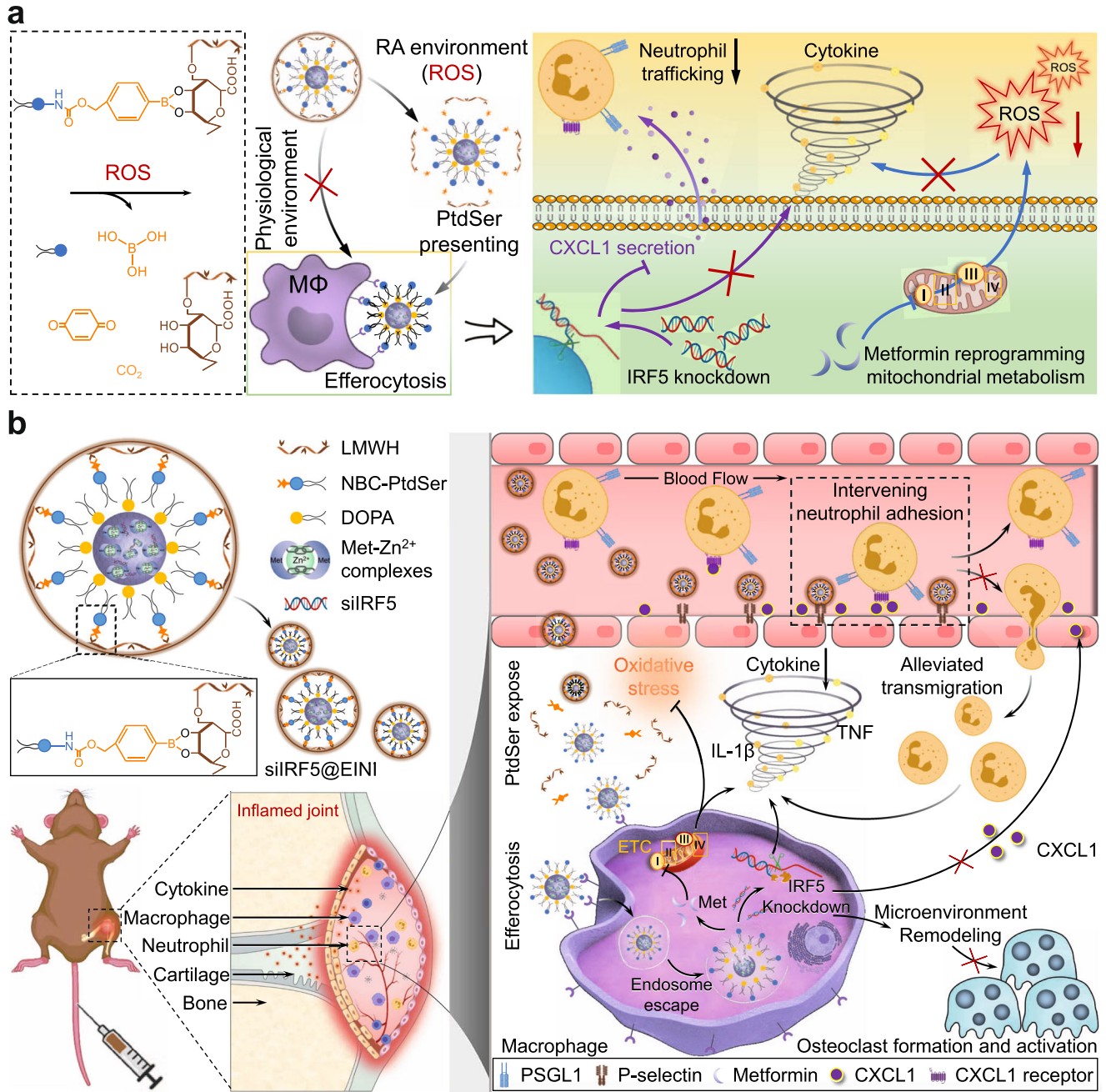

**Fig. 6 | Overview of the proposed mechanism of siIRF5@EINI-induced inflammatory regulation in RA. a** To manipulate the locoregional exposure of the phosphatidylserine (PtdSer) corona of the designed nanoimitator intra-articularly, a benzeneboronic acid pinacol ester group was used to make the nanoimitator responsive to local ROS stimuli. The ROS-abundant inflamed joint microenvironment triggers PtdSer presentation of the nanoimitator. In an efferocytosis-like manner, the PtdSer-coroneted core is in turn phagocytosed by synovial inflammatory macrophages, which synergistically terminate the SIM-initiated pathological cascades that pro-inflammatory cytokine production, oxidative stress, and recruitment of neutrophils. **b** Schematic illustration showing that the efferocytosis-informed nanoimitator terminates SIM-initiated pathological cascades, synergistically restores articular immune homeostasis, and ultimately reverses bone erosion. The siIRF5-carrying efferocytosis-informed nanoimitator (siIRF5@EINI) consisted of a drug-based core with an oxidative stress-responsive PtdSer corona and a shell composed of a P-selectin-blocking motif, low molecular weight heparin (LMWH). With the shielding of LMWH, siIRF5@EINI is endowed with stealth properties in the circulation, enhanced retention in inflamed regions, and a blocking function of P-selectin that retards the articular trafficking of neutrophils. Upregulated ROS triggered shell exfoliation and the PtdSer corona is then exposed. In an efferocytosis-like manner, the PtdSer-coroneted core is in turn phagocytosed by SIMs, which synergistically terminate SIM-initiated pathological cascades and serially reestablish intra-articular immune homeostasis, conferring a chondroprotective effect.

bovine serum (FBS), and PBS were obtained from Gibco (USA). DiR (D12731) and PKH67 were purchased from Invitrogen (USA). The anti-CD206 antibody, anti-iNOS antibody, anti-Ly-6G antibody, and anti-myeloperoxidase (MPO) antibody were purchased from BioLegend. All siRNAs used in the CIA model were synthesized by Genepharma (Shanghai, China), and the sequences used were as follows: (i) *IRF5*: 5′-dTdT-CUG CAG AGA AUA ACC CUG A-dTdT-3′ (sense) and 5′-dTdT UCA GGG UUA UUC UCU GCA G dTdT-3′ (antisense). A dye was introduced at the 5′-end of the antisense strand of siIRF5. (ii) Negative control scrambled siRNA (siN.C).

## Human synovium specimens

Human synovial tissue sections were obtained from ACPA$^+$ RA patients (1 male and 2 females; average age: 66.8 years) and healthy donors (2 males and 1 female; average age: 56.3 years) from the Department of Orthopaedic Surgery at Shandong Provincial Hospital Affiliated to Shandong First Medical University. All samples were obtained with written informed consent and collected using a standard protocol approved under the Review Board protocol of Shandong First Medical University (Approval no. 2019-272).

## Cell culture

Bone marrow-derived macrophages (BMDMs) were differentiated from bone marrow monocytes. The femurs were dissected from the C57BL/6 mice (7–8 weeks old, 22–24 g). Then, the bone marrow was rinsed with 1× PBS to isolate bone marrow cells, which was followed by erythrocyte lysis in ACK lysis buffer. After centrifugation, the cells were resuspended in a medium with M-CSF (20 ng/ml) to induce the maturation of macrophages. After 72 h of incubation, the macrophage differentiation medium was discarded, and the plates were washed with sterile PBS to remove nonadherent cells. The adherent BMDMs were harvested and used in subsequent experiments for at least 7 days after induction. The BMDMs were collected after detachment with 0.05% trypsin EDTA and centrifugation at $400 \times g$ for 5 min, and then the BMDMs were stained with PerCP-Cy5.5-conjugated anti-CD11b and FITC-conjugated anti-F4/80 antibodies. The induction of mature macrophages was evaluated by flow cytometry (Beckman Coulter, USA).

Mouse neutrophils were collected from the whole blood of C57BL/6 mice (7–8 weeks old, 22–24 g) using the Percoll gradient method[47]. Briefly, before the isolation of neutrophils, 1 mg/kg LPS was injected intraperitoneally into the mice to activate neutrophils in vivo. After 12 h, the second injection was conducted. Three hours after the second injection, blood was collected by submandibular bleeding. Pooled blood was centrifuged ($3220 \times g$, 5 min, 4 °C), and the buffy coat on top was added to a Percoll mixture solution consisting of 78%, 69%, and 52% (v:v) Percoll in PBS, followed by centrifugation at $1500 \times g$ for 30 min. The cells from the interface of the 69% and 78% gradient layers and the 78% fractions were recovered to a new tube. ACK lysis buffer was then added to the sample to lyse the erythrocytes. Neutrophils were purified by washing with ice-cold 1× PBS three times and resuspending the cells in a culture medium (Supplementary Fig. 12).

Mouse FLSs were isolated from the knee joint synovium of CIA mice[48]. Briefly, mouse synovial tissues were isolated from knee joints, and synoviocyte suspensions were minced and digested in collagenase for 4 h at 37 °C. Cell suspensions were passed through a 70 μm cell strainer and placed in tissue culture dishes containing DMEM supplemented with 10% FBS. FLSs between passages 4 and 8 were used for further analysis. Human FLSs were isolated from primary synovial tissue obtained from 3 patients with RA who met the American College of Rheumatology (formerly the American Rheumatism Association) revised criteria and had undergone total joint replacement surgery or synovectomy, as previously described. Informed consent was obtained from all patients, and the study protocol was approved by the Shandong First Medical University Ethics Committee. Pure FLSs were identified by flow cytometry using antibodies against the fibroblast marker CD90 and the macrophage marker CD14.

HUVECs were obtained from the cell bank of the Chinese Academy of Sciences (Shanghai, China) and maintained in DMEM (Gibco, USA) supplemented with 10% FBS and 1% penicillin/streptomycin. All cells were incubated at 37 °C in a humidified atmosphere with 5% CO$_2$.

## Synthesis of NBC-PtdSer

The NBC-PtdSer was synthesized according to the procedure in Supplementary Figs. 2a and 3a. First, synthesis of NBC, 4-(hydroxymethyl)

phenylboronic acid pinacol ester (0.25 g, 1.1 mmol) was dissolved in 10 ml of dry THF. Triethylamine (0.3 ml, 2.15 mmol) was added followed by 4-nitrophenyl chloroformate (0.27 g, 1.3 mmol), and the reaction was stirred at room temperature for 12 h. The reaction was diluted with ethyl acetate and washed with 1.0 M HCl followed by saturated NaHCO$_3$. The organic layer was dried over MgSO$_4$, filtered, and concentrated. NBC was purified on a silica gel column eluted with 5% EtOAc in hexanes to yield 0.26 g (0.65 mmol, 61% yield) as a white solid. The $^1$H NMR spectrum of the product was obtained on a Bruker Advance DRX-400 spectrometer (Germany) with tetramethylsilane (TMS) as an internal standard.

Then, for synthesis of NBC-PtdSer, PtdSer (100 mg, 0.1344 mmol) and NBC (64.4 mg, 0.1613 mmol) were dissolved in dry DMF (3 ml) under an N atmosphere. After adding triethylamine (56 μl, 0.4032 mmol), the reaction was stirred for 12 h before being quenched by pouring into 100 ml of 0.5 M HCl. The aqueous layer was extracted with $3 \times 25$ ml chloroform. The combined organic layer was washed with $5 \times 100$ ml water and once with brine. The collected organic phase was dried over Na$_2$SO$_4$, filtered, and concentrated under reduced pressure. The resulting crude product was subjected to silica gel column chromatography. Gradient elution from 100% chloroform to 20% MeOH in chloroform was needed to purify the product as a white solid.

## Preparation and characterization of the nanoformulation

siIRF5@EINI were prepared using a modified water-in-oil reverse microemulsion protocol. Briefly, 500 μl of 30 mM Zn(NO$_3$)$_2$·6(H$_2$O) and 100 μl siIRF5 (siRNA at an input concentration of 20 μM) were dispersed in 20 ml cyclohexane/Igepal CO-520 (71/29 V/V) solution to form a very well dispersed water-in-oil reverse microemulsion. To prepare the metformin loaded microemulsion, metformin solution (100 μl, 30 mM) and DOPA (100 μl, 35 mM) were added into a separate 20 ml oil phase. After mixing the above two microemulsions for 30 min to form the condensed cores, 40 ml of ethanol was added and the mixture was centrifuged at $12,500 \times g$ for 20 min to remove cyclohexane and the surfactant. To prepare the PtdSer-conjugated core, the pure pellets were suspended in 750 μl chloroform and mixed with 4 mg NBC-PtdSer. Then, the chloroform was evaporated and 4 ml of 1× PBS was added to form PtdSer-conjugated core. Additionally, the corona of nanocore was cloaked with LMWH via a reversible boronate ester linker by the reaction of cis-diols and phenylboronic acid groups, and an efferocytosis-informed nanoimitator was achieved.

The morphology and size of the nanoimitator were observed under a Hitachi H-7650 TEM (Hitachi, Japan) after negative staining with uranyl acetate solution (2%, wt/wt). The particle size distribution and zeta potential were measured by dynamic light scattering (DLS) using a Malvern Zetasizer Nano ZS. PtdSer presentation was validated by flow cytometry. Briefly, the nanoimitator was resuspended in PBS at pH 7.4 and incubated with or without H$_2$O$_2$ (0.1 mM). Then, these nanoformulations were incubated with FITC-Annexin V in the presence of Ca$^{2+}$ for 15 min, and the binding rate of FITC-Annexin V was detected by flow cytometry. To quantify metformin release, the nanoimitator was resuspended in PBS at pH 5.0 or pH 7.4. At predetermined time points, aliquots from each group were centrifuged to pellet the nanoparticles, and the concentration of released metformin in the supernatant was measured using an HPLC system (Agilent 1100, USA) under the following chromatographic conditions: column, C18 column ($5 \times 250$ mm, particle size 5 μm); mobile phase, methyl alcohol-15 mM KH$_2$PO$_4$ (60:40, v/v); flow rate, 1.0 ml/min; detection wavelength, 233 nm.

## Gel retardation assay

Free siIRF5 and siIRF5@EINI (in 1× PBS) were separately mixed with loading buffer and loaded into a 3% wt agarose gel with NA-Green (Beyotime Biotechnology Co., Ltd, China). Electrophoresis was conducted in 1× tris-acetate EDTA (TAE) buffer at 80 V for 30 min. To

investigate pH responsiveness, siIRF5@EINI was resuspended in PBS at pH 5.0 or pH 7.4 for 1 h. After that, the samples were analyzed by gel electrophoresis. The resulting gels were analyzed using a UV illuminator (IS-2200; Alpha Innotech, San Leandro, CA, USA) to show the location of siIRF5.

## Macrophage targeting abilities of the EINI in vitro

BMDMs or FLSs were seeded in 6-well tissue culture plates at 50% confluency and cultured overnight. The cell culture medium was changed, and 10 ng/ml recombinant human TNF was added for stimulation for 4 h. The nanoimitator was then added to the cells with or without $H_2O_2$ (0.1 mM) pretreatment. After incubation, the cells were washed three times with ice-cold PBS and fixed with 4% paraformaldehyde. Then the cells were incubated at 37 °C for 10 min with 10 nM DAPI and imaged with an EVOS inverted fluorescence microscope (Thermo Fisher Scientific). For flow cytometric analysis, cells were scraped and collected after PBS washing and then analyzed with a Beckman Coulter Gallios flow cytometer. The results were analyzed using FlowJo software. To verify the specific macrophage homing of the nanoformulation, PKH67-labeled FLSs were cocultured with unlabeled macrophages at a ratio of 1:1. Cells were stimulated with 10 ng/ml TNF for 4 h at 37 °C. $H_2O_2$-pretreated nanoimitator was added to the medium for 2 h. The uptake of the nanoimitator was observed by CLSM.

## siRNA tracking and release in the cell

To investigate the endocytic pathways and intracellular release behavior of siIRF5, macrophages were treated with the siIRF5@EINI that was pre-conditioned with the 0.1 mM $H_2O_2$. At predetermined times, the treated cells were rinsed with PBS and incubated at 37 °C for 30 min with 100 nM LysoTracker Green in a cell culture medium. After incubation, the cells were washed with ice-cold PBS, fixed with 4% PFA for 30 min at 4 °C, and subjected to DAPI staining. Fluorescence imaging was performed by CLSM.

## Evaluation of the effects of the nanoimitator on the adhesion and transmigration of neutrophils

To assay the blocking effect on cell adhesion, $2 \times 10^5$ HUVECs were seeded in 6-well plates and pre-stimulated by incubation for 6 h with TNF (10 ng/ml). Then, the cells were treated with RPMI medium, P-selectin antagonist (Human P-selectin/CD62P antibody, R&D Systems), and the nanoimitator for 30 min. A total of $1 \times 10^6$ activated neutrophils were seeded on top of the treated endothelial monolayer. After 30 min at 37 °C under gentle shaking, each well was washed twice with PBS to remove the non-adhered cells and labeled with FITC-labeled anti-Ly-6G antibody. Finally, neutrophils adhered to HUVEC monolayers were evaluated using CLSM and flow cytometry.

To quantitative analysis of neutrophils adhered to HUVEC monolayers, HUVECs ($2 \times 10^5$) were seeded in 6-well plates and pre-stimulated by incubation for 6 h with TNF (10 ng/ml). Then the cells were treated with RPMI medium, P-selectin antagonist (Human P-selectin/CD62P antibody, R&D Systems), and the nanoimitator for 30 min. Activated neutrophils ($1 \times 10^6$) were labeled with 5 μM calcein-AM and seeded on top of the treated endothelial monolayer. After 30 min at 37 °C under gentle shaking, each well was washed twice with PBS to remove the non-adhered cells, and the cells in each well were lysed with 2 mL dimethyl sulfoxide (DMSO), and their fluorescence intensities were detected by a fluorospectro-photometer at Ex = 495 nm and Em = 515 nm.

The inhibitory effects of the nanoimitator on the migration of neutrophils were evaluated by Transwell assays. Prior to the experiment, the isolated neutrophils were pre-stimulated with 10 μM fMLF for 15 min at 37 °C. The activated macrophages were treated with different formulations (the free siIRF5, 0.1 mM $H_2O_2$-pretreated siN.C@EINI, and siIRF5@EINI) for 24 h. Cell medium was collected and the

bottom of Transwell chambers was filled with cell medium from each formulation. Then, $1 \times 10^6$ preactivated neutrophils were seeded on a 3 μm polycarbonate membrane. After incubation for 1 h, the neutrophils that migrated to the lower chamber were stained with crystal violet and counted. The results were expressed as a percentage of the total number of neutrophils added.

## Gene silencing assay

The IRF5 gene silencing efficiency of the nanoimitator was investigated by quantitative real-time PCR (qRT-PCR). Macrophages were seeded in a six-well plate ($1 \times 10^6$ cells per well) in growth medium for 24 h. Cells were treated with TNF (10 ng/ml) for 6 h. The cells were incubated with the desired nanoformulations in 2 ml of fresh medium. After 48 h, the cells were washed with PBS, and total RNA was extracted using the RNeasy Mini Kit (Qiagen) according to the manufacturer's instructions. Real-time quantitative PCR was performed to measure the mRNA levels of IRF5 in macrophages. β-actin was chosen as an internal housekeeping gene to normalize IRF5 mRNA. The mRNA expression level was calculated based on the comparative Ct method ($2^{-\triangle\triangle Ct}$). IRF5 blockade would have a major impact on reducing the secretion level of macrophage-derived CXCL1; hence, CXCL1 mRNA expression was then evaluated by qRT-PCR. Real-time PCR was performed using specific primers (Supplementary Table 1).

## Phenotypic switching of macrophages

For fluorescence immunostaining, macrophages were stimulated with TNF (10 ng/ml) to induce inflammation. Then, $H_2O_2$-pretreated nanoformulations were added to the media, and PBS and free siRNA were prepared in parallel for comparison. After 4 h, the cells were incubated for another 24 h. Cells were fixed with 4% PFA for 15 min and permeabilized with 0.1% Triton X 100 (room temperature, 10 min). The cells were blocked with normal rat serum (37 °C, 1 h) and incubated with anti-iNOS or anti-CD206 antibody at 4 °C overnight, followed by counterstaining with DAPI for 10 min. Fluorescence imaging was performed with confocal microscopy. Quantitative analysis of the relative fluorescence intensity of CD206$^+$ staining versus iNOS$^+$ staining in each group ($n = 3$). At least three images per group were acquired for quantification, and positively stained areas were evaluated with ImageJ software. (http://imagej.net/). qRT-PCR analysis was performed with the same procedure described above. The primer sequences are shown in Supplementary Table 1.

## CIA model and treatment protocols

An autoimmune CIA mouse model was established by the following protocol. Immunization type II collagen (CII) lyophilized powder was dissolved in acetic acid (0.1 mol/l) and blended with an equal volume of an emulsion of complete Freund's adjuvant (CFA). A homogenizer was employed to adequately emulsify the mixture in an ice bath. Next, six-week-old male DBA1/J mice were injected intradermally at the tail root with 200 μg of bovine type II collagen (2 mg/ml) emulsified in 100 μl of complete Freund's adjuvant (4 mg/ml). On day 21, the mice received a booster immunization of type II collagen emulsified in incomplete Freund's adjuvant at the tail root.

To study therapeutic efficacy in CIA mice, the treatments began on day 7 after the second immunization. On day 28, the mice presented arthritis symptoms and were randomly divided into four groups ($n = 5$ mice in each group). Four different treatments (200 μL PBS, free siIRF5 solution, siIN.C@EINI, and siIRF5@EINI) were administered via the tail vein twice a week from day 28 to 60. The dose for each formulation was 1 nmol of siRNA dose per mouse, 10 mg/kg metformin-equivalent dose. The swelling of joint of the mice was evaluated by measuring the volume of the hindpaws with plethysmometer and calculating the average volume. Clinical scores (score -0–4) were given by a blinded researcher based on the following criteria: 0,

normal; 1, mild redness of ankle or tarsal joints; 2, mild redness and swelling extending from ankle to the tarsals; 3, moderate redness and swelling from ankle to metatarsal joints; 4, severe redness and swelling encompassing the ankle, foot, and digits. Scores from 0 to 4 were assigned for each paw and then added together to yield a final disease score[49].

## Study of the biodistribution of the EINI in CIA model

At 7 days after boost immunization, the DiR-labeled nanoimitator was intravenously injected into CIA mice via tail vein. At predetermined times, real-time imaging of nanoparticles in the arthritis area was performed with an IVIS system (CRI, Inc., excitation: 748 nm, optical filter: 780 nm). At 96 h post-injection, these mice were sacrificed to harvest the major organs (heart, liver, spleen, lung, and kidneys) and inflamed paws, quantitatively determining the clearance ratios of the nanoformulations in vivo.

Specifically, fluorescence molecular tomography-computed tomography (FMT-CT) was performed on day 2 post-injection to identify nanoimitator localization in the inflamed regions. To observe the specific homing of the nanoimitator to the inflamed synovium, 100 μl of Cy5.5-labeled nanoformulation was injected into CIA mice via the tail vein, with free Cy5.5-labeled siIRF5 as a control. After 72 h, the mice were euthanized, and the synovium was obtained and fixed with 4% PFA 48 h at 4 °C. After dehydration, the samples were embedded in Tissue-Tek OCT Compound (Sakura Finetek) for frozen sectioning. Then, the slides were stained with anti-F4/80 antibodies, followed by DAPI staining. Fluorescence imaging was performed by CLSM. The antibodies used are summarized in Supplementary Table 2.

## qRT-PCR analysis of IRF5 expression in collected synovial macrophages

Mouse synovial tissues were isolated from knee joints, digested with 1 mg/ml type I collagenase in HBSS, and incubated at 37 °C and 5% $CO_2$ in a humidified atmosphere for 30–45 min. Disaggregated tissue elements were passed through a 70 μm cell strainer. Synovial macrophages (CD45$^+$, CD11b$^+$, F4/80$^+$)[50] were sorted on a BECKMAN Moflo Astrios EQ (BECKMAN Coulter). Next, total RNA was isolated according to the manufacturer's instructions. Synthesis of cDNA and qRT-PCR was performed as previously described in vitro analysis.

## Histological analysis of inflamed joints

Histopathology was evaluated to determine the features of RA, including neutrophil migration, macrophage phenotype, angiogenesis, synovitis, and cartilage destruction. At the study endpoints, the mice were euthanized, and the hind ankle joints were collected for H&E staining and safranin-O or TRAP staining. To detect neutrophil infiltration, joint sections were dewaxed and stained with anti-MPO (Bioss Antibodies) primary antibodies. Biotinylated anti-rabbit IgG was used as the secondary antibody for chromogen development. To evaluate the phenotypic switching of macrophages, tissue sections were stained with anti-iNOS or anti-CD206 antibodies. Tissue sections were then stained with DAPI mounting solution, and observed using CLSM.

The hind paw joints were dissected and evaluated with micro-computed tomography (micro-CT) imaging. The micro-CT scanning parameters were voxel size, 9 μm; voltage, 90 kV; current, 88 μA; and exposure time, 4 min. To compare the therapeutic effects of the different groups, three-dimensional images were reconstructed using Mimics software (Materialize) and the mean bone mineral density (BMD) of the joints was calculated by CTAn software (Skyscan).

## Histological scoring of mouse arthritic joints

Histopathological scoring was performed. Briefly, joints of arthritic mice were assigned scores of 0 to 4 for inflammation based on H&E staining, according to the following criteria: 0 = normal; 1 = minimal infiltration of inflammatory cells in the periarticular area; 2 = mild

infiltration; 3 = moderate infiltration; and 4 = marked infiltration. Cartilage depletion was identified by diminished safranin O staining of the matrix and was scored on a scale of 0 to 4, where 0 = no cartilage destruction (complete staining with safranin O), 1 = localized cartilage erosions, 2 = more extended cartilage erosions, 3 = severe cartilage erosions, and 4 = depletion of entire cartilage. Histologic evaluations were performed in a blinded manner[51].

## Statistical analysis

The results are presented as mean ± s.d. Error bars represent the s.d. of the mean from independent samples. Two-tailed Student's $t$ test was applied to test the statistical significance of differences between two groups. Statistically significant differences among the groups were analyzed using one-way or two-way analysis of variance (ANOVA). GraphPad Prism 7.0 (GraphPad Software Inc.) and Microsoft Excel 2016 (Microsoft Inc.) were used for all statistical analysis. Statistical significance was set at *$P < 0.05$, **$P < 0.01$, ***$P < 0.001$, ****$P < 0.0001$.

## Reporting summary

Further information on research design is available in the Nature Portfolio Reporting Summary linked to this article.

## Data availability

All data generated from this study are available within the Article, Supplementary Information or from the corresponding authors upon request. IRF5 mRNA expression levels in patients with RA retrieved from the ArrayExpress (https://www.ebi.ac.uk/arrayexpress/) under Accession code E-MTAB-6141[52]. Source data are provided with this paper.

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

## Acknowledgements

This work was supported by the National Natural Science Foundation of China (82173763, 91842305, and 81771686 to X.J.), the ISF-NSFC Joint Scientific Research Program (52161145501 to X.J.), Funds for Youth Interdisciplinary and Innovation Research Groups of Shandong University (2020QNQT003 to X.J.), the Fundamental Research Funds of Shandong Province (ZR202206110012 to X.J.), and the Shandong Provincial Key Research and Development Program (Major Scientific and Technological Innovation Project) (2021CXGC010515, 2019JZZY011127

and 2019JZZY021013 to X.J.), and Shandong Provincial Natural Science Foundation (ZR2020MH260 to X.J.). Thanks for the technical support from Y. Yu, X.-M. Yu, J. Zhang, M.-L. Wu, and L.-M. Wang in Advanced Medical Research Institute/Translational Medicine Core Facility of Advanced Medical Research Institute, Shandong University. Figure 1a, the cartoon mouse in Figs. 5a and 6 were created with BioRender.com.

## Author contributions

X.J. and S.Z. conceived the study and designed experiments. S.Z. and Q.C. characterized the properties of the materials. S.Z., Y.L., W.J., C.T., and Z.L. performed the animal experiments. Z.M., C.C., J.Z., Y.L. P.S., R.Z., Z.Y., Y.W., and M.H. contributed to data analysis and interpretation. S.Z. crafted all the figures and wrote the manuscript. X.J., M.A., W.L., X.W., G.Y., B.S., J.L., K.Z. and Y.Z. edited and revised the manuscript and supervised the research.

## Competing interests

The authors declare no competing interests.

## Additional information

[1]NMPA Key Laboratory for Technology Research and Evaluation of Drug Products and Key Laboratory of Chemical Biology (Ministry of Education), Department of Pharmaceutics, School of Pharmaceutical Sciences, Cheeloo College of Medicine, Shandong University, 44 Cultural West Road, Jinan, Shandong Province 250012, China. [2]Department of Urology, Qilu Hospital, Cheeloo College of Medicine, Shandong University, Jinan, Shandong Province 250012, China. [3]Department of orthopaedic surgery, Shandong Provincial Hospital Affiliated to Shandong University, Jinan, Shandong Province 250021, China. [4]Department of orthopaedic surgery, Shandong Provincial Hospital Affiliated to Shandong First Medical University, Jinan, Shandong Province 250021, China. [5]Shandong University of Traditional Chinese Medicine, Jinan, Shandong Province 250355, China. [6]Department of Pharmacology and Toxicology, Shandong Institute for Food and Drug Control, Jinan, Shandong Province, China. [7]Neck-Shoulder and Lumbocrural Pain Hospital, Shandong First Medical University & Shandong Academy of Medical Sciences, Jinan, China. [8]Department of orthopaedic surgery, Qilu Hospital, Cheeloo College of Medicine, Shandong University, Jinan, Shandong Province 250012, China. [9]These authors contributed equally: Shengchang Zhang, Ying Liu, Weiqiang Jing. ✉e-mail: drzhangyk@163.com; xinyijiang@sdu.edu.cn

