## [Peer Review File · Nature Communications]

Remodeling Articular Immune Homeostasis with an Efferocytosis-inspired Nanoimitator Mitigates Rheumatoid ArthritisREVIEWER COMMENTS

Reviewer #1 macrophage reprogramming (Remarks to the Author):

The manuscript by Zhang and colleagues reports a novel compound designed to target phagocytic cells under inflammatory conditions. The authors provide proof of concept data for this compound and show data suggesting it can counteract adverse tissue remodelling in an in vivo mouse model of rheumatoid arthritis. Such a compound would be of great value clinically, and may in principle also be useful in other inflammatory pathologies in which macrophages play a key role. However, in its current form, the manuscript presents with several severe shortcomings, which would have to be addressed to elevate it to a level acceptable for publication at Nature Communications, and ascertain the usefulness of the compound. Most importantly, there appears to be an overall lack of rigor.

Please find below a detailed list of issues and questions to be addressed.

Major

1. Targeting selectivity, specificity, endosomal escape:

- The authors are making a strong point about their compound selectively targeting synovial joints, and more specifically, macrophages within the joints. This claim is not fully supported by the data shown, and additional data will be necessary to substantiate this claim. The authors postulate that their compound is targeted to endothelium through its "shell" made of low molecular weight heparin, which recognizes P-selectin on endothelial cells. The "corona" of the compound is designed to be responsive to oxidative stress, making it supposedly more active in an inflammatory environment. In vivo imaging of mice that have received the compound carrying a fluorescent cargo (Figure 5) suggests that the compound is indeed enriched in joints, in addition to the liver and presumably kidney (annotated as K in the figure = explanation of these abbreviations is missing from the figure legends, see next point about general lack of rigor). There also appears to be signal at the base of the tail, which is not commented on, but would be in line with inflammation owing to adjuvant injection in the CIA model. To interpret this, the authors need to show the whole mouse (not images cut off near the base of the tail), and as also highlighted below, specify when in relation to the CIA model the compound was given and include longitudinal clinical data.
- Distribution of the compound should also be assessed in non-inflamed ie treatment-naïve mice.
- We are led to believe that this is due to the inflammatory state of the joints, along with an impaired microvasculature. However, it is entirely unclear at which stage of the CIA arthritis model the compound was given, and there is no data showing the extent of inflammation. These data have to be included to draw any further conclusions. Furthermore, according to supplementary figure 11c, inflammation is also observed in the lung. This is different from the representative image in the main figure, and requires clarification. Finally, based on the information given, there is no reason to assume that targeting is arthritis-specific, rather, this should be true for inflamed tissues more generally. In principle, it would thus be desirable that the authors showed their compound is targeted to sites of inflammation in other conditions. In fact, this could underscore the usefulness of this approach for other diseases, and would thus be of interest for a wider audience. In the current form, it is impossible to assess if the effects of the drug are in preventing adverse joint remodelling, or rather restoring these, as timeline of treatment relative to arthritis induction is uncertain.
- The statement that "the compound would escape the lyso/endosome to release it in the cytoplasm" is quite strong and lack of evidence. The author should at least show a plot profile demonstrating that there is no colocalization and higher magnifications or use softwares (Imaris?) to show that it is in the cytoplasm and not the endosomes as the images are not clear enough to conclude (figure 3e).
- Another major concern relates to the specificity of targeting *Irf5*. The scrambled siRNA control has pronounced effects already (Figure 6). This appears to not be mentioned or discussed. Equally, the relevant statistics for the comparison between this and other negative controls are not shown. This point is a major shortcoming, putting in question the selectivity of their targeting approach.
- Similar to the specificity, the data supporting endosomal escape of the compound are not fully convincing in their current form.

2. Mechanism of action: The authors don't confirm in vivo what they see in vitro with the huvecs. Is the compound reaching endothelial cells and blocking the transmigration of neutrophils? Are ROS present in their model (in vivo)?

3. Arthritis model: Further to the comments above, the authors need to provide longitudinal clinical data on any mice they claim are arthritic. In the current form, the only data presented on disease symptoms is histology (Figure 6), presumably taken at disease endpoint, though this too needs to be clarified. Throughout the manuscript, the authors need to include data for clinical parameters, i.e. clinical scoring of joint inflammation as well as measurements for paw swelling and arthritis incidence within treated cohort.

4. Rigor, other:

- Several legends to supplementary figures are missing.

- In most experiments, a low number of replicates was used. Additional clarification is needed if data points represent biological replicates or technical replicates.

 - o For example, in Figure 6, is n=5 per group or a total n=5? If the latter, this will need repeating.

 - o Figure 6b: There are 10 datapoints, but the n is supposedly 5 mice. This suggests these are individual ankles corresponding to 5 mice? This is misleading. A mean of both ankles/mouse might be better.

- Figure 6: Histological images seem to be taken in different joint regions, which makes it hard to compare. The authors should try to quantify around the same areas and do so on several slides. Then show adjacent slides e.g. in f, g and h.

- Characterisation of "inflammatory" macrophages: CD206 and iNOS are limited markers. iNOS+ staining seems very strong compared to CD206+ cells. Counterstaining with a pan macrophage marker is required. Phenotypic characterisation of macrophages should be confirmed by flow cytometry. Tim4 is a phosphatidyl receptor and is associated with tissue residency and homeostasis in other tissues. Are the macrophages that take up the compound Tim4+ or can all macrophages take it up to the same degree?

Minor

1. Rigor, additional clarification needed:

- Figure 2d: Axes need to be annotated.

- The authors used HUVECS co-cultured with FLS and BMDM. How many wells were used per experiments? Is it 1 well per experiment? Are HUVECS derived from different batches/donors?

- Figure 3c: The authors state that the compound is specifically engulfed by macrophages in the presence of ROS existed, however, FLS show an MFI of 10.000. Is this the background level? The authors should show data and include raw flow cytometry plots comparing the fluorescence of both cell types with and without the compound to conclude on specificity.

- Figure 3b, e: Add quantification in addition to representative images.

- Figure 3e: The statement that "the compound would escape the lyso/endosome to release it in the cytoplasm" is strong and currently lacking enough supporting evidence. The authors should at least show data demonstrating that there is no colocalization and ideally use higher magnification views. Imaging analysis software should be used to quantify cytoplasmic versus endosomal localization, since the images shown currently are not clear enough to conclude.

- Throughout: Where was mouse synovium isolated from, which joints?

- The data for gait analysis using the cat walk system are virtually impossible to interpret without additional information: What is shown, what does it mean? Quantification is needed for all experimental animals analysed.

- Figure 4h: It is unclear what the ratio shown refers to and how it was determined. Please clarify.

- Figure 6: Use consistent annotation of experimental groups throughout figure (and rest of the manuscript) – i.e. do not use G1, 2, 3, 4 in i-m, as this misleading.

- Rat tissues are mentioned in the supplementary data (Figure S15). Please clarify. Manuscript otherwise states mice as the only animal model used.

- Summary figures or cartoons outlining proposed mechanism underlying the effects of their compound should be moved to the end of the manuscript and clearly highlighted as "proposed mechanism".

- Line 64: The pathological cascades initiated by SIM should be specified (ROS production, cytokine production). This is relevant for the entire scope of the manuscript.

2. Language, general: While it is appreciated that the authors may not be native English speakers, the manuscript requires editing for syntax, grammar, spelling and overall clarity throughout. Non-exhaustive examples are:

- line 102: efferocytosis-mimetic = should be efferocytosis-mimicking
- line 256: over-zealous?
- line 423: please rephrase sentence

3. Terminology, acronyms: There appears to be an overuse of non-standard acronyms, e.g. NE instead of neutrophils. This makes reading of the article unnecessarily difficult. Similarly, some terms may not be commonly used or agreed upon, e.g. "payload" (instead of drug cargo?).

4. Figures, legends, annotations: Where histology or fluorescent imaging is shown, the annotation of what is stained for needs to be included in the main figure, not just the legend. This is common practice and enables the reader to easily identify what they are looking at.

5. Targeting approach: The authors should include a discussion or rationale for using this type of chemistry and compare it to other approaches used (in preclinical models) to target macrophages, such as liposomes (see e.g. PMID: 31375534). This will be [particularly useful for a non-expert audience, i.e. readers with expertise in macrophages and inflammatory disease, but not drug design.

6. Methods:

- Please specify how healthy donor synovial tissue was obtained.
- How did you check the purity of your sorting experiments or in vitro culture experiments?
- line 618: how were neutrophils dissociated from the HUVEC monolayer? In Supplementary Figure 8, they were shown as two independent populations.
- line 660: Please specify the site of injection for the immunisation boost
- line 721: use linear regression statistics to account for all covariates (e.g. sex)

7. BMDM: The authors' definition of bone marrow-derived macrophages is misleading. These are not primary macrophages isolated from mouse tissues, but rather in vitro generated from isolated progenitors.

8. Other:

- Figure 1a: has no added value. The mechanisms of action of the compound seem to not depend on the presence or absence of ACPA. Furthermore, the contribution of ACPA titers was not further investigated in the CIA model. The authors could include a section in the discussion, in which they address if and how their compound might behave differently in ACPA positive and negative RA.
- Line 67: states that immunosuppressive therapeutics often weaken the immune system with increased risk of infections. Did you test if compound-treated mice are more vulnerable to infections?
- Line 80: please stress this is a gain-of-function polymorphism
- Figure 1B: requires co-straining with a macrophage marker to link IRF5 expression specifically to macrophages.
- Figure 2: Please clarify how the compound concentration was assessed. In the follow-up experiments, no concentration was specified. Did you try different concentrations in your experimental setup? Please comment on that.
- Figure 2E: This graph appears to be inserted in the wrong place? There is an underlying image?
- Figure 3, line 192: figure legend is missing
- Figure 5, line 285: please specify the strain that was used
- Supplementary Figure 11: please stratify WBCs into lymphocytes, neutrophils, and monocytes. It seems that in the control group, there is a non-significant increase in the number of WBCs. How do you explain this?
- Supplementary Figure 15: Angiogenesis effects are unclear. Please add additional markers like CD31

9. Discussion: The authors should cover the following points:

- Potential use of the compound for other disease contexts, potential of the same drug design targeting other genes, including other disease-relevant transcription factors
- line 460 states that metformin in the nano-initiator can prevent complex I-derived ROS but according to Figure 4f it is also changing the expression of both anti- and pro-inflammatory markers. This needs to be discussed.

Reviewer #2 Arthritis, nanoparticles, macrophages (Remarks to the Author):

This is a novel, thorough and exciting study using an siRF5-laden efferocytosis-inspired nanoimitator (siRF5@EINI) which consists of siRNA directed against IRF5, coated with PtdSer, cloaked with low molecular weight heparin.

The authors show that systemic administration preferentially distributes to the paw. Biodistribution studies were however performed over a limited period of time, over 48 hours. Given that the actual studies were conducted over a longer time period, biodistribution over longer periods of time, albeit with limitations on cell tracking understood would be a valuable addition. Also were all tracking studies done in non CIA mice to understand if the preferential homing and retention in the paw was induced by the Collagen induced RA?

Did the authors obtain FLS from human synovium from healthy (n=3) vs. RA (n=7) donors? There is no detail in the Materials and Methods about the derivation of the FLS.

In co-culture experiments were human FLS co-cultured with murine macrophages? This is in Figure 3. Why were human FLS co-cultured with murine macrophages? It would make more sense to use human macrophages, differentiated from peripheral monocytes.

I understand the rationale to use H₂O₂ for inducing ROS production. However, the concentrations of H₂O₂ used are extremely high and not justified. Could the authors provide justification for such high concentrations, particularly in light of Ransy et al., 2020 IJMS "Use of H₂O₂ to cause oxidative stress, the catalase issue"

The effect of neutrophil adhesion on HUVEC monolayer (Fig 4B and Suppl Fib 8) are not very convincing. A very small percentage of neutrophils seem to attach to the HUVEC layer to begin with and there is a small reduction in this signal by addition of siRF5@EINI, but not to the extent noted with the addition of the P-selectin antagonist.

The authors are to be commended on performing toxicology studies at 60 days. However the lung pathology as shown in Suppl Fig 11c with siRF@EINI seems more aberrant, but there is little to no discussion on this.

There are numerous grammatical mistakes and typos in the document. A thorough edit should be performed. LMWH is often mis-abbreviated as LWMH, for instance.

Reviewer #3 nanoparticles, arthritis (Remarks to the Author):

This manuscript introduces a nanoparticle therapy for rheumatoid arthritis. Anti-inflammation drugs (siRF5 and Metformin) are surrounded by an ROS sensitive heparin shell (stealth in circulation and targeting of P selectin overexpressed on subsynovial capillary endothelium) and a phosphatidylserine corona. The PtdSer corona mediates uptake via efferocytosis by synovial inflammatory macrophages with a goal of anti-inflammation polarization.

Overall the nanoparticle platform is innovative and in vitro data supports that the goals of each nanoparticle component were met. A reduction in IRF5, CXCL1 and MPO exist in the inflamed joint, plus altered macrophage phenotype and reduced ankle diameter were shown in treated mice using a collagen-induced arthritis murine model. Further the nanoparticle was able to protect against bone erosion.

Minor:

- 1) Grammatical errors need correcting.
- 2) Fig. 1 legend is missing a description of part "c"

- 3) Page 6, line 142-144 incorrectly states that the DLS is determined using TEM
- 4) Fig. 2c, yellow axis label is hard to see
- 5) Fig.2d, x and y-axis labels are missing
- 6) Text on page 7 states that data in Fig. 2g shows ROS elimination of the nanoimitator (does this mean removal of heparin and if yes, is this measured)?
- 7) Fig. 3b shows data from activated FLSs. Are these cells also stimulated with TNF (text just states that macrophages are activated)
- 8) Page 15, line 331, please summarize the dosing in text (number and amount of NPs/injection)

**Response to reviewers' comments:**

We are grateful for the constructive feedback from the reviewers. According to the
reviewers' suggestions, we have added additional data to validate our conclusion. The
manuscript has been substantially revised. Below, please find our point-by-point
responses to the reviewers' comments.

**Reviewer #1**

The manuscript by Zhang and colleagues reports a novel compound designed to target
phagocytic cells under inflammatory conditions. The authors provide proof of concept
data for this compound and show data suggesting it can counteract adverse tissue
remodelling in an in vivo mouse model of rheumatoid arthritis. Such a compound would
be of great value clinically, and may in principle also be useful in other inflammatory
pathologies in which macrophages play a key role. However, in its current form, the
manuscript presents with several severe shortcomings, which would have to be
addressed to elevate it to a level acceptable for publication at Nature Communications,
and ascertain the usefulness of the compound. Most importantly, there appears to be an
overall lack of rigor.

Please find below a detailed list of issues and questions to be addressed.

**Major**

1. Targeting selectivity, specificity, endosomal escape:

**Q1.** The authors are making a strong point about their compound selectively targeting
synovial joints, and more specifically, macrophages within the joints. This claim is not
fully supported by the data shown, and additional data will be necessary to substantiate
this claim. The authors postulate that their compound is targeted to endothelium through
its "shell" made of low molecular weight heparin, which recognizes P-selectin on
endothelial cells. The "corona" of the compound is designed to be responsive to
oxidative stress, making it supposedly more active in an inflammatory environment. In
vivo imaging of mice that have received the compound carrying a fluorescent cargo
(Figure 5) suggests that the compound is indeed enriched in joints, in addition to the

liver and presumably kidney (annotated as K in the figure = explanation of these
abbreviations is missing from the figure legends, see next point about general lack of
rigor). There also appears to be signal at the base of the tail, which is not commented
on, but would be in line with inflammation owing to adjuvant injection in the CIA
model. To interpret this, the authors need to show the whole mouse (not images cut off
near the base of the tail), and as also highlighted below, specify when in relation to the
CIA model the compound was given and include longitudinal clinical data.

**A:** We sincerely appreciate the reviewer's specific comments and constructive
suggestion. We have thoroughly double-checked the manuscript. In the revised
manuscript, the annotations in the figures have been defined in the figure legends.
Following the reviewer's suggestion, we have shown the whole mouse in the IVIS
imaging assay. The updated version of Fig. 5a is shown below.

**Fig. 5a** In vivo fluorescence images of CIA models taken at different time points post-
injected with Dir-labeled EINI and free Dir.

In Figure 6a of the original submission, the therapeutic regimen has been defined.
In the Materials and Methods of our original submission, we have added the statement
“To study therapeutic efficacy with CIA mice, the treatments began at day 7 after the
second immunization. The CIA animals were randomly assigned into four groups (1
50 nmol of siRNA dose per mouse, 10 mg/kg metformin-equivalent dose, $n = 5$). 200 μ L

PBS (G1); free siIRF5 solution (G2); siIN.C@EINI (G3) and siIRF5@EINI (G4) were
 administered by tail vein twice a week from day 28 to 60.” In the revised submission,
 we have highlighted the statement as follows.

“To study therapeutic efficacy in CIA mice, the treatments began on day 7 after the
 second immunization. On day 28, the mice presented arthritis symptoms and were
 randomly divided into four groups ($n = 5$ mice in each group). Four different treatments
 (200 μ L PBS; free siIRF5 solution; siIN.C@EINI and siIRF5@EINI) were
 administered via the tail vein twice a week from day 28 to 60. The dose for each
 formulation was 1 nmol of siRNA dose per mouse, 10 mg/kg metformin-equivalent
 dose.”

We agree with the reviewer that the longitudinal in vivo data should be presented.
 We repeated the in vivo experiments with CIA mice, as illustrated in Fig. 6a. The
 therapeutic efficacy was evaluated by measuring hindpaw swelling volume with a
 plethysmometer and scoring arthritis of paws with a clinical scoring system. We have
 added these data in the revised manuscript as follows.

“Meanwhile, the paws of mice receiving PBS showed a natural disease progression
 and developed severe swelling; this effect was significantly lessened for mice treated
 with siIRF5@EINI, as quantified by measuring the hindpaw volume (Fig. 6b).
 Moreover, blinded scoring of the swelling and redness of mouse paws was conducted
 to evaluate the severity of arthritis in the experimental mice. siIRF5@EINI-treated mice
 had significantly lower arthritis scores than the other groups, indicating the best
 treatment effect (Fig. 6c).”

 **Fig. 6** The severity of arthritis was determined via (b) the relative paw volume changes

over time of the CIA mice and (c) an arthritis scoring system. Data are presented as the
mean \pm s.d. ($n = 5$ biologically independent animals per group). *** $P < 0.001$, **** P
< 0.0001 .

**Q2.** Distribution of the compound should also be assessed in non-inflamed ie treatment-
naïve mice.

**A:** We thank the reviewer for this detailed comment. According to the reviewer’s
suggestion, in the revised submission, we have also added the data for tracking studies
in unimmunized control mice, which are shown below.

“In contrast, the fluorescence signal arising from the paws of nanoimitator-injected
non-arthritic mice reached a maximum in the initial 3 h and decayed very rapidly
(Supplementary Fig.17).”

**Supplementary Figure 17. In vivo biodistribution of the nanoimitator in non-**
**arthritic DBA/1J mice. a,** Fluorescence images of non-arthritic DBA/1J mice
receiving an intravenous injection of the Dir-labeled nanoimitator. **b,** Fluorescence
intensity of the nanoimitator in the joints of the non-arthritic DBA/1J mice as a function
of time. Data are the mean \pm s.d. ($n = 3$ independent experiments).

**Q3.** We are led to believe that this is due to the inflammatory state of the joints, along
with an impaired microvasculature. However, it is entirely unclear at which stage of the
CIA arthritis model the compound was given, and there is no data showing the extent
of inflammation. These data have to be included to draw any further conclusions.
Furthermore, according to supplementary figure 11c, inflammation is also observed in
the lung. This is different from the representative image in the main figure, and requires
clarification. Finally, based on the information given, there is no reason to assume that

targeting is arthritis-specific, rather, this should be true for inflamed tissues more
generally. In principle, it would thus be desirable that the authors showed their
compound is targeted to sites of inflammation in other conditions. In fact, this could
underscore the usefulness of this approach for other diseases, and would thus be of
interest for a wider audience. In the current form, it is impossible to assess if the effects
of the drug are in preventing adverse joint remodelling, or rather restoring these, as
timeline of treatment relative to arthritis induction is uncertain.

**A:** We sincerely appreciate this insightful comment and constructive suggestion. In
Figure 6a of the original submission, the therapeutic regimen has been defined. In the
Supporting Information of our original submission, we have added the statement “To
study therapeutic efficacy with CIA mice, the treatments began at day 7 after the second
immunization. The CIA animals were randomly assigned into four groups (1 nmol of
siRNA dose per mouse, 10 mg/kg metformin-equivalent dose, $n = 5$). 200 μ L PBS (G1);
free siIRF5 solution (G2); siIN.C@EINI (G3) and siIRF5@EINI (G4) were
administered by tail vein twice a week from day 28 to 60.” To avoid any
misunderstanding, we have highlighted the therapeutic regimen in our revised
submission as follows.

“To study therapeutic efficacy in CIA mice, the treatments began on day 7 after the
second immunization. On day 28, the mice presented arthritis symptoms and were
randomly divided into four groups ($n = 5$ mice in each group). Four different treatments
(200 μ L PBS; free siIRF5 solution; siIN.C@EINI and siIRF5@EINI) were
administered via the tail vein twice a week from day 28 to 60. The dose for each
formulation was 1 nmol of siRNA dose per mouse, 10 mg/kg metformin-equivalent
dose.”

In the revised submission, we have also added longitudinal data to evaluate the
treatment effect in vivo. siIRF5@EINI significantly slows RA progression and
effectively inhibits the swelling of the paws and joints of CIA mice, thus demonstrating
a superior therapeutic effect (Fig. 6b and Fig. 6c).

**Fig. 6** The severity of arthritis was determined via **(b)** the relative paw volume changes
 over time of the CIA mice and **(c)** an arthritis scoring system. Data are presented as the
 mean \pm s.d. ($n = 5$ biologically independent animals per group). *** $P < 0.001$, **** P
 < 0.0001 .

Pulmonary involvement is a common extra-articular manifestation of rheumatoid
 arthritis (RA), including lung involvement in the form of airway or parenchymal
 inflammation and fibrosis, and occurs, to some extent, in 60-80% of patients with RA
 (1). A study showed that collagen-induced arthritis (CIA) in DBA/1J mice is
 accompanied by pulmonary inflammation (2). When systemically administered to mice
 with CIA, the nanoimitator effectively accumulated in the inflamed joints. The potential
 inflammatory targeting ability of the nanoimitator would allow the nanoimitator
 accumulate naturally in the lung upon intravenous administration. However, due to the
 lack of the impaired microvasculature, nanoimitators may undergo rapid clearance from
 the lung compared with inflamed joints. Thus, ex vivo analysis of excised organs did
 not show evidence of nanoparticle accumulation in the lung.

According to the reviewer's suggestion, we further evaluated the localization of the
 nanoimitator in the inflamed colon in dextran sulfate sodium (DSS)-treated mice and
 in the inflamed bladder in a cystitis model. We have added the following text to our
 updated submission.

"To further investigate the targeting effects of nanoimitator on inflammation, we
 constructed another acute inflammatory model, a murine model of DSS-induced acute
 colitis, according to the previous protocol (3). We assessed the targeting ability of
 nanoimitator to inflammation sites by intravenous injection of Dir-labeled

nanoimitators into mice after DSS administration. The administered nanoimitator
 accumulated in DSS-inflamed colon; however, nanoimitator was not detected in the
 colon of healthy mice (Supplementary Fig. 18a, b). Consistently, we also observed a
 similar trend in inflammation targeting in the cystitis model (Supplementary Fig. 18c,
 157 d). In summary, the proposed efferocytosis-inspired nanoimitator could be broadly
 applicable for the inflammation-targeted drug delivery.”

 **Supplementary Figure 18. Inflammation-targeted drug delivery of the**
 **nanoimitator. a, b** At 6 h after treating animals with nanoimitator, their organs were
 imaged with an in vivo imaging system (IVIS) (a) and quantitative analysis of the
 fluorescence intensity in the colon was performed (b). Data are presented as the mean
 \pm s.d. ($n = 3$ biologically independent animals per group). *** $P < 0.001$. c, d In vivo
 fluorescence imaging shows the accumulation of the DiR-labeled nanoimitator in
 bladder (c) and quantitative analysis of the fluorescence intensity (d). Data are
 presented as the mean \pm s.d. ($n = 3$ biologically independent animals per group). * $P <$
 0.05.

**Q4.** The statement that “the compound would escape the lyso/endosome to release it in
the cytoplasm” is quite strong and lack of evidence. The author should at least show a
plot profile demonstrating that there is no colocalization and higher magnifications or
use softwares (Imaris?) to show that it is in the cytoplasm and not the endosomes as the
images are not clear enough to conclude (figure 3e).

**A:** We sincerely appreciate this insightful comment and constructive suggestion.
Following the reviewer’s suggestion, we have updated Fig. 3e, f and added data on the
quantitative analysis of co-localization of Cy5.5-siIRF5 with endo/lysosomes labeled
with LysoTracker Green as follows.

“Next, we wished to ascertain whether the siIRF5-laden nanoimitator would escape
lyso/endosomes and release the siRNA in cytoplasm after efficient uptake by the
macrophages. Co-localization with CLSM revealed that the nanoimitators were
internalized by the cell via an endocytosis pathway and then escaped from
lyso/endosomes, during which the siRNA were released into the cytoplasm (Fig. 3e).
These observations were confirmed by quantitative analyses of the co-localization of
Cy5.5-siIRF5 with endo/lysosomes in confocal fluorescence images using Manders’
coefficients M1 and M2. As shown in Fig. 3f, M1 is close to 1 at 1 h, indicating the
nanoimitators were taken up via endocytosis and stayed in endo/lysosomes. In contrast,
M1 is much less than 1 at the other time points (particularly 6 h), indicating that the
nanoparticles can achieve effective endo/lysosomal escape of the encapsulated siRNA.
The M2 data show that nearly all of the endo/lysosomes were not merged with Cy5.5-
siIRF5 at 6 h. Collectively, these data suggest that the low-pH-triggered structural
disassembly of the nanoimitator facilitates endo/lysosomal escape for cytosolic
delivery of siRNAs.”

**Fig. 3 e** Fluorescence visualization of siRNA localization in macrophages 1, 2 or 6
 196 hours after incubation with siIRF5@EINI. The cell nuclei were stained using DAPI
 (blue), the endo/lysosomes were stained using LysoTracker Green (green), and siIRF5
 was labeled with Cy5.5 (red). Scale bar, 20 μ m. **f** Quantitative analysis of the co-
 localization of Cy5.5-siIRF5 with endo/lysosomes labelled with LysoTracker Green.
 Manders' coefficient M1 denotes the fraction of Cy5.5-siIRF5 overlapping with
 LysoTracker Green, and M2 denotes the fraction of LysoTracker Green overlapping
 with Cy5.5-siIRF5. The coefficients are close to 1 if they are highly co-localized ($n =$
 5 images from three independent experiments). Data are shown as the mean \pm s.d. ****** P
 < 0.01 , ******* $P < 0.001$, ******** $P < 0.0001$.

**Q5.** Another major concern relates to the specificity of targeting Irf5. The scrambled
 siRNA control has pronounced effects already (Figure 6). This appears to not be
 mentioned or discussed. Equally, the relevant statistics for the comparison between this
 and other negative controls are not shown. This point is a major shortcoming, putting
 in question the selectivity of their targeting approach.

- Similar to the specificity, the data supporting endosomal escape of the compound are
 not fully convincing in their current form.

**A:** We thank the reviewer for this comment. As is well known, in the context of
autoimmune or inflammatory disease pathologies, metformin has immunosuppressive
effects, such as inhibiting mitochondrial ROS, inducing M2 macrophage polarization
and interfering cytokine synthesis (4, 5). To enable macrophage-targeted gene delivery
and efficient lysosome escape, siRNA-laden metformin-Zn²⁺-based nanocores were
introduced as solid nanoimitator core. The metformin-laden nanoformulation can
induce immunosuppressive effects.

In our original submission, we systemically investigated whether exposure to
metformin could induce immunosuppressive effects. Macrophage-derived cytokines
were quantitatively analyzed by qRT-PCR, which indicated M2 polarization of
macrophages (Fig. 4f). Similar results were obtained with images of
immunofluorescence staining of BMDMs (Fig. 4g and Fig. 4h). Treatment with the
nanoformulation carrying metformin along induced a strong increase in the size of the
M2-like subpopulation with highly expressed CD206, consistently indicating that
metformin in the nanoimitator skewed the phenotype of macrophages from M1 to M2.
Moreover, metformin can target the electron transport chain of mitochondria and thus
reprogram mitochondrial metabolism. As shown in Supplementary Fig. 10a, b,
metformin-containing treatment significantly prevented excessive ROS production,
which synergistically led to immunoregulatory phenotypic conversion. These data
suggested that metformin-containing nanoimitator promoted polarization to the
immunoregulatory M2 phenotype and thus displayed obvious therapeutic potency.

Following the reviewer's constructive suggestion, in the revised submission, we
further added the relevant statistics for the comparison between siN.C@EINI and the
negative control in the updated version of Fig. 6. siIRF5@EINI-treated mice had
significantly lower arthritis scores than the other groups, indicating the best treatment
effect (Fig. 6c). Although it was not as good as that in the siIRF5@EINI group, a
relatively good arthritis score was observed in the siN.C@EINI group. However,
serious cartilage damage and significant bone loss were observed in the siN.C@EINI
group. The difference in efficacy between the two groups may be attributed to the

existence of siIRF5, which induces a cascade of anti-inflammatory events via
macrophage phenotypic regulation and serially confers a chondroprotective effect.

In the revised manuscript, we have updated Fig. 3e, f and added the data from the
quantitative analysis of the co-localization of Cy5.5-siIRF5 with endo/lysosomes
labeled with LysoTracker Green.

**Q6. 2. Mechanism of action:** The authors don't confirm in vivo what they see in vitro
with the huvecs. Is the compound reaching endothelial cells and blocking the
transmigration of neutrophils? Are ROS present in their model (in vivo)?

**A:** We sincerely appreciate this insightful comment and constructive suggestion. In our
original submission, we determined that the nanoimitator could efficiently accumulate
intra-articularly after systemic administration (Fig. 5) and subsequently decreased the
unchecked infiltration of neutrophils in the RA joints (Fig. 6e and Fig. 6i).
Immunohistochemical staining of joint sections with anti-myeloperoxidase (MPO)
antibodies, a well-established method to measure neutrophil infiltration, indicates that
the nanoimitator competitively binds P-selectin on vascular endothelial cells,
decreasing the infiltration of neutrophils into arthritic joints (Fig. 6e and Fig. 6i). In the
revised submission, we have highlighted these data as follows.

"As shown in Fig. 6d and Fig. 6h, the siIRF5@EINI-treated group showed minimal
MPO expression with few positively stained cells in the inflamed sites, while many
MPO-positive cells were detected in the control joints. Competitive binding of P-
selectin on vascular endothelial cells can decrease the infiltration of neutrophils into
arthritic joints and thus promote restoration of the articular immune microenvironment."

"To further confirm that the nanoimitator retards the articular trafficking of
neutrophils, immunofluorescence of synovial slices was used to evaluate inflammatory
neutrophils infiltration. We found that the iv injected nanoimitator could adhered to the
blood vessel walls of the synovium (Fig. 4b). In contrast, control nanoparticles did not
specifically attach to the synovium vasculature. With the capability of vessel binding,
the nanoimitator thus interrupted the endothelium tethering of neutrophils and
prevented their subsequent extravasation."

**Fig. 4b** Immunofluorescence images of neutrophil infiltration in the synovium after
 treatment with PBS, control nanoparticles (control NPs) or siRF5@EINI. Neutrophils
 were immunostained with an anti-Ly-6G antibody (green) and endothelial cell were
 stained using CD31 (blue). The nanoparticles were visualized by siRNA labeled with
 Cy5.5 (red). White arrows indicate firmly adhered neutrophils. Yellow arrows indicate
 the co-localization of the nanoimitator and endothelial cells labeled with anti-CD31
 antibodies. ($n = 3$ independent experiments). Scale bars, 50 μm .

Reactive oxygen species (ROS) have distinct contribution to the destructive,
 proliferative synovitis of rheumatoid arthritis and play a prominent role in
 inflammatory cell-signaling events (6). The CIA mouse model is characterized by
 chronic inflammation, and the immunologic and histopathological lesions in CIA are
 similar to that in human rheumatoid arthritis (7). Recent insights have indicated that
 ROS are upregulated in the CIA model mice (8, 9).

**Q7. 3.** Arthritis model: Further to the comments above, the authors need to provide
 longitudinal clinical data on any mice they claim are arthritic. In the current form, the
 only data presented on disease symptoms is histology (Figure 6), presumably taken at
 disease endpoint, though this too needs to be clarified. Throughout the manuscript, the
 authors need to include data for clinical parameters, i.e. clinical scoring of joint
 inflammation as well as measurements for paw swelling and arthritis incidence within
 treated cohort.

**A:** We thank you for this comment and constructive suggestion. Following this
 suggestion, we repeated the in vivo experiments with DBA/1J mice as illustrated in Fig.
 6a. The therapeutic efficacy was evaluated by measuring hindpaw swelling volume by
 a plethysmometer and scoring arthritis in paws with a clinical scoring system. We have
 added these data in the revised manuscript as follows.

“Meanwhile, the paws of mice receiving PBS showed a natural disease progression
 and developed severe swelling; this effect was significantly lessened for mice treated
 with siRF5@EINI, as quantified by measuring the hindpaw volume (Fig. 6b).
 Moreover, blinded scoring of the swelling and redness of mouse paws was conducted
 to evaluate the severity of arthritis in the experimental mice. siRF5@EINI-treated mice
 had significantly lower arthritis scores than the other groups, indicating the best
 treatment effect (Fig. 6c).”

 **Fig. 6** The severity of arthritis was determined via (b) the relative paw volume changes
 over time of the CIA mice and (c) an arthritis scoring system. Data are presented as the
 mean \pm s.d. ($n = 5$ biologically independent animals per group). $***P < 0.001$, $****P$
 < 0.0001 .

To evaluate arthritis incidence, mice with an average severity score of 1 or greater
 were judged to be arthritis mice (8). Actually, in our original submission, we
 investigated the therapeutic potency of the nanoformulations using the established CIA
 mouse model. The CIA mouse model was established by following a previously
 published protocol with immunizations on days 1 and 21. On day 28, the mice presented
 arthritis symptoms and were randomly divided into four treatment groups ($n = 5$ mice
 in each group). Four different treatments (200 μ L PBS; free siRF5 solution;

siIN.C@EINI and siIRF5@EINI) were administered by tail vein twice a week from
319 day 28 to 60. To avoid any misunderstanding, we have highlight the therapeutic regimen
in our revised submission as follows.

“On day 28, the mice presented arthritis symptoms and were randomly divided into
four groups ($n = 5$ mice in each group). Four different treatments (200 μ L PBS; free
siIRF5 solution; siIN.C@EINI and siIRF5@EINI) were administered via the tail vein
twice a week from day 28 to 60.”

4. Rigor, other:

**Q8.** Several legends to supplementary figures are missing.

**A:** We thank the reviewer for this detailed comment. The figure legend has been defined
in the corresponding figures.

**Q9.** In most experiments, a low number of replicates was used. Additional clarification
is needed if data points represent biological replicates or technical replicates.

For example, in Figure 6, is $n=5$ per group or a total $n=5$? If the latter, this will need
repeating.

**A:** Thank you for the detailed comment. In our original submission, each experiment
was repeated independently at least three times. Replicates represent different mice
subjected to the same treatment. In CIA studies, $n = 5$ biologically independent animals
338 per group. We apologize for the ambiguous description in our original submission. To
339 avoid any misunderstanding, we have highlighted the number of replicates for
experiments in our revised submission.

**Q10.** Figure 6b: There are 10 datapoints, but the n is supposedly 5 mice. This suggests
these are individual ankles corresponding to 5 mice? This is misleading. A mean of both
ankles/mouse might be better.

**A:** Thank you for the constructive suggestion. Following the reviewer’s suggestion, the
figure has been corrected and defined as follows: “Changes in the average hind ankle
in diameter on day 60 after CIA induction compared to day 0.”

**Supplementary Figure 21b.** Change in the average hind ankle in diameter on day 60
 after CIA induction compared to day 0. Data are presented as the mean \pm s.d. ($n = 5$
 biologically independent animals per group). $**P < 0.01$, $***P < 0.001$.

**Q11.** Figure 6: Histological images seem to be taken in different joint regions, which
 makes it hard to compare. The authors should try to quantify around the same areas and
 do so on several slides. Then show adjacent slides e.g. in f, g and h.

**A:** Thank you for this comment and constructive suggestion. In the revised submission,
 we have shown the same areas in the histological analysis of inflamed joints. The
 updated versions of Fig. 6d, 6f and 6g are shown below.

**Fig. 6 d** The infiltration of neutrophils in arthritic joints in each group. Joint sections in
 each group were stained with anti-myeloperoxidase (MPO) antibodies. Scale bar, 200
 362 μm . **f, g** Histological analysis with H&E (scale bar 200 μm) (**f**) and safranin O staining

(scale bar 200 μ m) (g) of the joint tissues excised from the mice after different
treatments. ($n = 5$ biologically independent animals per group).

**Q12.** Characterisation of “inflammatory” macrophages: CD206 and iNOS are limited
markers. iNOS⁺ staining seems very strong compared to CD206⁺ cells.
Counterstaining with a pan macrophage marker is required. Phenotypic characterisation
of macrophages should be confirmed by flow cytometry. Tim4 is a phosphatidyl
receptor and is associated with tissue residency and homeostasis in other tissues. Are
the macrophages that take up the compound Tim4⁺ or can all macrophages take it up
to the same degree?

**A:** We are grateful for the reviewer’s insightful comments and constructive suggestion.
We further confirmed the SIM phenotypic subset induction by siRF5@EINI treatment.
Following the reviewer’s suggestion, we co-stained for the pan macrophage marker
F4/80 (red) and either iNOS (green) or CD206 (green) to determine phenotypic
alterations of synovial macrophage induction by siRF5@EINI treatment.
Immunofluorescence staining demonstrated that CD206⁺ macrophages were rare in the
PBS-treated group, while siRF5@EINI treatment led to abundant CD206 expression
In contrast, the amount of iNOS co-localization with F4/80 was quite considerable in
the PBS-treated group and was significantly reduced with siRF5@EINI treatment.

“Immunofluorescence analysis indicated that the proportion of the M1
subpopulation was significantly decreased, and the proportion of the M2 subpopulation
was markedly increased in the siRF5@EINI group (Fig. 6e and Fig. 6i).”

**Fig. 6e** Synovium sections were stained for the pan macrophage marker F4/80 (red)
and costained for either CD206 (green) or iNOS (green) to determine the phenotype of

SIMs in CIA models mice treated with different formulations. Nuclei were stained with
 DAPI (blue). White arrows indicate double-positive cells ($F4/80^+CD206^+$ and
 $F4/80^+iNOS^+$). Scale bars, 50 μm . ($n = 5$ biologically independent animals per group).

**Figure 6i** Quantification of $F4/80^+CD206^+$ staining and $F4/80^+iNOS^+$ staining in
 macrophages. Data are presented as the mean \pm s.d. ($n = 5$ biologically independent
 samples). $***P < 0.001$. NS, not significant.

Phenotypic changes in the synovial macrophage subpopulation were further
 explored using flow cytometry analysis. Consistently, an increased population of
 macrophages with the M2 phenotype was also observed with $siRF5@EINI$ treatment,
 demonstrating that $siRF5@EINI$ induced an M2-phenotype shift in synovial
 macrophages (Supplementary Fig. 26).

**Supplementary Figure 26. Flow cytometry analysis of M1 and M2 macrophage**
 **populations in synovial tissue from different treatment groups. Representative flow**

cytometric analysis images (a) and relative quantification of M2-like macrophages
(CD206⁺) and M1-like macrophages (CD80⁺) gating on CD11b⁺F4/80⁺CD45⁺ cells (b).
For the gating strategy for macrophages analysis refer to Supplementary Fig. 28. Data
are presented as the mean ± s.d. (*n* = 5 biologically independent samples). *****P* <
0.0001.

As is well known, the engulfment of apoptotic cells requires two steps. In the
tethering step, the phosphatidylserine (PtdSer) receptor Tim4 tightly binds PtdSer on
apoptotic cells and recruits it to the macrophage surface. In the tickling or uptake step,
soluble proteins such as protein S/Gas6 or MFG-E8 bind PtdSer on apoptotic cells and
activate their receptors (MerTK or integrin, respectively) on phagocytes, leading to
PtdSer-mediated engulfment (10, 11). Hence, both tethering and tickling are essential
steps in the efficient engulfment of apoptotic cells.

However, macrophages are heterogeneous. Accordingly, macrophages seem to use
a different set of PtdSer receptors to engulf apoptotic cells that is dependent on tissue
context. Either cell surface PtdSer receptor-Tim4 and CD300 or a secreted protein that
strongly binds PtdSer on apoptotic cells and a membrane protein on the macrophage
can trigger PtdSer-mediated professional engulfment of apoptotic cells (10). Thus,
synovial macrophages recognize the nanoimitator via surface ligands or secreted
proteins and actively internalize the nanoimitator.

Minor

1. Rigor, additional clarification needed:

**Q13.** Figure 2d: Axes need to be annotated.

**A:** We appreciate the reviewer's detailed comment. The axes have been annotated in
the updated Fig. 2d.

**Q14.** The authors used HUVECS cocultured with FLS and BMDM. How many wells
were used per experiments? Is it 1 well per experiment? Are HUVECS derived from
different batches/donors?

**A:** We thank the reviewer for the detailed comment. In Fig. 3d of the original
submission, the coculture experiments in the in vitro assay were indeed repeated three
434 times rather than one. In the figure legend of our original submission, we have added
the statement “In all datasets, the experiments were repeated three times independently.”

Notably, in our original submission, the HUVECs used in the in vitro assay were
obtained from the cell bank of the Chinese Academy of Sciences (Shanghai, China).
We apologize for the ambiguous description in our original submission. To avoid any
misunderstanding, we have highlighted the culture of HUVECS for the in vitro assay
in our revised submission as follows.

“HUVECs were obtained from the cell bank of the Chinese Academy of Sciences
(Shanghai, China) and maintained in DMEM (Gibco, USA) supplemented with 10%
FBS and 1% penicillin/streptomycin. All cells were incubated at 37 °C in a humidified
atmosphere with 5% CO₂.”

**Q15.** Figure 3c: The authors state that the compound is specifically engulfed by
macrophages in the presence of ROS existed, however, FLS show an MFI of 10.000. Is
this the background level? The authors should show data and include raw flow
cytometry plots comparing the fluorescence of both cell types with and without the
compound to conclude on specificity.

**A:** We thank the reviewer for this detailed comment and constructive suggestion.
According to the reviewer’s suggestion, in the revised submission, we have also added
the data for flow cytometric analysis of cellular uptake in the updated version of Fig.
3b, which is shown below.

**Fig. 3b** Representative flow cytometric analysis of nanoimitators internalized by TNF-
α activated macrophages or FLSs.

**Q16.** Figure 3b, e: Add quantification in addition to representative images.

**A:** We sincerely appreciate the reviewer's specific suggestion. In the revised
submission, we have also added the data for flow cytometric analysis of cellular uptake
in the updated version of Fig. 3c. Additionally, we determined the co-localization of
Cy5.5-siIRF5 with endo/lysosomes labeled with LysoTracker Green in the confocal
fluorescence images using Manders' coefficients M1 and M2 (Fig. 3f).

**Q17.** Figure 3e: The statement that "the compound would escape the lyso/endosome to
release it in the cytoplasm" is strong and currently lacking enough supporting evidence.
The authors should at least show data demonstrating that there is no colocalization and
ideally use higher magnification views. Imaging analysis software should be used to
quantify cytoplasmic versus endosomal localization, since the images shown currently
are not clear enough to conclude.

**A:** We sincerely appreciate this insightful comment and constructive suggestion.
Following the reviewer's suggestion, we have updated Fig. 3e and Fig. 3f, and added
the data for quantitative analysis of the co-localization of Cy5.5-siIRF5 with
endo/lysosomes labeled with LysoTracker Green as follows.

**Fig. 3 e** Fluorescence visualization of siRNA localization in macrophages 1, 2 or 6
 478 hours after incubation with siIRF5@EINI. The cell nuclei were stained using DAPI
 (blue), the endo/lysosomes were stained using LysoTracker Green (green), and siIRF5
 were labeled with Cy5.5 (red). Scale bar, 20 μ m. **f** Quantitative analysis of the co-
 localization of Cy5.5-siIRF5 with endo/lysosomes labeled with LysoTracker Green.
 Manders' coefficient M1 denotes the fraction of Cy5.5-siIRF5 overlapping with
 LysoTracker Green, and M2 denotes the fraction of LysoTracker Green overlapping
 with Cy5.5-siIRF5. The coefficients are close to 1 if they are highly co-localized ($n =$
 5 images from three independent experiments). Data are shown as the mean \pm s.d. $**P$
 < 0.01 , $***P < 0.001$, $****P < 0.0001$.

**Q18.** Throughout: Where was mouse synovium isolated from, which joints?

**A:** We thank the reviewer for the detailed comment. In the updated Supplemental
 Information of the revised submission, we have added the detailed methods for the
 experiments.

"Mouse synovial tissues were isolated from knee joints."

**Q19.** The data for gait analysis using the cat walk system are virtually impossible to
 interpret without additional information: What is shown, what does it mean?
 Quantification is needed for all experimental animals analysed.

**A:** We appreciate the reviewer’s insightful comments and constructive suggestion. As
 is well known, the improved clinical scores that measure the severity of CIA
 corresponded to changes in multiple gait parameters that reflect functional recovery
 (decrease in stride frequency, increase in stride length and stance times) (12). In the
 revised manuscript, we further performed gait analyses of mice in the CIA model,
 including testing for stride frequency, stride length and stance time, after each treatment
 following the reviewer’s suggestion. We have added the following text to our updated
 submission.

“Regarding the gait analyses (Supplementary Fig. 22b), the siRF5@EINI-treated
 group showed significant improvements in various gait indices for the mice in this
 model, including a decrease in stride frequency, increase in stride length, and improved
 stance times.”

**Supplementary Figure 22b.** Main related indices for gait analysis in mice. Stride
 frequency (the average number of times a paw contacts the belt per second) increased
 linearly with increasing clinical scores. Stride length (the distance between the initial
 contact of the same paw in a complete stride) progressively decreased with increasing
 clinical scores. The stance time (the weight-bearing portion of the stride in which the
 paw remains in contact with the belt) decreased progressively with increasing clinical

scores. Data are presented as the mean \pm s.d. ($n = 3$ independent experiments). $*P <$
0.05 , $**P < 0.01$. NS, not significant.

**Q20.** Figure 4h: It is unclear what the ratio shown refers to and how it was determined.
Please clarify.

**A:** We thank the reviewer for this detailed comment. The ratio of CD206 to iNOS
indicates the quantitation of the relative fluorescence intensity of CD206⁺ staining
versus iNOS⁺ staining in each group ($n = 3$). We apologize for the ambiguous
description in our original submission. In this revised version, we have added a clearer
description as follows.

“Quantitative analysis of the relative fluorescence intensity of CD206⁺ staining
versus iNOS⁺ staining in each group ($n = 3$). At least three images per group were
acquired for quantification, and positively stained areas were evaluated with ImageJ
software. (<http://imagej.net/>.”

**Q21.** Figure 6: Use consistent annotation of experimental groups throughout figure (and
rest of the manuscript) – i.e. do not use G1, 2, 3, 4 in i-m, as this misleading.

**A:** We thank the reviewer for this detailed comment. The annotation of experimental
groups was made consistent in the revised submission.

**Q22.** Rat tissues are mentioned in the supplementary data (Figure S15). Please clarify.
Manuscript otherwise states mice as the only animal model used.

**A:** We thank the reviewer for this detailed comment. In our original submission, all of
the in vivo assays were comprehensively investigated in a CIA murine model. We
apologize for our mistake in stating that the joint tissues were isolated from rats in our
original submission. We have corrected the mistake in our revised submission.

**Q23.** Summary figures or cartoons outlining proposed mechanism underlying the
effects of their compound should be moved to the end of the manuscript and clearly
highlighted as “proposed mechanism”.

**A:** We sincerely appreciate the reviewer’s specific suggestion. According to the
 reviewer’s suggestion, in the revised submission, the summary figures have been
 moved to the end of the manuscript and clearly highlighted as the “proposed
 mechanism”.

**Fig. 7 Overview of the proposed mechanism of siIRF5@EINI-induced**
 **inflammatory regulation in RA.** To manipulate the locoregional exposure of the
 PtdSer corona of the designed nanoimitator intra-articularly, a benzeneboronic acid
 pinacol ester group was used to make the nanoimitator responsive to local ROS.
 The ROS-abundant inflamed joint microenvironment triggers PtdSer presentation of
 the nanoimitator. In an efferocytosis-like manner, the PtdSer-coroneted core was in turn
 phagocytosed by synovial inflammatory macrophages, which synergistically
 terminated the SIM-initiated pathological cascades that proinflammatory cytokine
 production, oxidative stress, and recruitment of neutrophils.

 **Q24.** Line 64: The pathological cascades initiated by SIM should be specified (ROS
 production, cytokine production). This is relevant for the entire scope of the manuscript.

**A:** We sincerely appreciate the reviewer’s specific suggestion. In the revised
 manuscript, we further highlighted the critical role of SIMs in initiating and
 perpetuating RA progression following the reviewer’s suggestion. We have added the
 following text to our updated submission.

“Among these cells, synovial inflammatory macrophages (SIMs) play a pivotal role
 in orchestrating the cytokine environment by releasing various types of
 proinflammatory cytokines and reactive oxygen species, which are thought to underlie
 articular immune dysfunction, synovitis and ultimate joint erosion.”

**Q25.** 2. Language, general: While it is appreciated that the authors may not be native
English speakers, the manuscript requires editing for syntax, grammar, spelling and
overall clarity throughout. Non-exhaustive examples are:

- line 102: efferocytosis-mimetic = should be efferocytosis-mimicking

- line 256: over-zealous?

- line 423: please rephrase sentence

**A:** We thank the reviewer for this detailed comment. We have thoroughly double-
checked the manuscript. In the revised manuscript, the typographical and grammatical
errors have been corrected.

**Q26.** 3. Terminology, acronyms: There appears to be an overuse of non-standard
acronyms, e.g. NE instead of neutrophils. This makes reading of the article
unnecessarily difficult. Similarly, some terms may not be commonly used or agreed
upon, e.g. “payload” (instead of drug cargo?).

**A:** We thank the reviewer for the comments and detailed suggestion. We have
thoroughly double-checked the manuscript. In the revised manuscript, the non-standard
acronyms and terminology have been corrected.

**Q27.** 4. Figures, legends, annotations: Where histology or fluorescent imaging is shown,
the annotation of what is stained for needs to be included in the main figure, not just
the legend. This is common practice and enables the reader to easily identify what they
are looking at.

**A:** We thank the reviewer for this constructive suggestion. The annotation has been
added to the updated version of the figures where histology or fluorescent imaging is
shown.

**Q28.** 5. Targeting approach: The authors should include a discussion or rationale for
using this type of chemistry and compare it to other approaches used (in preclinical
models) to target macrophages, such as liposomes (see e.g. PMID: 31375534). This
will be [particularly useful for a non-expert audience, i.e. readers with expertise in

macrophages and inflammatory disease, but not drug design.

**A:** We thank the reviewer for this insightful comment and literature information. It is
well accepted that macrophages internalize apoptotic cells specifically and promptly,
which is closely linked to the presence of phosphatidylserine (PtdSer) on the surface.
The exposure of PtdSer on the outer leaflet of the plasma membrane of apoptotic cells
is a major eat-me signal for macrophages. Our strategy exploits the principle underlying
the phagocytosis of apoptotic cells, called efferocytosis, which is a critical innate
function of macrophages. Actually, in our original submission, we have added the
statement as follows.

“The phagocytosis of the apoptotic cells, called efferocytosis, is one of the critical
innate functions of macrophages, which maintains tissue homeostasis. During
efferocytosis, phosphatidylserine (PtdSer) exposure on the outer leaflet of the plasma
membrane in the apoptotic cells is a key "eat-me" signal for macrophages. PtdSer
coronation may therefore enhance the targeted-internalization of nano-formulation by
macrophages. However, after systemic administration, PtdSer-coroneted nano-
formulation may be up-taken by the panmacrophages, which results in severe adverse
effects. Of note, intra-articular oxidative stress caused by high level of reactive oxygen
species (ROS) is one of the typical characteristics of RA lesions. As such, ROS-
responsive exfoliation may be an efficient way to manipulate the locoregional exposure
of PtdSer corona of the designed nano-formulation intra-articular, which enabled the
specific-phagocytosis of the nanoformulation by synovial inflammatory macrophages
(SIMs).”

Moreover, we have also highlighted the innovations of our current strategy.
Compared to previous work, our strategy is unique in multiple aspects.

First, the principle underlying the strategy to target macrophage is mimicking the
process of apoptotic cell clearance by macrophages, called efferocytosis, which is a
critical innate functions of macrophages.

Second, for macrophage-targeted drug delivery, an siIRF5-carrying efferocytosis-
inspired nanoimitator (siIRF5@EINI) was for the first time sequentially assembled
from a drug-based core with an oxidative stress-responsive PtdSer corona and an outer

shell of low-molecular-weight-heparin (LMWH). (i) With the shielding of LMWH,
siIRF5@EINI was endowed with stealth properties in the circulation, an enhanced
retention in inflamed regions, and a blocking function of P-selectin that retards the
articular trafficking of neutrophils. (ii) The design of the oxidative stress-responsive
PtdSer corona endows siIRF5@EINI with ROS-responsive PtdSer presentation, which
enables the specific phagocytosis of the nanoformulation by SIMs.

Finally, the traditional therapeutic approaches for inflammatory disease focus on
suppressing the inflammatory process, for instance, by inhibiting cytokines or depleting
M1 macrophages (13, 14). In our study, by manipulating of the shielding and exposure
of PtdSer in the nanoformulation, we demonstrated that SIMs in RA joints could be
precisely targeted in situ. Functionally, the EINI effectively repolarized SIMs to an
immunoregulatory phenotype in situ, serially terminating SIM-initiated pathological
cascades and ultimately reestablished the articular immune homeostasis.

6. Methods:

**Q29.** Please specify how healthy donor synovial tissue was obtained.

**A:** We thank the reviewer for the detailed comment. The synovial tissue from healthy
individuals was isolated during orthopedic procedures performed by orthopedic
surgeons.

**Q30.** How did you check the purity of your sorting experiments or in vitro culture
experiments?

**A:** We thank the reviewer for the detailed comment. In Supplementary Figure 5 and
Supplementary Figure 7 of our original submission, the purity of BMDMs and
neutrophils were already shown as follows.

“We characterized the F4/80 and CD11b double positive mouse bone marrow
derived macrophages (BMDMs) with flow cytometry (Supplementary Fig. 9).”

“Flow cytometric analysis of the purity of neutrophils doubly stained with FITC-
conjugated Ly-6G and PerCP-Cy5.5-conjugated CD11b antibodies (Supplementary Fig.
12).”

In the revised submission, we have also added the data for synovial macrophage
sorting in the updated version of Supplementary Fig. 24, which is shown below.

“Macrophages were FACS-sorted on a BD FACS Aria Cell Sorting System (BD
Biosciences) with >97% purity by the Montpellier RIO imaging IGMM - Cytometry
platform.”

**Supplementary Figure 24.** Gating strategy used to identify synovial macrophages in
rheumatoid arthritis tissue.

**Q31.** line 618: how were neutrophils dissociated from the HUVEC monolayer? In
Supplementary Figure 8, they were shown as two independent populations.

**A:** We thank the reviewer for the detailed comment. For flow cytometric analysis, cells
were scraped and collected after a PBS wash and then analyzed with a Beckman Coulter
Gallios flow cytometer. The results shown as two independent populations. Among
these populations, the Ly-6G-positive in HUVECs were thought to be adherent to
neutrophils and the other population was unbound HUVECs.

To more thoroughly investigate the influence of the nanoimitator on
neutrophil/HUVEC adhesion and quantitative analysis of neutrophils adhered to
HUVEC monolayers, in the revised submission, we further verified the blocking effect
of the nanoimitator on neutrophil-HUVEC adhesion in the updated version of
Supplementary Figure 8, which is shown below.

“HUVECs (2×10^5) were seeded in 6-well plates and pre-stimulated by incubation
for 6 h with TNF- α (10 ng/ml). Then the cells were treated with RPMI medium, P-
selectin antagonist (Human P-selectin/CD62P antibody, R&D Systems) and the
nanoimitator for 30 min. Activated neutrophils (1×10^6) were labeled with 5 μ M
calcein-AM and seeded on top of the treated endothelial monolayer. After 30 min at
37 $^{\circ}$ C under gentle shaking, each well was washed twice with PBS to remove the non-

adhered cells and the cells in each well were lysed with 2 mL dimethyl sulfoxide
(DMSO), and their fluorescence intensities were detected by a fluorospectro-
photometer at Ex = 495 nm and Em = 515 nm. The in vitro adhesion experiment results
further confirmed that neutrophil/endothelial adhesion was weakened after incubation
with the nanoimitator (Supplementary Fig. 14).”

**Supplementary Figure 14. Quantitative analysis of neutrophils adhered to**
**HUVEC monolayers.** Neutrophils were stained with calcein-AM. The cells in each
well were lysed with 2 mL dimethyl sulfoxide (DMSO), and their fluorescence
intensities were detected by a fluorospectro-photometer at Ex = 495 nm and Em = 515
697 nm. Data are reported as the mean ± s.d. ($n = 3$ independent experiments). **** $P <$
0.0001. NS, not significant.

**Q32.** line 660: Please specify the site of injection for the immunisation boost

**A:** We thank the reviewer for the detailed comment. We apologize for the ambiguous
description in our original submission. To avoid any misunderstanding, we have
highlighted the mouse models of inflammatory arthritis in our revised submission as
follows.

“Next, six-week-old male DBA1/J mice were injected intradermally at the tail root
with 200 μ g of bovine type II collagen (2 mg/ml) emulsified in 100 μ l of complete
Freund’s adjuvant (4 mg/ml). On day 21, the mice received a booster immunization of
type II collagen emulsified in incomplete Freund’s adjuvant at the tail root.”

**Q33.** line 721: use linear regression statistics to account for all covariates (e.g. sex)

**A:** We appreciate the reviewer's comments. In fact, in our original submission, we have
controlled for the covariates associated with disease activity, including sex and age. We
apologize for the ambiguous description in our original submission. To avoid any
misunderstanding, we have highlighted the characteristics (e.g., sex and age) of the
DBA1/J mice that were used in the in vivo assay in our revised submission.

**Q34.** 7. BMDM: The authors' definition of bone marrow-derived macrophages is
misleading. These are not primary macrophages isolated from mouse tissues, but rather
in vitro generated from isolated progenitors.

**A:** We thank the reviewer for this detailed comment. We agree with the reviewer that
the bone marrow-derived macrophages were generated from isolated progenitors rather
than primary macrophages isolated from mouse tissues. We apologize for the
misleading description in our original submission. To avoid any misunderstanding, we
have highlighted the definition of bone marrow-derived macrophages in our revised
submission as follows.

"Bone marrow-derived macrophages (BMDMs) were differentiated from bone
marrow monocytes using methods as previously described."

8.Other:

**Q35.** - Figure 1a: has no added value. The mechanisms of action of the compound seem
to not depend on the presence or absence of ACPA. Furthermore, the contribution of
ACPA titers was not further investigated in the CIA model. The authors could include
a section in the discussion, in which they address if and how their compound might
behave differently in ACPA positive and negative RA.

**A:** We sincerely appreciate this insightful comment and constructive suggestion. The
ACPA⁺ subset of the disease accounts for approximately two-thirds of all cases of RA
and generally has a more severe disease course (15). ACPA in the inflamed synovium
have been shown to associate with citrullinated antigens to form ACPA-immune
complexes (ACPA-ICs), resulting in progression of the inflammatory process (16).

ACPA-ICs subsequently drive a strong proinflammatory cytokine response in
macrophage colony-stimulating factor differentiated macrophages (17). Furthermore,
ACPA might contribute to osteoclastogenesis and the development of joint pathology
by facilitating the proinflammation polarization of macrophages (18, 19).

Interferon regulatory factor 5 (IRF5) is a master regulator in defining the classical
inflammatory phenotype of macrophages (20) and translates various signals related to
SIMs in RA synovium (21, 22). In humans, gain-of-function polymorphisms in the
IRF5 gene have been associated with an increased risk of developing autoimmune
disease including RA (23, 24). Bioinformatically, we found that the expression of IRF5
had a positive correlation with the ACPA titer in the RA synovium (Fig. 1a).
Consistently, in patients with ACPA⁺ RA, the expression level of the IRF5 protein was
obviously elevated (Fig. 1b). Targeted silencing of IRF5 in SIMs may therefore be an
efficient strategy that could facilitate the anti-inflammatory polarization of
macrophages and thus abort the SIM-initiated cascades in ACPA⁺ RA. Following the
reviewer's constructive suggestion, we have pointed to the potential role of
nanoimitator in ACPA⁺ RA in the Discussion section of the revised submission as
follows.

"Patients with RA can be divided into two major subsets based on the presence
versus absence of anti-citrullinated protein antibodies (ACPAs) (25). The ACPA⁺ subset
of the disease accounts for approximately two-thirds of all cases of RA and generally
has a more severe disease course (15). ACPA in the inflamed synovium have been
shown to associate with citrullinated antigens to form ACPA-immune complexes
(ACPA-ICs), resulting in progression of the inflammatory process (16). ACPA-ICs
subsequently drive a strong pro-inflammatory cytokine response in macrophage
colony-stimulating factor differentiated macrophages (17). Furthermore, ACPA might
contribute to osteoclastogenesis and the development of joint pathology by facilitating
the proinflammation polarization of macrophages (18, 19). Here, we showed that the
expression of IRF5 was positively correlated with the ACPA titer in the RA synovium
(Fig. 1a). Consistently, in patients with ACPA⁺ RA, the expression level of IRF5 protein
was obviously elevated (Fig. 1b). Targeted silencing of IRF5 in SIMs could facilitate

the anti-inflammatory polarization of macrophages (Fig. 6e, i and Supplementary Fig.
26) and thus about the SIM-initiated cascades in ACPA⁺ RA (Fig. 6g, l). Our data
provide evidence for the reestablishment of articular immune homeostasis during
ACPA⁺ RA immunopathogenesis through reprogramming of macrophages to an
immunoregulatory phenotype and reveal the importance of precision therapy based on
ACPA status. Although they exhibiting similar clinical arthritic symptom, the immune
pathogenesis of ACPA⁻ and ACPA⁺ RA patients are quite different (26), and they might
require tailored treatment strategies. In subsequent work, we will systematically study
whether the long-term outcomes differed for these two subsets of RA patients, in the
hope that this lead to stratified treatment in RA.”

**Q36.** Line 67: states that immunosuppressive therapeutics often weaken the immune
system with increased risk of infections. Did you test if compound-treated mice are
more vulnerable to infections?

**A:** We sincerely appreciate this insightful comment and constructive suggestion. In the
revised manuscript, we further evaluated whether nanoimitator-treated mice showed a
weakened immune system with an increased risk of infection following the reviewer’s
suggestion. We have added the following text to our updated submission.

“Next, we examined whether the administration of nanoimitator compromises the
host immune defence against *Candida albicans*. *Candida albicans*, an opportunistic
pathogen with a high fatality rate, has shown an increased infection rate in
immunocompromised individuals under a variety of conditions (27, 28). A dosing
scheme similar to the therapeutic regimen in the CIA mouse model was used: six-week-
old DBA/1J mice received an intravenous injection of the nanoimitator (1 nmol of
siRNA dose per mouse, 10 mg/kg metformin-equivalent dose, $n = 3$) twice a week for
a total of five weeks. Sterile PBS, as a negative control, was injected intravenously to
mice on the same days. Methotrexate (5 mg/kg) were used as positive controls for
immune suppression (29, 30) and injected intravenously to mice twice times a week
from day 28 to 60. For the evaluation of immune defence, nanoimitator-treated and

untreated DBA/1J mice were injected intravenous with 1×10^5 conidia *C. albicans* CA4
after the last administration.

Treatment of mice with nanoimitator induced a marked reduction in the fungal load
in the spleen compared with that in the methotrexate-treated mice (Supplementary Fig.
20a, b). Importantly, the nanoimitator did not affect the proportions of CD19⁺, CD4⁺ or
CD8⁺ lymphocytes in the blood and spleen (Supplementary Fig. 20c), which is often
associated with therapeutics for autoimmune diseases (31). However, methotrexate
reduced the sizes of lymphocyte subpopulations in the blood and spleen. Overall,
administration of the nanoimitator did not significantly affect the normal adaptive
immune responses of the treated mice.”

**Supplementary Figure 20. Assessment of the impact of nanoimitator on the**
 **immune responses of mice to *Candida albicans* infection. a**, Ex vivo culture of *C.*
 *albicans* with spleen lysates from the mice subjected to different treatments. **b**, The
 quantitative results of each group's splenic bacteria load. Data are presented as the mean
 \pm s.d. ($n = 3$ independent experiments). $**P < 0.01$, NS, not significant. **c**, CD19⁺, CD4⁺
 and CD8⁺ cell percentages in the blood and spleen of DBA/1J mice treated with
 nanoimitator, PBS or methotrexate. Data are presented as the mean \pm s.d. ($n = 3$
 independent experiments). $*P < 0.05$, $**P < 0.01$, $***P < 0.001$, $****P < 0.0001$. NS,
 not significant.

**Q37.** Line 80: please stress this is a gain-of-function polymorphism

**A:** We are grateful for the reviewer's insightful comments and constructive suggestion.

According to the reviewer's suggestion, in the revised submission, we have stressed the

"Gain-of-function polymorphisms in the IRF5 gene".

**Q38.** Figure 1B: requires co-straining with a macrophage marker to link IRF5
expression specifically to macrophages.

**A:** We are grateful for the reviewer's insightful comments and constructive suggestion.

We have added the data in the revised manuscript as follows.

**Supplementary Figure 1.** Immunostained sections of IRF5 in synovium from RA
patients and CIA mice. IRF5 is green and CD68⁺ or F4/80⁺ macrophages are colored
red. Nuclei are stained with DAPI (blue). Scale bar = 50 μ m.

**Q39.** Figure 2: Please clarify how the compound concentration was assessed. In the
follow-up experiments, no concentration was specified. Did you try different
concentrations in your experimental setup? Please comment on that.

**A:** We are grateful for the reviewer's insightful comments. The concentration of the
nanoimitator was determined by the encapsulation efficiency and loading capacity of
the drug in the nanoformulation. The encapsulation efficiency of the siRNA was
calculated as the ratio of the amount of siRNA encapsulated in the nanoimitator to the
total amount of siRNA fed for encapsulation. The loading content of the siRNA in the
nanoimitator was calculated as the ratio of the amount of siRNA encapsulated in the

nanoimitator to the total amount of nanoimitator including the siRNA. Both the
encapsulation efficiency and loading content were quantified by using siIRF5^{Cy5.5} for
encapsulation. Similarly, the encapsulation efficiency and loading capacity of
metformin in nanoimitator were determined in by UV-Vis spectrometer at $\lambda = 233$ nm
as described previously.

In our pilot study, we varied the siRNA concentration to quantify the encapsulation
efficiency of the nanoformulation, and the related information has been added to the
revised submission as follows.

“In terms of siRNA loading, the encapsulation efficiency was quantified by
incorporating Cy5.5-labeled siRNA at increasing concentrations (Supplementary Fig.
5). It was demonstrated that siRNA could be incorporated with high efficiency over a
wide range of inputs, and siIRF5@EINI nanoparticles were fabricated using 20 μ M
siRNA, 30 mM $Zn(NO_3)_2 \cdot 6(H_2O)$, and 30 mM metformin for subsequent studies.”

**Supplementary Figure 5.** Encapsulation efficiency of siRNA inside siIRF5@EINI at
various siRNA inputs. Data are presented as the mean \pm s.d. ($n = 3$ independent
experiments).

**Q40.** Figure 2E: This graph appears to be inserted in the wrong place? There is an
underlying image?

**A:** Thank you for the detailed comment. We apologize for the mistake in arranging the
order of Fig. 2e in our original submission. We have updated Fig. 2e in our revised
submission.

**Q41.** Figure 3, line 192: figure legend is missing

**A:** Thank you for the detailed comment. We have added the figure legend for the
experiments.

**Q42.** Figure 5, line 285: please specify the strain that was used

**A:** We appreciate the reviewer's detailed suggestion. The strain of mice used for the
CIA model has been defined in the revised manuscript.

"A collagen-induced arthritis (CIA) murine model based on DBA/1J mice was first
established as previously reported."

**Q43.** Supplementary Figure 11: please stratify WBCs into lymphocytes, neutrophils,
and monocytes. It seems that in the control group, there is a non-significant increase in
the number of WBCs. How do you explain this?

**A:** We sincerely appreciate this insightful comment and constructive suggestion.
According to the reviewer's suggestion, in the revised submission, we have updated
Supplementary Fig. 11a, and the counts of differential white blood cells (neutrophils,
lymphocytes and monocytes) were determined.

"No adverse effect was detected in the hematological assay of the siIRF5@EINI-
treated mice compared with normal mice in terms of white blood cell (WBC) counts,
including counts of neutrophils, lymphocytes and monocytes (Supplementary Fig. 19a)."

**Supplementary Figure 19a.** During the experimental cycle, mice were followed until
sacrifice on day 60, and white blood cell (WBC) counts, including counts of neutrophils,

lymphocytes and monocytes, were assessed. Data are presented as the mean \pm s.d. ($n =$
3 independent experiments). * $P < 0.05$, NS, not significant.

As is well known, Rheumatoid arthritis often exhibits variable disease activity
over time with exacerbations (flares) and periods of low disease activity (32, 33). This
likely results in supra- or sub-WBC levels during periods of high or low disease activity,
respectively. As such, the mean WBC count may show no differences between the
control group and the siIRF5@EINI group. The counts of differential white blood cells
(neutrophils, lymphocytes and monocytes) intuitively reflect the change in WBC counts.

**Q44.** Supplementary Figure 15: Angiogenesis effects are unclear. Please add additional
markers like CD31

**A:** We thank the reviewer for this constructive suggestion. The IHC staining images of
the angiogenesis marker CD31 are included in the updated version of Supplementary
Fig. 27a.

**Supplementary Figure 27a.** Representative IHC staining images of the angiogenesis
marker CD31 in the synovium from mice receiving the indicated treatment. The arrows
indicate typical CD31⁺ microvessels. Scale bar = 100 μ m. ($n = 5$ biologically
independent animals per group).

9. Discussion: The authors should cover the following points:

**Q45.** Potential use of the compound for other disease contexts, potential of the same
drug design targeting other genes, including other disease-relevant transcription factors

**A:** We sincerely appreciate the reviewer's specific suggestion. In the Discussion of our
original submission, we have added the statement "Our work therefore provides a
regulatory strategy for macrophage heterogeneity for RA reversible treatment, which

may be extended to various macrophage-involving autoimmune diseases, such as
atherosclerosis, idiopathic pulmonary fibrosis and inflammatory bowel disease.”

We apologize for the insufficient discussion in our original submission. According
to the reviewer’s suggestion, we have added the discussion in the revised manuscript
as follows.

“Although we concentrated on IRF5 in this study, it is logical to extend this
approach to other diseases harboring common transcription factors alterations, such as
hypoxia-inducible factor (HIF)-1 α or runt-related transcription factor 1 (RUNX1).”

**Q46.** line 460 states that metformin in the nano-initiator can prevent complex I-derived
ROS but according to Figure 4f it is also changing the expression of both anti- and pro-
inflammatory markers. This needs to be discussed.

**A:** We are grateful for the reviewer’s detailed comment and constructive suggestion. In
the context of autoimmune or inflammatory diseases, metformin has
immunosuppressive effects (4). Studies suggest that metformin interferes with key
immunopathological mechanisms involved in systemic autoimmune diseases, such as
ROS production, macrophage polarization and cytokine synthesis (5). Complex I-
derived ROS can drive proinflammatory cytokine production. Metformin acts as a
mitochondrial complex I inhibitor that can prevent complex I-derived ROS and thus
reprogram mitochondrial metabolism in macrophages to elicit anti-inflammatory gene
expression (34, 35).

In the Discussion of our original submission, we have added the statement “In
addition, the metformin in the nanoimitator can prevent complex I-derived ROS and
thus promote the SIMs-targeted reprogramming. Our data indicates the dissociated
metformin from the EINI significantly decreased the production of ROS in SIMs
(Supplementary Fig. 10), which further blocked the ROS-mediated inflammatory signal
and was conducive to reconstructing the intra-articular immune homeostasis
synergistically. ” We apologize for the insufficient discussion in our original submission.
We have added the discussion in the revised manuscript as follows.

“In addition, metformin interferes with key immunopathological mechanisms
involved in systemic autoimmune diseases, such as ROS production, macrophage
polarization and cytokine synthesis. The metformin in the nanoimitator can prevent
complex I-derived ROS and thus promote the SIMs-targeted reprogramming (34, 35).
Our data indicate that the metformin dissociated from the EINI significantly decreased
the production of ROS in SIMs (Supplementary Fig. 16), which further blocked the
ROS-mediated inflammatory signal (Fig. 4f) and was conducive to reestablishing the
intra-articular immune homeostasis synergistically.”

**Reviewer #2**

This is a novel, thorough and exciting study using an siIRF5-laden efferocytosis-
inspired nanoimitator (siIRF5@EINI) which consists of siRNA directed against IRF5,
coated with PtdSer, cloaked with low molecular weight heparin.

**Q1.** The authors show that systemic administration preferentially distributes to the paw.
Biodistribution studies were however performed over a limited period of time, over 48
958 hours. Given that the actual studies were conducted over a longer time period,
biodistribution over longer periods of time, albeit with limitations on cell tracking
understood would be a valuable addition. Also were all tracking studies done in non
CIA mice to understand if the preferential homing and retention in the paw was induced
by the Collagen induced RA?

**A:** We sincerely appreciate this insightful comment and constructive suggestion. We
agree with the reviewer that the study of biodistribution over longer periods of time
would be a valuable addition in this study. In our original submission, we tracked the
trafficking profile of nanoimitators in vivo by frozen sectioning (Fig. 5e and Fig. 5f)
and immunofluorescence staining (Fig. 5g) of the synovium on 72 h after systemic
administration of nanoimitators. These data indicated that the nanoimitator could
efficiently accumulate intra-articularly and specifically target synovial macrophages in
situ. On the basis of the inflamed joint accumulation study in vivo, we injected
nanoimitator two times a week into each mouse, which yielded a marked advantage in
reducing side effects compared with daily administration. Therefore, the drug has the

potential to increase patient compliance in clinical application.

According to the reviewer's suggestion, in the revised submission, we have also
added the data for tracking biodistribution over longer periods of time in the updated
version of Fig. 5a, b, c. Meanwhile, we determined the biodistribution of the EINI in
unimmunized control mice, which is shown below.

"As shown in Fig. 5a, the nanoimitator was efficiently deposited in the inflamed
joints of the diseased mice 96 h post intravenous administration. In contrast, the
fluorescence signal arising from the paws of nanoimitator-injected non-arthritis mice
reached a maximum in the initial 3 h and decayed very rapidly (Supplementary Fig. 17).
The major organs and paws of the CIA mice from each group were collected 96 hours
post-injection and subjected to fluorescence analysis *ex vivo*. siIRF5@EINI was
prominently enriched in the arthritic area of the CIA mice (Fig. 5b). Quantification of
the fluorescent intensity further confirmed a significant increase in mean fluorescence
intensity (MFI) in siIRF5@EINI -treated mice compared with that in the free drug-
treated group (Fig. 5c)."

**Fig. 5. a** In vivo fluorescence images of CIA models taken at different time points post-
 injection with Dir-labeled EINI and free Dir. **b** Fluorescence images of the excised
 major organs and paws harvested from the mice at 96 h post injection. H, Heart; Li,
 Liver; S, Spleen; Lu, Lung; K, Kidneys; P, Paws. **c** Quantification of fluorescence
 intensity in the inflamed joints and major organs of CIA mice. Data are the mean \pm s.d.
 ($n = 3$ independent experiments). $*P < 0.05$.

**Supplementary Figure 17. In vivo biodistribution of nanoimitators in non-**
 **arthritic DBA/1J mice. a** Fluorescence images of non-arthritic DBA/1J mice receiving
 an intravenous injection of Dir-labeled nanoimitator. **b** Fluorescence intensity of the

nanoimitator in the joints of the non-arthritic DBA/1J mice as a function of time. Data
are the mean \pm s.d. ($n = 3$ independent experiments).

**Q2.** Did the authors obtain FLS from human synovium from healthy ($n=3$) vs. RA ($n=7$)
donors? There is no detail in the Materials and Methods about the derivation of the FLS.

**A:** We thank the reviewer for this comment and insightful suggestion. In our original
submission, to verify the specific uptake of nanoimitator by macrophages, murine
fibroblast-like synoviocytes (FLSs) were chosen as the control cells in a coculture
system. We apologize for the ambiguous description in our original submission. To
avoid any misunderstanding, we have highlighted that murine FLSs were used for
coculture experiments in our revised submission. In the updated Supplemental
Information of the revised submission, we have added the detailed methods for the
experiments.

“Murine FLSs were isolated from the knee joint synovium of CIA mice (36). Briefly,
mouse synovial tissues were isolated from knee joints, and synoviocyte suspensions
were minced and digested in collagenase for 4 h at 37°C. Cell suspensions were passed
through a 70- μ m cell strainer and placed in tissue culture dishes containing DMEM
supplemented with 10% FBS. FLSs between passages 4 and 8 were used for further
analysis. Human FLSs were isolated from primary synovial tissue obtained from 3
patients with RA who met the American College of Rheumatology (formerly, the
American Rheumatism Association) revised criteria and had undergone total joint
replacement surgery or synovectomy, as previously described. Informed consent was
obtained from all patients, and the study protocol was approved by the Shandong First
Medical University Ethics Committee. Pure FLSs ($>90\%$ CD90⁺/ $<1\%$ CD14⁺) were
identified by flow cytometry using antibodies against the fibroblast marker CD90 and
the macrophage marker CD14”

Supplementary Figure 10. Characterization FLSs. Flow cytometry using antibodies against the fibroblast marker CD90 and the macrophage marker CD14 led to the identification of pure FLSs (>90% CD90⁺ / <1% CD14⁺).

**Q3.** In coculture experiments were human FLS cocultured with murine macrophages?

This is in Figure 3. Why were human FLS cocultured with murine macrophages? It would make more sense to use human macrophages, differentiated from peripheral monocytes.

**A:** We are grateful for the reviewer's insightful comments and constructive suggestion.

In Figure 3d of the original submission, the coculture experiments were murine FLSs cocultured with murine macrophages. We apologize for the ambiguous description in our original submission. To avoid any misunderstanding, we have highlighted that murine FLSs were used for coculture experiments in our revised submission.

We agree with the reviewer that using human macrophages, differentiated from peripheral monocytes, in coculture experiments is one of the most important data points in this study. In the revised manuscript, we further evaluated the specific engulfment of nanoimitators by macrophages using human peripheral blood-derived macrophages (HPBDMs) following the reviewer's suggestion. We have added the following text to our updated submission.

"We further evaluated the specific engulfment of nanoimitators by macrophages in vitro using human FLSs isolated from the synovium of RA patients (Supplementary Fig. 10) and healthy human peripheral blood-derived macrophages (HPBDMs)

harvested and differentiated with consent. Consistently, the results indicated that there
was nearly no uptake of nanoimitators (red) by PKH67-labeled FLSs (green), which
emphasized the specific uptake of nanoimitators by macrophages (Supplementary Fig.
11).”

**Supplementary Figure 11.** Cell-specific uptake of the nanoimitator by human
peripheral blood macrophages in a coculture pattern with human FLSs. Scale bar, 50
1055 μm and 10 μm .

**Q4.** I understand the rationale to use H₂O₂ for inducing ROS production. However, the
concentrations of H₂O₂ used are extremely high and not justified. Could the authors
provide justification for such high concentrations, particularly in light of Ransy et al.,
2020 IJMS "Use of H₂O₂ to cause oxidative stress, the catalase issue"

**A:** We are grateful for the reviewer’s insightful comments. We agree with the reviewer
that direct application of H₂O₂ to cells results in the generation of oxygen (O₂) because
of the presence of catalase, and it may complicate the interpretation of experiments. In
our original submission, the use of H₂O₂ for inducing ROS production in the in vitro
assay was to deshield LMWHs to re-expose the PtdSer corona and thereby enhance the
uptake of the PtdSer-coronated core in an efferocytosis-like manner rather than
triggering cellular oxidative stress. Previous reports indicate that at a concentration of
0.1 mM, H₂O₂ can efficiently cleave the ROS linker in vitro (37, 38). As such, we chose
0.1 mM H₂O₂ to pretreat the nanoimitator and then incubated it with the cells. In the
Materials and Methods of our original submission, we have added the statement
“macrophages were treated with siIRF5@EINI that were pre-conditioned with 0.1mM
of H₂O₂.” We apologize for the ambiguous description in our original submission. To
avoid any misunderstanding, we have highlighted the method of incubating ROS-
pretreated nanoimitators with cells in the in vitro assay in our revised submission.

**Q5.** The effect of neutrophil adhesion on HUVEC monolayer (Fig 4B and Suppl Fib 8)
are not very convincing. A very small percentage of neutrophils seem to attach to the
HUVEC layer to begin with and there is a small reduction in this signal by addition of
siRF5@EINI, but not to the extent noted with the addition of the P-selectin antagonist.

**A:** We thank the reviewer for the detailed comment. In Fig. 4b and Supplementary Fig.
8 of our original submission, the data for siRF5@EINI blocking the P-selectin initiated
cell adhesion cascade was determined by CLSM and flow cytometry. The results
showed that the adhesion of neutrophils to HUVECs was weakened after preincubation
with the nanoimitator.

To more thoroughly investigate the influence of the nanoimitator on
neutrophil/HUVEC adhesion and quantitative analysis of neutrophils adhered to
HUVEC monolayers, in the revised submission, we further verified the blocking effect
of the nanoimitator on neutrophil-HUVEC adhesion in the updated version of
Supplementary Figure 8, which is shown below.

“HUVECs (2×10^5) were seeded in 6-well plates and pre-stimulated by incubation
for 6 h with TNF- α (10 ng/ml). Then the cells were treated with RPMI medium, P-
selectin antagonist (Human P-selectin/CD62P antibody, R&D Systems) and the
nanoimitator for 30 min. Activated neutrophils (1×10^6) were labeled with 5 μ M
calcein-AM and seeded on top of the treated endothelial monolayer. After 30 min at
37 °C under gentle shaking, each well was washed twice with PBS to remove the non-
adhered cells and the cells in each well were lysed with 2 mL dimethyl sulfoxide
(DMSO), and their fluorescence intensities were detected by a fluorospectro-
photometer at Ex = 495 nm and Em = 515 nm. The in vitro adhesion experiment results
further confirmed that neutrophil/endothelial adhesion was weakened after incubation
with the nanoimitator (Supplementary Fig. 14).”

**Supplementary Figure 14. Quantitative analysis of neutrophils adhered to**

**HUVEC monolayers.** Neutrophils were stained with calcein-AM. The cells in each

well were lysed with 2 mL dimethyl sulfoxide (DMSO), and their fluorescence

intensities were detected by a fluorospectro-photometer at Ex = 495 nm and Em = 515

1105 nm. Data are presented as the mean ± s.d. ($n = 3$ independent experiments). **** $P <$

0.0001. NS, not significant.

**Q6.** The authors are to be commended on performing toxicology studies at 60 days.

However the lung pathology as shown in Suppl Fig 11c with siRF@EINI seems more

aberrant, but there is little to no discussion on this.

**A:** We thank the reviewer for the detailed comment. To better evaluate the

histopathological changes of major organs, we have updated Supplementary Figure 11c,

and histology sections are shown in a larger format as follows.

“Histopathological examinations of lung sections indicated that siIRF5@EINI

treated mice had mild inflammation around the airways and interstitial infiltration of

inflammatory cells compared with control group (Supplementary Fig. 19c). These

histopathological changes may attributed to inflammation-related cytokine decreases at

the systemic level (Fig. 6n). Importantly, compared to normal mice, the treatment with

siIRF5@EINI did not result in any overt tissue damage.”

**Supplementary Figure 19c.** Histological sections of major organs on the 60th day.
 Scale bar, 50 μ m. ($n = 3$ biologically independent animals per group).

**Q7.** There are numerous grammatical mistakes and typos in the document. A thorough
 edit should be performed. LMWH is often mis-abbreviated as LWMH, for instance.

**A:** We thank the reviewer for this detailed comment. We have thoroughly double-
 checked the manuscript. In the revised manuscript, the typographical and grammatical
 errors have been corrected.

**Reviewer #3**

This manuscript introduces a nanoparticle therapy for rheumatoid arthritis. Anti-
 inflammation drugs (siIRF5 and Metformin) are surrounded by an ROS sensitive
 heparin shell (stealth in circulation and targeting of P selectin overexpressed on
 subsynovial capillary endothelium) and a phosphatidylserine corona. The PtdSer
 corona mediates uptake via efferocytosis by synovial inflammatory macrophages with
 a goal of anti-inflammation polarization.

Overall the nanoparticle platform is innovative and in vitro data supports that the goals
 of each nanoparticle component were met. A reduction in IRF5, CXCL1 and MPO exist
 in the inflamed joint, plus altered macrophage phenotype and reduced ankle diameter
 were shown in treated mice using a collagen-induced arthritis murine model. Further
 the nanoparticle was able to protect against bone erosion.

Minor:

**Q1.** Grammatical errors need correcting.

**A:** We thank the reviewer for this detailed comment. We have thoroughly double-
checked the manuscript. In the revised manuscript, the typographical and grammatical
errors have been corrected.

**Q2.** Fig. 1 legend is missing a description of part “c”

**A:** We sincerely appreciate the reviewer’s specific comments. According to the
reviewer’s suggestion, in the revised submission, we have added the description for
Figure 1c, which is shown below.

“The siIRF5-carrying efferocytosis-inspired nanoimitator (siIRF5@EINI)
consisted of a drug-based core with an oxidative stress-responsive phosphatidylserine
(PtdSer) corona and a shell composed of P-selectin-blocking motif, low molecular
weight heparin (LMWH). With the shielding of LMWH, siIRF5@EINI was endowed
with the stealth properties in the circulation, enhanced retention in inflamed regions,
and a blocking function of P-selectin that retards the articular trafficking of neutrophils.
Upregulated ROS triggered shell exfoliation and the PtdSer corona was then exposed.
In an efferocytosis-like manner, the PtdSer-coroneted core was in turn phagocytosed by
SIMs, which synergistically terminated SIM-initiated pathological cascades and
serially reestablished intra-articular immune homeostasis, conferring a
chondroprotective effect.”

**Q3.** Page 6, line 142-144 incorrectly states that the DLS is determined using TEM

**A:** We appreciate the reviewer’s detailed comment. We apologize for the mistake in
describing that the DLS was determined using TEM. We have corrected the mistake in
the revised submission.

“Under transmission electron microscopy (TEM), the obtained nanoimitator
showed a spherical morphology (Fig. 2b). Dynamic light scattering (DLS)
measurements revealed that the nanoimitator had a hydrodynamic diameter of ~72 nm
(Fig. 2c).”

**Q4.** Fig. 2c, yellow axis label is hard to see

**A:** We thank the reviewer for this constructive suggestion. In the revised manuscript,
we have optimized the letter size in related Figures, and the font size and images are
more uniform and clearer.

**Q5.** Fig.2d, x and y-axis labels are missing

**A:** We appreciate the reviewer's detailed comment. In the revised submission, the x-
and y-axis labels are indicated in the updated version of Fig. 2d.

**Q6.** Text on page 7 states that data in Fig. 2g shows ROS elimination of the nanoimitator
(does this mean removal of heparin and if yes, is this measured)?

**A:** We are grateful for the reviewer's insightful comments. The nanoimitator consisted
of a drug-based core with an oxidative stress-responsive phosphatidylserine (PtdSer)
corona and a shell composed of a P-selectin-blocking motif, low molecular weight
heparin (LMWH). While the nanoimitator is inert under normal physiological
conditions, the ROS-abundant inflamed joint microenvironment triggers shell
exfoliation to realize phagocytosis of the PtdSer-coroneted core by SIMs. In our
original submission, we have added the statement "The ROS-responsive exfoliation of
EINI was further determined with the H₂O₂ scavenging assay." According to the
reviewer's suggestion, we have also added the data on the detection of exfoliated
heparin in the revised submission. The amount of LMWH in each solution was
determined by the conventional colorimetric method (39), which is shown below.

"The ROS-treated nanoimitator exhibited a much higher LMWH release efficiency
(Supplementary Fig. 8), which may be attributed to ROS-responsive exfoliation of the
nanoimitator."

**Supplementary Figure 8. The release of LMWH from the nanoimitator in PBS**
 **with or without 0.1 mM H₂O₂.** Data are presented as the mean \pm s.d. ($n = 3$
 independent experiments). * $P < 0.05$, **** $P < 0.0001$.

**Q7.** Fig. 3b shows data from activated FLSs. Are these cells also stimulated with TNF
 (text just states that macrophages are activated)

**A:** We sincerely appreciate the reviewer’s detailed comment. We agree with the
 reviewer that FLSs should also be stimulated with TNF- α in in vitro experiments.
 Notably, in our original submission, the FLSs used in the in vitro assay were indeed
 TNF- α activated. In the Supporting Information of our original submission, we have
 added the statement “BMDMs or FLSs were seeded in 6-well tissue culture plates at
 50% confluency and cultured overnight. The cell culture medium was changed, and 10
 1210 ng/ml recombinant human TNF- α was added for stimulation for 4 h.” We apologize for
 the ambiguous description in our original submission. To avoid any misunderstanding,
 we have highlighted that TNF- α activated FLSs were used for in vitro experiments in
 our revised submission.

“With fibroblast-like synoviocytes (FLSs) activated with TNF- α as a control.”

**Q8.** Page 15, line 331, please summarize the dosing in text (number and amount of
 NPs/injection)

**A:** We sincerely appreciate the reviewer’s specific comments and constructive
 suggestion. In the Supporting Information of our original submission, we have added

the statement “The CIA animals were randomly assigned into four groups (1 nmol of
siRNA dose per mouse, 10 mg/kg metformin-equivalent dose, $n = 5$). 200 μ L PBS (G1);
free siIRF5 solution (G2); siIN.C@EINI (G3) and siIRF5@EINI (G4) were
administered by tail vein twice a week from day 28 to 60.” We apologize for the
ambiguous description in our original submission. To avoid any misunderstanding, we
have highlighted the dosage of the nanoimitator used in the in vivo assay in our revised
submission as follows.

“To study therapeutic efficacy in CIA mice, the treatments began on day 7 after the
second immunization. On day 28, the mice presented arthritis symptoms and were
randomly divided into four groups ($n = 5$ mice in each group). Four different treatments
(200 μ L PBS, free siIRF5 solution, siIN.C@EINI and siIRF5@EINI) were
administered via the tail vein twice a week from day 28 to 60. The dose for each
formulation was 1 nmol of siRNA dose per mouse, 10 mg/kg metformin-equivalent
dose.”

**References**

- 1. D. Wang *et al.*, Mechanisms of lung disease development in rheumatoid arthritis.
*Nature Reviews Rheumatology* **15**, 581-596 (2019).
- 2. E. Schurgers *et al.*, Pulmonary inflammation in mice with collagen-induced
arthritis is conditioned by complete Freund's adjuvant and regulated by
endogenous IFN- γ . **42**, 3223-3234 (2012).
- 3. Y. Lee *et al.*, Hyaluronic acid–bilirubin nanomedicine for targeted modulation
of dysregulated intestinal barrier, microbiome and immune responses in colitis.
*Nature Materials* **19**, 118-126 (2020).
- 4. F. Marcucci, E. Romeo, C. A. Caserta, C. Rumio, F. Lefoulon, Context-
Dependent Pharmacological Effects of Metformin on the Immune System.
*Trends in Pharmacological Sciences* **41**, 162-171 (2020).
- 5. U. Francesco *et al.*, Metformin and Autoimmunity: A "New Deal" of an Old
Drug. **9**, 1236 (2018).
- 6. H. M. Khojah, S. Ahmed, M. S. Abdel-Rahman, A.-B. Hamza, Reactive oxygen
and nitrogen species in patients with rheumatoid arthritis as potential
biomarkers for disease activity and the role of antioxidants. *Free Radical*
*Biology and Medicine* **97**, 285-291 (2016).
- 7. J. S. Courtenay, M. J. Dallman, A. D. Dayan, A. Martin, B. Mosedale,
Immunisation against heterologous type II collagen induces arthritis in mice.
*Nature* **283**, 666-668 (1980).

- 8. H. S. Kim *et al.*, DJ-1 controls bone homeostasis through the regulation of
osteoclast differentiation. *Nature Communications* **8**, 1519 (2017).
- 9. J. Wu *et al.*, TNF antagonist sensitizes synovial fibroblasts to ferroptotic cell
death in collagen-induced arthritis mouse models. *Nature Communications* **13**,
676 (2022).
- 10. K. Segawa, S. Nagata, An Apoptotic ‘Eat Me’ Signal: Phosphatidylserine
Exposure. *Trends in Cell Biology* **25**, 639-650 (2015).
- 11. P. R. Hoffmann *et al.*, Phosphatidylserine (PS) induces PS receptor-mediated
macropinocytosis and promotes clearance of apoptotic cells. **155**, 649-660
(2001).
- 12. J. Vincelette *et al.*, Gait analysis in a murine model of collagen-induced arthritis.
*Arthritis Research & Therapy* **9**, R123 (2007).
- 13. W. He, N. Kapate, C. W. Shields IV, S. J. A. d. d. r. Mitragotri, Drug delivery to
macrophages: a review of targeting drugs and drug carriers to macrophages for
inflammatory diseases. **165**, 15-40 (2020).
- 14. C. Deng *et al.*, Targeted apoptosis of macrophages and osteoclasts in arthritic
joints is effective against advanced inflammatory arthritis. **12**, 1-15 (2021).
- 15. V. Malmström, A. I. Catrina, L. Klareskog, The immunopathogenesis of
seropositive rheumatoid arthritis: from triggering to targeting. *Nature Reviews*
*Immunology* **17**, 60-75 (2017).
- 16. L. Laurent *et al.*, Fcγ receptor profile of monocytes and macrophages from
rheumatoid arthritis patients and their response to immune complexes formed
with autoantibodies to citrullinated proteins. *Annals of the Rheumatic Diseases*
**70**, 1052 (2011).
- 17. C. Clavel, L. Ceccato, F. Anquetil, G. Serre, M. Sebbag, Among human
macrophages polarised to different phenotypes, the M-CSF-oriented cells
present the highest pro-inflammatory response to the rheumatoid arthritis-
specific immune complexes containing ACPA. *Annals of the Rheumatic*
*Diseases* **75**, 2184 (2016).
- 18. S. Fukui, N. Iwamoto, A. Takatani, T. Igawa, A. J. F. i. I. Kawakami, M1 and
M2 Monocytes in Rheumatoid Arthritis: A Contribution of Imbalance of
M1/M2 Monocytes to Osteoclastogenesis. **8**, 1958 (2017).
- 19. W. Zhu *et al.*, Anti-Citrullinated Protein Antibodies Induce Macrophage Subset
Disequilibrium in RA Patients. *Inflammation* **38**, 2067-2075 (2015).
- 20. T. Krausgruber *et al.*, IRF5 promotes inflammatory macrophage polarization
and TH1-TH17 responses. **12**, 231-238 (2011).
- 21. A. Takaoka *et al.*, Integral role of IRF-5 in the gene induction programme
activated by Toll-like receptors. **434**, 243-249 (2005).
- 22. M. Weiss *et al.*, IRF5 controls both acute and chronic inflammation. **112**, 11001-
11006 (2015).
- 23. R. Dieguez-Gonzalez *et al.*, Association of interferon regulatory factor 5
haplotypes, similar to that found in systemic lupus erythematosus, in a large
subgroup of patients with rheumatoid arthritis. **58**, 1264-1274 (2008).
- 24. R. R. Graham *et al.*, A common haplotype of interferon regulatory factor 5

- (IRF5) regulates splicing and expression and is associated with increased risk of systemic lupus erythematosus. **38**, 550-555 (2006).
- 25. X. M. Matthijssen, E. Niemantsverdriet, T. W. Huizinga, A. H. J. P. m. van der
Helm-van Mil, Enhanced treatment strategies and distinct disease outcomes
among autoantibody-positive and-negative rheumatoid arthritis patients over 25
1305 years: a longitudinal cohort study in the Netherlands. **17**, e1003296 (2020).
- 26. X. Wu *et al.*, Single-cell sequencing of immune cells from anticitrullinated
peptide antibody positive and negative rheumatoid arthritis. *Nature*
*Communications* **12**, 4977 (2021).
- 27. M. Bassetti, M. Peghin, J.-F. Timsit, The current treatment landscape:
candidiasis. *Journal of Antimicrobial Chemotherapy* **71**, ii13-ii22 (2016).
- 28. J. Kim, P. Sudbery, *Candida albicans*, a major human fungal pathogen. *The*
*Journal of Microbiology* **49**, 171-177 (2011).
- 29. G. R. Burmester, J. E. Pope, Novel treatment strategies in rheumatoid arthritis.
*The Lancet* **389**, 2338-2348 (2017).
- 30. J. S. Smolen *et al.*, EULAR recommendations for the management of
rheumatoid arthritis with synthetic and biological disease-modifying
antirheumatic drugs: 2016 update. *Annals of the Rheumatic Diseases* **76**, 960
(2017).
- 31. P. Zhao *et al.*, Depletion of PD-1-positive cells ameliorates autoimmune disease.
*Nature Biomedical Engineering* **3**, 292-305 (2019).
- 32. D. E. Orange *et al.*, RNA Identification of PRIME Cells Predicting Rheumatoid
Arthritis Flares. *New England Journal of Medicine* **383**, 218-228 (2020).
- 33. N. Joshi *et al.*, Towards an arthritis flare-responsive drug delivery system.
*Nature Communications* **9**, 1275 (2018).
- 34. E. L. Mills, L. A. O'Neill, Reprogramming mitochondrial metabolism in
macrophages as an anti-inflammatory signal. *European Journal of Immunology*
**46**, 13-21 (2016).
- 35. D. G. Ryan, L. A. J. O'Neill, Krebs Cycle Reborn in Macrophage
Immunometabolism. *Annual Review of Immunology* **38**, 289-313 (2020).
- 36. T. Honda, E. Segi-Nishida, Y. Miyachi, S. Narumiya, Prostacyclin-IP signaling
and prostaglandin E2-EP2/EP4 signaling both mediate joint inflammation in
mouse collagen-induced arthritis. *Journal of Experimental Medicine* **203**, 325-
335 (2006).
- 37. S. Hu *et al.*, Exosome-eluting stents for vascular healing after ischaemic injury.
*Nature Biomedical Engineering* **5**, 1174-1188 (2021).
- 38. Y. Lu *et al.*, Microthrombus-Targeting Micelles for Neurovascular Remodeling
and Enhanced Microcirculatory Perfusion in Acute Ischemic Stroke. *Advanced*
*Materials* **31**, 1808361 (2019).
- 39. J. Jong, R. A. Wevers, C. Laarakkers, B. J. C. C. Poorthuis, Dimethylmethylene
blue-based spectrophotometry of glycosaminoglycans in untreated urine: A
rapid screening procedure for mucopolysaccharidoses. **35**, 1472-1477 (1989).

REVIEWER COMMENTS

Reviewer #1 (Remarks to the Author):

The authors have made substantial revisions to the manuscript and answered most of the questions raised. We appreciate their efforts. Before accepting the paper following issues needs to be clarified:

Q1: Explain why there is an accumulation of the compound in the liver, kidney, and tail? Can the accumulation of the compound in the liver and kidney be explained by the degradation of the compound in these tissues? Can the accumulation of the compound in the base of the tail be explained by the adjuvant injection in the CIA model inducing tail inflammation?

Q2: How do you explain accumulation in the paws of non-arthritis mice upon nano-imitator administration (Supplementary figure 17)? Lungs are usually inflamed in the model. What do you mean by: "due to the lack of the impaired microvasculature, nano-imitators may undergo rapid clearance from the lung compared with inflamed joints". The text describes the EINI is enriched in the joint synovium due to a strong dysregulation (leakage) of the synovial microvasculature. What is meant by dysregulation?

Q23/Q35:

- Figure 1C can be removed and merged with Figure 7
- Remove Figure 1. Make it a full supplementary figure.
- During the in vitro studies / in vivo studies no distinction was made between ACPA+ or ACPA-. No conclusions can be drawn regarding the better clinical response in ACPA+ subsets. This would require additional clinical studies.
- Supplementary figure 1 is more valuable compared to Figure 1b please swap. Add representative images of HC synovium in Supplementary Figure 1.

Q43: Lymphocyte numbers are non-significantly decreased in the siRF5@EINI condition compared to controls (in a non-arthritis context). How do you explain this difference?

Additional comments:

- Please refer to established facts in the present tense: e.g. L40 "When systemically administered, the LMWH on the EINI first bind"; L117: "and the PtdSer corona are"
- Figure 2d: clarify figure legend, explain you're measuring annexin binding.
- L217: different patients were used for FLS and macrophage isolation. How do you avoid GvHD?
- Same antibodies were used for both humans and mice? Please add an additional table in the supplementary figures summarising all AB used.
- L275: A wrong conclusion is made. Showing the nano-imitator binding the vessel does not prove it is the mechanism why there is less tethering of neutrophils. Rephrase sentence!
- L393: Change comparison. "No differences were seen compared to control"
- Figure 5d can be removed
- Supplementary Figure 10: CD14 is 3.51% in mice so no <1% as described in the figure legend.

Reviewer #2 (Remarks to the Author):

The authors have addressed my comments.

Reviewer #3 (Remarks to the Author):

The authors have adequately addressed my concerns.

**Response to reviewers' comments:**

We appreciate very much the time and effort the reviewers dedicated to reviewing this
manuscript. According to the additional constructive feedback from the reviewers, we
have made corresponding changes in the revised manuscript and Supplementary
Information. Below, please find our point-by-point responses to the reviewers'
comments.

**Reviewer #1**

The authors have made substantial revisions to the manuscript and answered most of
the questions raised. We appreciate their efforts. Before accepting the paper following
issues needs to be clarified:

**Q1: Explain why there is an accumulation of the compound in the liver, kidney,
and tail? Can the accumulation of the compound in the liver and kidney be
explained by the degradation of the compound in these tissues? Can the
accumulation of the compound in the base of the tail be explained by the adjuvant
injection in the CIA model inducing tail inflammation?**

**Response:** We sincerely appreciate the reviewer's detailed comment. We agree with the
reviewer that the nanoimitator undergoes degradation and excretion from the liver and
kidneys postinjection. The liver is the major organ for the metabolism and clearance of
nanoimitators. In this case, the nanoformulation exhibited delayed clearance in these
organs. A critical concern for nanoparticle-based therapies is whether the delivery of
nanoparticles may cause toxicity in normal tissues. The potential toxicity of the
systemically administered nanoformulation was also tested in our original submission.
Histological analysis of organs revealed the biosafety of the nanoimitator in vivo
(Supplementary Fig. 19c).

The reason for the accumulation of the nanoimitator at the base of the tail within 48
27 h postinjection is that tail inflammation was induced by the adjuvant injection during
establishment of the collagen-induced arthritis model. In contrast, the nanoimitator
exhibited much faster clearance in the tail of non-arthritic mice.

**Q2: How do you explain accumulation in the paws of non-arthritic mice upon**
**nano-imitator administration (Supplementary figure 17)? Lungs are usually**
**inflamed in the model. What do you mean by: “due to the lack of the impaired**
**microvasculature, nano-imitators may undergo rapid clearance from the lung**
**compared with inflamed joints”. The text describes the EINI is enriched in the**
**joint synovium due to a strong dysregulation (leakage) of the synovial**
**microvasculature. What is meant by dysregulation?**

**Response:** We thank the reviewer for this comment. The retention of nanoimitators in
the paws of non-arthritic mice has mainly been attributed to the non-specific biological
distribution of nanoimitators in the systemic circulation. Consistent with previous
findings (1, 2), the nanoformulation could accumulate in the paws of non-arthritic mice;
however, paw clearance in these mice was relatively rapid (<12 h). This was mainly
due to the rapid elimination of the nanoformulation from the bloodstream 12 h after the
intravenous injection.

In rheumatoid arthritis, the synovial microvasculature is highly dysregulated, which
may lead to vascular leakage (3, 4). During synovial inflammation, activated
endothelial cells can lose their polarity, detach and protrude into the vessel lumen,
thereby disrupting the pericyte layer (5). The resulting poorly organized vessel is
dysfunctional. In addition, neovascularization facilitates immune cell infiltration and
subsequent activation of resident synovial cells, transforming the normal relatively
acellular synovium into an invasive tumour-like pannus (5, 6).

**Q23/Q35:**

- - **Figure 1C can be removed and merged with Figure 7**
- - **Remove Figure 1. Make it a full supplementary figure.**
- - **During the in vitro studies / in vivo studies no distinction was made between**
**ACPA+ or ACPA-. No conclusions can be drawn regarding the better clinical**
**response in ACPA+ subsets. This would require additional clinical studies.**
- - **Supplementary figure 1 is more valuable compared to Figure 1b please swap.**

**Add representative images of HC synovium in Supplementary Figure 1.**

**Response:** We sincerely appreciate the reviewer's detailed comments and constructive
suggestions. Following the reviewer's suggestion, Figure 1c has been merged with
Figure 7, and Figure 1a, b has been transferred to the Supplementary Information and
merged with Supplementary Figure 1. Additionally, we have added representative
immunofluorescence images of healthy control (HC) synovium in Supplementary
Figure 1.

**Supplementary Figure 1.** Immunostained sections of IRF5 in synovium from healthy
control (HC). IRF5 is green and CD68⁺ or F4/80⁺ macrophages are colored red. Nuclei
are stained with DAPI (blue). Scale bar = 50 μ m.

We agree with the reviewer that additional clinical studies are required to determine
the therapeutic benefits of the nanoimitator for ACPA⁺ RA. In our study, we
demonstrated that targeted silencing of IRF5 in synovial inflammatory macrophages
(SIMs) could terminate SIM-initiated pathological cascades and serially reestablish
intra-articular immune homeostasis, conferring a chondroprotective effect.

ACPAs in the inflamed synovium have been shown to drive a strong
proinflammatory cytokine response in macrophages and contribute to
osteoclastogenesis and the development of joint pathology (7-9). Here, we showed that
the expression of IRF5 was positively correlated with the ACPA titer in the RA
synovium. Consistently, in patients with ACPA⁺ RA, the expression level of protein
IRF5 was obviously elevated. Considering the fact that IRF5 is a master regulator

involved in defining the classical inflammatory phenotype of macrophages and
translates various signals related to SIMs in the RA synovium, we proposed that
targeted silencing of IRF5 in SIMs may be an efficient strategy to facilitate anti-
inflammatory polarization of macrophages and thus abort SIM-initiated cascades in
ACPA⁺ RA. In a collagen-induced arthritis mouse model, a model that recapitulates
many clinical and immunological features of RA, including the generation of
autoantibodies (ACPAs and rheumatoid factor) and autoreactive T cells (10, 11),
treatment with the nanoimitator exhibited potential therapeutic potency to control
ACPA⁺ RA.

In contrast to the well-characterized pathogenic mechanisms of ACPA⁺ RA, the
etiology of ACPA⁻ RA remains largely unknown (12). The therapeutic potency of the
nanoimitator in ACPA⁻ RA requires additional studies. Importantly, a better
understanding of the immunopathogenesis of ACPA⁻ RA will help in evaluating the
therapeutic potency of the nanoimitator, and pave the way for personalized treatment.
The current study data provide evidence for the reestablishment of articular immune
homeostasis during ACPA⁺ RA immunopathogenesis through reprogramming of
macrophages into an immunoregulatory phenotype, and we hope that this will lead to
stratified treatment for RA.

**Q43: Lymphocyte numbers are non-significantly decreased in the siRF5@EINI**
**condition compared to controls (in a non-arthritic context). How do you explain**
**this difference?**

**Response:** We appreciate the reviewer's comments. Indeed, compared with normal
mice (in a non-arthritic context), siRF5@EINI-treated mice exhibited comparable
lymphocyte counts. This result suggested that our nanoimitator possessed excellent
biocompatibility. More importantly, the nanoimitator not only halted the progression of
rheumatoid arthritis, but also concomitantly preserved normal adaptive immunity
(Supplementary Fig. 20). Compared to PBS-treated mice, treatment with siRF5@EINI
resulted in a significant reduction in lymphocyte counts. The reason for the difference

in lymphocyte counts between the PBS and siIRF5@EINI groups was that
siIRF5@EINI treatment led to an effective reduction in inflammation at the systemic
level (Fig. 6n).

**Additional comments:**

• **Please refer to established facts in the present tense: e.g. L40 “When systemically**
**administered, the LMWH on the EINI first bind”;** L117: **“and the PtdSer corona**
**are”**

**Response:** We appreciate the reviewer’s detailed suggestion. Following the reviewer’s
suggestion, we have used the present tense when describing established facts in the
revised manuscript.

• **Figure 2d: clarify figure legend, explain you’re measuring annexin binding.**

**Response:** We sincerely appreciate the reviewer’s specific comments and constructive
suggestion. Following the reviewer’s suggestion, we have clarified the figure legend
for Figure 2d.

“Flow cytometry analysis of PtdSer-presenting with FITC-Annexin V treatment.”

• **L217: different patients were used for FLS and macrophage isolation. How do**
**you avoid GvHD?**

**Response:** We appreciate the reviewer’s comments. Previous reports indicate that
donor monocyte-derived macrophages promote graft-versus-host disease (GVHD) (13,
14). However, alloactivated macrophages mediate cytotoxicity to epidermal cells
through a specific cytokine milieu in vitro, and cytotoxicity to epidermal cells in vitro
occurs in a dose-dependent manner (15). To avoid potential alloactivated macrophage-
mediated cytotoxicity, FLSs were cocultured with macrophages at a ratio of 1:1 in our
coculture system. Additionally, the isolated cells were used for only in vitro
experiments. Therefore, our experiments avoided GVHD mediated by alloactivated
macrophages to some extent.

• **Same antibodies were used for both humans and mice? Please add an additional**
 **table in the supplementary figures summarising all AB used.**

**Response:** We thank the reviewer for this detailed comment. The source of the antibody
 was selected strictly based on the species of the sample to be tested. In the updated
 Supplemental Information of the revised submission, we have summarized the
 antibodies used in our experiments.

Supplementary Table 2. Summary of antibodies

Antibody	Application	Catalog No.; Supplier	Dilution
Rabbit anti-CD68	IF	76437T; Cell Signaling	1:400
Rabbit anti-F4/80	IF	Ab6640; Abcam	1:200
Rabbit anti-iNOS	IF	Ab178945; Abcam	1:250
Rabbit anti-CD206	IF	Ab64693; Abcam	1:500
Rabbit anti-IRF5	Western	10547-1-AP; proteintech	1:5000
Rabbit anti-IRF5	IHC	76983S; Cell Signaling	1:1000
Rabbit anti- Myeloperoxidase	IHC	ab188211; Abcam	1:8000
FITC anti-mouse F4/80	FC	123107; BioLegend	1:200
PerCP-Cy5.5 anti-mouse CD11b	FC	101227; BioLegend	1:100
FITC anti-mouse Ly-6G	FC	127605; BioLegend	1:100
APC/Cyanine7 anti-mouse CD45	FC	103115; BioLegend	1:100
PE/Dazzle™ 594 anti-mouse CD80	FC	104738; BioLegend	1:200
PE anti-mouse CD206	FC	141705; BioLegend	1:100
PE anti-mouse CD45	FC	103106; BioLegend	1:200
APC/Cyanine7 anti-mouse CD3	FC	100222; BioLegend	1:200
APC anti-mouse CD4	FC	100412; BioLegend	1:200
PE/Cyanine7 anti-mouse CD8a	FC	100722; BioLegend	1:200
APC anti-mouse CD19	FC	152409; BioLegend	1:200

PE anti-mouse CD90	FC	Ab24904; Abcam	1:100
FITC anti-mouse CD14	FC	123307; BioLegend	1:100
PE anti-human CD90	FC	328109; BioLegend	1:100
FITC anti-human CD14	FC	325603; BioLegend	1:100

*IF: Immunofluorescence; IHC: Immunohistochemistry; FC: Flow Cytometry;

• **L275: A wrong conclusion is made. Showing the nano-imitator binding the vessel**
 **does not prove it is the mechanism why there is less tethering of neutrophils.**
 **Rephrase sentence!**

**Response:** We sincerely appreciate this insightful comment and constructive
 suggestion. We apologize for the misleading conclusion in our original submission.
 According to the reviewer’s suggestion, we have rephrased the sentence as follows.

“Through its capability to bind to P-selectin overexpressed on the endothelium in
 RA subsynovial capillaries, the nanoimitator thus interrupted the endothelium tethering
 of neutrophils and prevented their subsequent extravasation.”

• **L393: Change comparison. “No differences were seen compared to control”**

**Response:** We appreciate the reviewer’s detailed comment. Following this suggestion,
 we have rewritten the sentence as follows.

“No difference was seen in the fungal load in the spleen between mice treated with
 the nanoimitator and those in the PBS group.”

• **Figure 5d can be removed**

**Response:** We thank the reviewer for this comment. Following the reviewer’s
 suggestion, we have removed Figure 5d from the revised manuscript.

• **Supplementary Figure 10: CD14 is 3.51% in mice so no <1% as described in the**
 **figure legend.**

**Response:** We sincerely appreciate the reviewer’s detailed comment. We apologize for

the mistake in describing the purity of mouse FLSs. We have corrected the mistake in
the revised submission.

“Flow cytometry using antibodies against the fibroblast marker CD90 and the
macrophage marker CD14 led to the identification of pure FLSs. Mouse FLSs (>90%
CD90⁺, <4% CD14⁺) and human FLSs (>90% CD90⁺, <1% CD14⁺) were positive for
the fibroblast marker CD90 and negative for the macrophage marker CD14.”

**References**

- 1. H. Liang *et al.*, Cationic nanoparticle as an inhibitor of cell-free DNA-induced
inflammation. *Nature Communications* **9**, 4291 (2018).
- 2. Y. He *et al.*, Drug targeting through platelet membrane-coated nanoparticles for
the treatment of rheumatoid arthritis. *Nano Research* **11**, 6086-6101 (2018).
- 3. U. Fearon, M. M. Hanlon, A. Floudas, D. J. Veale, Cellular metabolic
adaptations in rheumatoid arthritis and their therapeutic implications. *Nature*
*Reviews Rheumatology* **18**, 398-414 (2022).
- 4. Z. Tu *et al.*, Design of therapeutic biomaterials to control inflammation. *Nature*
*Reviews Materials* **7**, 557-574 (2022).
- 5. U. Fearon, M. Canavan, M. Biniiecka, D. J. Veale, Hypoxia, mitochondrial
dysfunction and synovial invasiveness in rheumatoid arthritis. *Nature Reviews*
*Rheumatology* **12**, 385-397 (2016).
- 6. M. Biniiecka *et al.*, Dysregulated bioenergetics: a key regulator of joint
inflammation. *Annals of the Rheumatic Diseases* **75**, 2192 (2016).
- 7. L. Laurent *et al.*, Fcγ receptor profile of monocytes and macrophages from
rheumatoid arthritis patients and their response to immune complexes formed
with autoantibodies to citrullinated proteins. *Annals of the Rheumatic Diseases*
**70**, 1052 (2011).
- 8. C. Clavel, L. Ceccato, F. Anquetil, G. Serre, M. Sebbag, Among human
macrophages polarised to different phenotypes, the M-CSF-oriented cells
present the highest pro-inflammatory response to the rheumatoid arthritis-
specific immune complexes containing ACPA. *Annals of the Rheumatic*
*Diseases* **75**, 2184 (2016).
- 9. S. Fukui *et al.*, M1 and M2 Monocytes in Rheumatoid Arthritis: A Contribution
of Imbalance of M1/M2 Monocytes to Osteoclastogenesis. *Frontiers in*
*Immunology* **8**, (2018).
- 10. M. E. M. El Shikh *et al.*, Extracellular traps and PAD4 released by macrophages
induce citrullination and auto-antibody production in autoimmune arthritis.
*Journal of Autoimmunity* **105**, 102297 (2019).
- 11. G. R. Meehan *et al.*, Preclinical models of arthritis for studying immunotherapy
and immune tolerance. *Annals of the Rheumatic Diseases* **80**, 1268 (2021).
- 12. K. Li, M. Wang, L. Zhao, Y. Liu, X. Zhang, ACPA-negative rheumatoid arthritis:

- From immune mechanisms to clinical translation. *eBioMedicine* **83**, 104233
(2022).
- 13. K. A. Alexander *et al.*, CSF-1-dependant donor-derived macrophages mediate
chronic graft-versus-host disease. *The Journal of Clinical Investigation* **124**,
4266-4280 (2014).
- 14. K. Wu *et al.*, The gut microbial metabolite trimethylamine N-oxide aggravates
GVHD by inducing M1 macrophage polarization in mice. *Blood* **136**, 501-515
(2020).
- 15. L. Jardine *et al.*, Donor monocyte-derived macrophages promote human acute
graft-versus-host disease. *The Journal of Clinical Investigation* **130**, 4574-4586
(2020).

REVIEWERS' COMMENTS

Reviewer #1 (Remarks to the Author):

The authors have now sufficiently responded to all concerns. Congratulations on their important work.